# New insights into the vertical structure of the September 2015 dust storm employing 8 ceilometers and auxiliary measurements over Israel

Leenes Uzan[1,2], Smadar Egert[1], Pinhas Alpert[1]

[1] Department of Geosciences, Raymond and Beverly Sackler Faculty of Exact Sciences, Tel-Aviv University, Tel Aviv, 6997801, Israel.
[2] The Israeli Meteorological Service, Beit Dagan, Israel.

Correspondence to: Leenes Uzan (Leenesu@gmail.com)

# Abstract

On 7 September 2015 an unprecedented and unexceptional extreme dust storm struck the Eastern Mediterranean (EM) basin. Here, we provide an overview of the previous studies and describe the dust plume evolution over a relatively small area, i.e., Israel. This study presents vertical profiles provided by an array of 8 ceilometers covering Israeli shore, inland and mountain regions. We employ multiple tools including spectral radiometers (AERONET), ground particulate matter concentrations, satellite images, global/diffuse/direct solar radiation measurements and radiosonde profiles. Main findings reveal that the dust plume penetrated Israel on the 7 September from the northeast in a downward motion to southwest. On 8 September, the lower level of the dust plume reached 200 m above ground level, generating aerosol optical depth (AOD) above 3, and extreme ground particulate matter concentrations up to ~10,000 μm $m^{-3}$. A most interesting feature on 8 September was the very high variability in the surface solar radiation in the range of 200-600 W $m^{-2}$ (22 sites) over just a distance of several hundred km in spite of the thick dust layer above. Furthermore, 8 September shows the lowest radiation levels for this event. On the following day, the surface solar radiation increased, thus enabling a late (between 11-12 UTC) sea breeze development mainly in the coastal zone associated with a creation of a narrow dust layer detached from the ground. On 10 September the AOD values started to drop down to ~ 1.5, the surface concentrations of particulate matter decreased as well as the ceilometers aerosol indications (signal counts) although CALIPSO revealed an upper dust layer remained.

# 1. Introduction

An exceptionally extreme dust storm prevailed over the Eastern Mediterranean (EM) on September 2015. The Israeli meteorological service (IMS) declared the dust storm to be extraordinary as it occurred on early September (7-10 September), extended over a time span of 100 hours creating extreme ground level particulate matter (PM) concentrations (e.g. 100 times above the hourly average of PM10 in Jerusalem). On 7 September, prior to the penetration of the dust storm over Israel, IMS reported (http://www.ims.gov.il/IMS/CLIMATE; in Hebrew)  a heat wave over Israel causing harsh weather conditions of  80-90% relative humidity, 42 ℃ in valleys, 38 ℃ in mountains. On 8 September, visibility decreased below 3 km and consequently, inland aviation was prohibited until 9 September (Fig.1).

Concurrently, severe ground level PM concentrations resulted with a public warning from outdoor activities issued by the environmental protection ministry. Finally, on 11 September, as visibility increased, the IMS confirmed the dust storm ended, whereas the heat wave was over two days later, on 13 September, subsequent to a profound change in weather conditions. The PM concentrations declined to values measured prior to the dust storm (http://www.svivaaqm.net/Default.rtl.aspx; in Hebrew) only on 14 September, though the AERONET measurements (https://aeronet.gsfc.nasa.gov) stationed in central and southern Israel reveal that the aerosol optical depth (AOD) resumed to values prior to the dust storm only on 17 September.

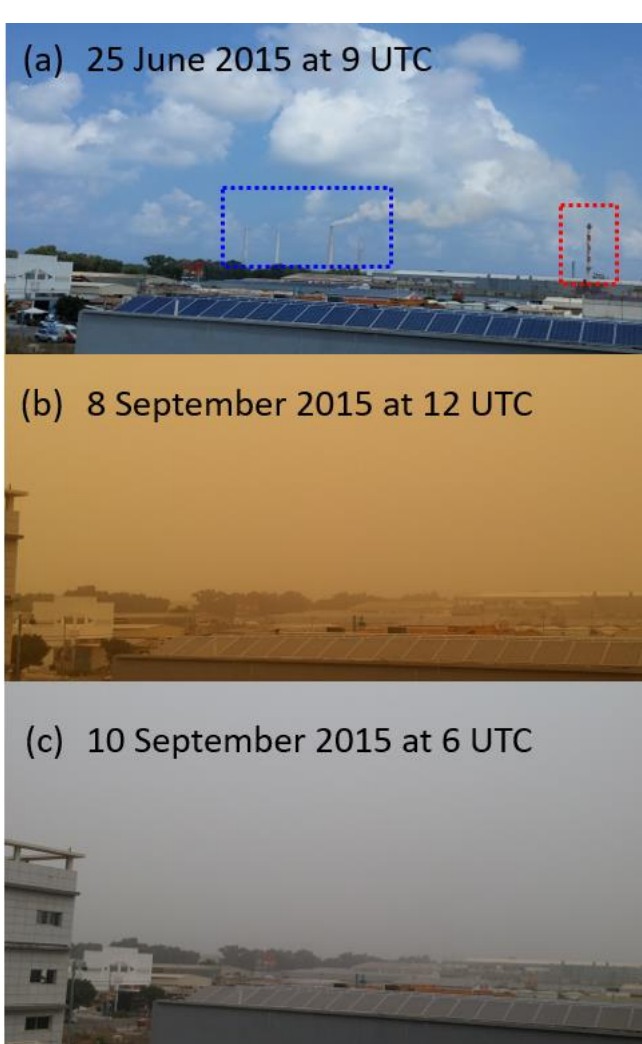

Figure 1. Photographs taken from the central coast of Israel, adjacent to the Hadera ceilometer, 3.5 km southeast to stacks of a power plant (indicated by a blue rectangle) and 600 m north to a factory stack (indicated by a red rectangle). The photographs were taken prior to the dust storm, on 25 July 2015 (a), and during the dust storm, on 8 September 2015 (b) and 10 September 2015 (c). Notice that the stacks that are visible on a clear day (a) are invisible during the dust storm (b, c).

Investigation of the mechanisms leading the severe dust storm was performed by Gasch et al. (2017) using a state of the art dust transport model ICOsahedral Nonhydrostatic (ICON) with the Aerosol and Reactive Trace gases (ART) (Rieger, et al., 2015). The model concentrated on the EM with one global domain (40 km grid spacing, and 90 vertical levels from 20 m to 75 km) and 4 nested grids (20, 10, 5 and 2.5 km grid spacing and 60 vertical levels from 20 m to 22.5 km). Simulations were done for three consecutive days from 6-8 September. Model results delineated an unusual early incidence of an active Red Sea Trough (Fig.2; Alpert et al, 2004) over Mesopotamia, followed by meso-scale convective systems over the Syrian-Iraqi border generating three cold-pool outflows. On the night between 5 and 6 September, a convective system fueled by an inflow along the eastern side of the Red Sea Trough, moved northeast over the Turkish-Syrian border region. The convective system intensified overnight and generated a first weak cold pool outflow on 6 September. After sun rise, an increase of surface wind speeds caused dust pick up over Syria. The atmospheric instability over the Syrian-Iraqi border created a second convective cold pool outflow from the Zagros mountain range west into Syria. The gust from the second cold pool outflow ignited a third cold pool outflow at 20 UTC which moved southerly along the eastern flank of the Red Sea Trough. On 7 September 10 UTC, rainfall and an increase of surface wind speeds north-west of Syria strengthened the third cold pool outflow leading to transportation of enormous dust emissions (up to 5 km) southwest. By nightfall of 7 September, the aged second cold pool outflow merged with the third cold pool outflow, over Jordan and southwestern Syria. After midnight, between 7 and 8 September, the dust transported over Israel. Model simulations were compared to in-situ measurements and satellite images: visible electromagnetic spectrum from Moderate Resolution Imaging Spectroradiometer (MODIS: https://modis.gsfc.nasa.gov/) aboard the Aqua satellite; AOD from the Terra satellite; RGB dust product from the Spinning Enhanced Visible and InfraRed Imager (SEVIRI) upon the Meteosat Second Generation (MSG) satellite; Total attenuated backscatter from Cloud-Aerosol Lidar upon the Infrared Pathfinder Satellite Observations (CALIPSO: https://www-calipso.larc.nasa.gov/). Investigation over Israel employed measurements from ground level meteorological stations (3 sites) and PM measurements (3 sites). Results revealed the model lacked sufficient development of a super critical flow, which in effect produced the excessive surface wind speeds. Eventually, this misled the forecast of the dust advection southwest into Israel.

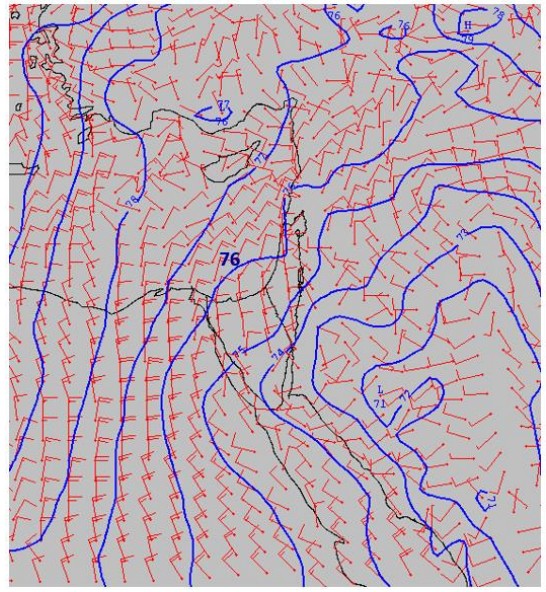


Figure 2.  925 mb map from 7 September 2015 12 UTC of geopotential height of 1 dm interval (blue
lines, the 76 dm line is passing over Israel) and wind (red arrows, 10 KT each line). Source: IMS from
UKMO British Met Office model.


The fact that forecast models did not succeed in predicting this outstanding dust event motivated
Mamouri et al. (2016) to study its origin and development. Their research presented dust load
observations in the Cyprus region. Luckily, at the time of the dust storm, an EARLINET (European
Aerosol Research lidar Network: https://www.earlinet.org/) Raman lidar stationed in Limassol provided
vertical dust profiles and valuable optical dust properties of backscatter, extinction, lidar ratio and linear
depolarization ratio. They analyzed the optical thickness (AOT) and Angström exponent derived from
the MODIS Aqua satellite. MODIS Aqua AOT measurements were compared to the Limassol lidar
observations, AOD measurements from two AERONET sites (Cyprus and Israel) and ground level PM10
concentration from four Cyprus sites. On 7 September, EARLINET lidar observations measured two dust
layers (extending up to 1.7 km ASL and between 1.7-3.5 km ASL).  The dust particle extinction
coefficient measured in Limassol had reached 1000 $Mm^{-1}$ followed by high PM10 concentration of 2000
$\mu m\ m^{-3}$. Extreme values over Limassol, were reported on 8 September as MODIS Aqua AOT
observations exceeded 5 (assuming overestimation up to 1.5) and hourly PM10 concentration of about
8,000 $\mu m\ m^{-3}$ (with uncertainties in the order of 50%).   Unfortunately, on 8 September, the lidar was
intentionally shut down to avoid potential damage to the instrument. Lidar observations indicated another
dense dust outbreak (1-3 km ASL) reaching Limassol on 10 September, also visible by AOT MODIS
Aqua. The researchers concluded the scale of the dust storm features was too small for global and regional
dust transport models. They presumed that the initiation of the dust plume was due to an intense dust

storm (haboob) in northeastern Syria and northern Iraq, leading to vigorous downbursts which consequently pushed huge amounts of dust and sand to the atmosphere. The lidar observations indicated a double layer structure of the dust, 1.5 and 4 km ASL, pointing to multiple dust sources.

Solomos et al., (2016) continued the investigation of the formation and mechanism of the dust storm over Cyprus by a high regional atmospheric model of the integrated community limited area modeling system (RAMS-ICLAMS). The model simulations focused on the generation of the dust storm on 6 and 7 September. Model results were fine-tuned by observations from EARLINET lidar stationed in Limassol, radiosonde data from five sites (Cyprus, Israel, Jordan, and two from Turkey) and satellite imagery from MSG SEVIRI and CALIPSO CALIOP. The model was set to three grid space domains: an external grid of 12X12 km, (over the EM) an inner set at 4X4 km (over northern EM) and 2X2 km grid for cloud resolving (over northeastern Syria). The vertical structure consisted of 50 terrain following levels up to 18 km. The researchers estimated a strong thermal low over Syria was followed by convection activity over the Iraq-Iran-Syria-Turkey borderline.  Combined with land use changes (aftermath of the war held in Syria), these conditions manufactured the extreme dust storm. The model succeeded to describe the dust westward flow of a haboob containing the dust previously elevated over Syria also observed by MSG SEVIRI and EARLINET lidar. However, there were some inaccuracies in the quantification of dust mass profiles. The researchers attributed the model discrepancies to the limited ability of the model to properly resolve dust and atmospheric properties (e.g. change of land use and intense downward mixing).

Pu and Ginoux, (2016) examined the connection between the natural climate variability (the Pacific decadal oscillation) and the dust optical depth (DOD) in Syria between the years 2003-2015. DODs were derived by the deep blue algorithm (Hsu et al., 2013) aerosol product from MODIS Terra and MODIS Aqua satellite (10 km resolution). AODs were estimated by the European Centre for Medium-Range Weather Forecasts (ECMWF) reanalysis model (horizontal resolution of 80 km and 37 vertical levels) and produced by the Geophysical Fluid Dynamics Laboratory (GFDL) Atmospheric model (AM3) (Donner et al., 2011). In addition, the AM3 model produced mass distribution and optical properties of aerosols, their chemical production, transport, and dry or wet deposition. Comparison of the model AODs, AERONET AOD measurements and DODs from satellite observations revealed the model underestimated the AODs particularly in the EM. The authors assumed that the soil moisture parameter in the model were not set properly resulting in the AOD dissimilarities.

The impact of the conflict in Syria on the aridity of the region and therefore, a possible direct impact on the generation of the September dust storm was examined by Parolari et al., (2016). The researchers conducted simulations using the Advanced Research Weather Research and Forecasting (WRF-ARW) model from 30 August 2015 to 10 September 2015 over the EM. The model consisted of two nested domains (9 and 3 km grid spacing and 35 vertical levels). Daily and monthly AOD data from MODIS were computed by the deep blue algorithm over land. The monthly average of September 2015 vegetation status in the region was estimated by MODIS normalized difference vegetation index (NDVI). Historical data was divided into two periods: none-drought (2001-2006) and drought (2007-2010). Wind shear stress was calculated to estimate wind erosion. Main findings reveal that the enhanced dust uplift and transportation of the September 2015 dust storm was due to meteorological conditions rather than the land-use changes attributed to the civil conflict in Syria. WRF simulations revealed northwesterly winds west of the low pressure zone in the Syrian-Iraqi border were associated with dust storms in the Middle East (Rao et al., 2003). The source of elevated dust concentrations over the EM coast on 7 and 8 September were attributed to the cyclone front movement. On 6 September low level winds (700 hPa) were opposite to the northwesterly high level winds (300 hPa), consequently generating enhanced surface shear stress and transported re-suspended PM westward. Furthermore, based on the past 20 years, the Israeli summer of 2015 was unusually dry and hot and therefore enabled easier updraft of dust soil increasing the probability of dust emissions.

Jasmin (2016) compared the dust plume aerosol content provided by MSG SEVIRI observations, to the generation of the dust storm produced by the open source Meteoinfo model (Wang, 2014). The Meteoinfo model was based on meteorological variables from ECMWF. The model revealed a formation of two simultaneous dust storms, from northern Syria and the Egyptian Sinai desert, as a result of updrafts created by low pressure systems.

The aforementioned studies (summarized in Table 1) focused on the generation of the dust storm in the Syria region based on transport models, satellite imagery and in situ measurements. In our study we focus on the evolution of the dust plume over Israel in the lower atmosphere based on an array of 8 ceilometers and auxiliary instruments described in Sect. 2. The list of instruments includes; ceilometers, PM measurements, AERONET, radiosonde, solar radiation and satellite imagery. Sect. 3 presents the delineation of the dust plume spatial and temporal scheme from 7 - 10 September 2015. We discuss and compare the results between the different measurements. Conclusions and main findings of the dust plume progress in the lower atmosphere are given in Sect. 4.

## 2. Instruments

### 2.1 Ceilometers

Lidars are widely used for aerosol studies (Ansmann et al., 2011; Papayannis et al., 2008) including desert dust characteristics and transport process (Mona et al., 2012). Ceilometers, initially intended for cloud level height detection, are automatic low cost lidars widespread in airports and weather stations worldwide. As single wavelength lidars, ceilometers cannot produce the information aerosol properties such as size distribution, scattering and absorption coefficients. Nevertheless, with improvement of hardware and firmware over the years, ceilometers have become a valuable tool in the study of the atmospheric boundary layer and the vertical distribution of aerosols layers (Haeffelin and Angelini, 2012; Ansmann et al., 2003). Furthermore, in 2013 ceilometers were assimilated in the EUMETNET (European Meteorological Services network) profiling program across Europe (http://eumetnet.eu/activities/observations-programme/current-activities/e-profile/alc-network/). The main research tool in this study is the Vaisala ceilometers type CL31, commonly deployed worldwide.

CL31 is a pulsed elastic micro lidar, employing an Indium Gallium Arsenide (InGaAs) laser diode transmitter of near infrared wavelength (910 nm ±10 nm at 25˚C). In order to compensate the low pulse energy of the laser (hence defined "eye-safe") and to provide sufficient signal to noise ratio, the pulse repetition rate is of 10 kHz (Vaisala ceilometer CL31 user's guide: http://www.vaisala.com). The backscatter signals are collected by an avalanche photodiode (APD) receiver and designed into range corrected signal profiles within a reporting interval of 2-120 s (determined by the user) given in relative unites (signal counts). The ceilometer profiles are automatically corrected by a cosmetic shift of the backscatter signal (to better visualize the clouds base), an obstruction correction (when the ceilometers' window is blocked by a local obstacle) and an overlap correction (to the height where the receiver field of view reaches complete overlap with the emitted laser beam).

Vaisala provides a scaling factor transferring signal counts to attenuated backscatter units by a multiplication factor of $10^{-8}$. The scaling factor was obtained using a calibration procedure operated on a several instruments and cross-checked by signal integral from water clouds. The uncertainty of calibrated attenuated backscatter profile (with a 100% clean window condition) was of ± 10 %. The uncertainty for the estimated attenuated backscatter was of ± 20 % (Münkel C., private communication). However, Kotthaus et al., (2016) emphasize that this internal calibration applied to convert the signal count output

to attenuated backscatter units does not always fully represent the actual lidar constant, therefore, it is not accurate enough for meteorological research. Hence, in this study we defined the ceilometer profiles as range corrected signal profiles in arbitrary units.

Kotthaus et al., (2016) examined the Vaisala CL31 ceilometer by comparing attenuated backscatter profiles from 5 units with different specification of senor hardware, firmware and operation settings (noise, height and time reporting interval). Research findings show the instrument characteristics that affect the quality and availability of the attenuated backscatter profiles are as follows: At high altitudes, a discontinuity in the attenuated backscatter profile is evident at two height points, ~ 4949 and 7000 m. Background signals (instrument related) and cosmetic shift (firmware dependent) tend to be either negative or positive up to 6000m and then switch signs above ~ 6000 m. Below 70 m an overlap correction is applied internally by the ceilometer sensor as well as an obstruction correction (below 50 m). Between 50-80 m hardware related perturbation cause a slight offset in the attenuated backscatter values. The authors advise the user-defined reporting interval should be no shorter than 30s to avoid consecutive profiles partial overlap. Background noise reduction can be achieved by a procedure based on long averaging period at nighttime during a clear atmosphere. A range corrected attenuated backscatter can be derived by the attenuated backscatter profiles during an existence of a stratocumulus cloud.

Weigner et al, (2014) studied different retrieval methods to derive the aerosol backscatter coefficient from the ceilometers' attenuated backscatter profiles based on a comparison to auxiliary collocated instruments such as a Sunphotmeter or a multiwavelength lidar. They focused on calibration methods, the rage detection limitations by the overlap function and the sensitivity of the attenuated backscatter signal to relative humidity. Although, the ceilometer wavelength range (given as 905 ±3 nm) is influenced by water vapor absorption, in the case of aerosol layer detection, water vapor distribution has a small effect on the signal change, indicating the mixed layer height (MLH) or an elevated mixed layer, as the aerosol backscatter itself remains unchanged (Wiegner and Gasteiger, 2015). Consequently, except for a case of a dry layer in a humid MLH, water vapor is unlikely to lead misinterpretation of the aerosol stratification. Fortunately, most algorithms are based on a significant signal slope to define the aerosol layers, therefore, can be determined from uncalibrated ceilometer attenuated backscatter profiles.

In this research, ceilometer array is comprised of 8 units in different sites (Fig 3 and Tables 2-3), 6 of which are owned by a governmental office. The ceilometers are CL31 type apart for ceilometer CL51 stationed in the Weizmann Institute which has a higher backscatter profile range (up to 15.4 km, Münkel et al., 2011). Unfortunately, calibration procedures were not held and maintenance (cleaning of

the ceilometer window) was done regularly only in the Beit Dagan ceilometer. Apart from the Beit Dagan and Weizmann ceilometers we could not retrieve the technical information of firmware and hardware type (Table 4). However, we have been confirmed (personal communication) that the combination of hardware and firmware had been done following Kotthaus et al (2016). The Beit Dagan ceilometer signal count were found to be weaker (up to 800 signal count compared to 10,000 in the other CL31 ceilometers) due to different hardware definitions. Therefore, in order to present the Beit Dagan range corrected signal profiles aligned with the profiles of the other ceilometers (given in Fig. 17), the Beit Dagan range corrected signal values were multiplied by 12.5 (10,000/800). We address the aforementioned limitations of the ceilometers measurements in the first range gates as we refer to the ceilometer signal count from 100 m AGL. Due to the extreme AOD values of the September dust storm, high extinction of the ceilometer signal limited the height of profile analyzed down to 1000 m AGL.

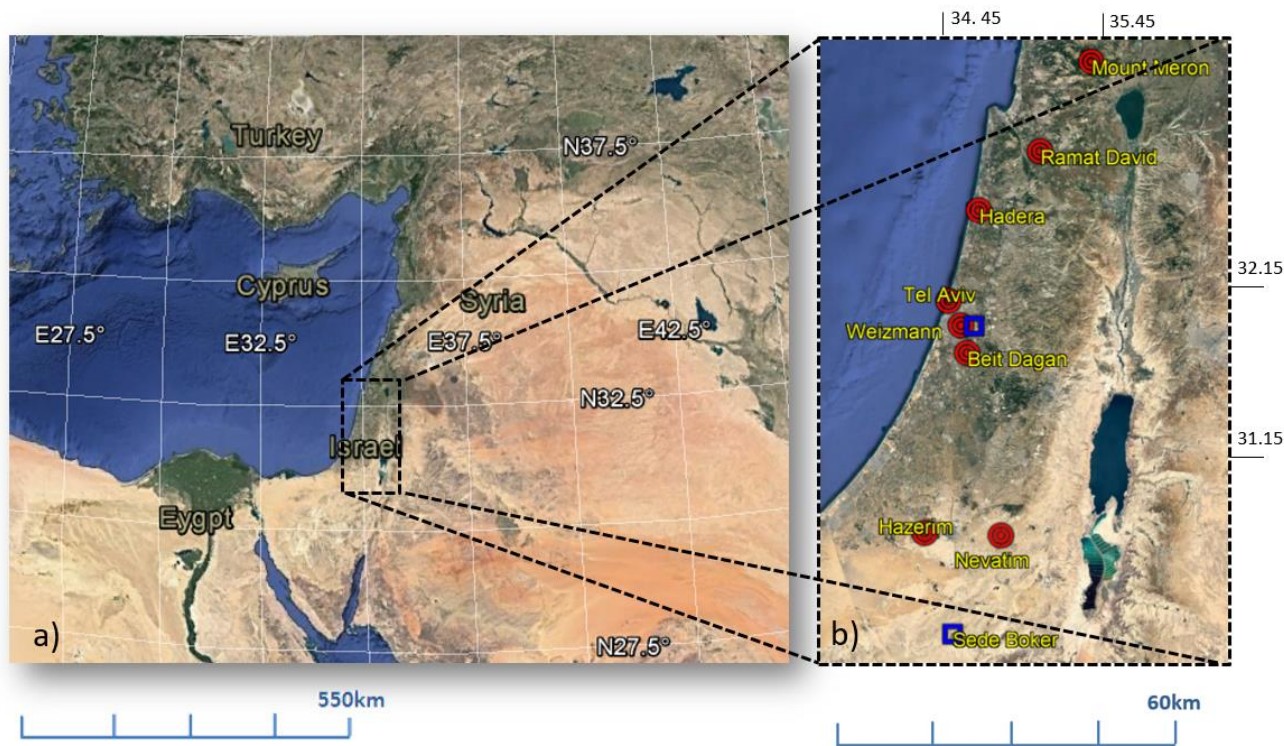

Figure 3. Google Earth map of the large domain (a) and Israel (b) with indications of 8 ceilometer sites (red circle; detail in Table 2) and two AERONET sites (blue square).

## 2.2 Radiosonde

Radiosonde (RS) type Vaisala RS41-SG is launched by the IMS twice a day at 00 UTC and 12 UTC from the Beit Dagan site adjacent to the Beit Dagan ceilometer. The radiosonde produces profiles of humidity, temperature, pressure, wind speed and wind direction. The output files were downloaded from the University of Wyoming site (http://weather.uwyo.edu/upperair/sounding.html, station number 40172). With respect to Stull (1988), the MLH was defined by the RS profiles as the height where an inversion in the temperature was identified along with a significant drop in relative humidity, strong wind shear and an increase in the virtual temperature (Uzan et al., 2016; Levi et al., 2011).

## 2.3 Particulate matter monitors (PM10, PM2.5)

PM monitors are low-volume flow rate Thermo Fisher Scientific type FH 62 C14 (beta attenuation method) and type 1405 TEOM (Tapered Element Oscillating Microbalance method). Both instruments report PM concentration every 5 min. The location of PM measurement sites is given in Tables 5. In the beta attenuation method (https://www3.epa.gov/ttnamti1/files/ambient/inorganic/overvw1.pdf) low-energy beta rays are focused on deposits on a filter tape and attenuated according to the approximate exponential function of particulate mass (i.e., Beer's Law). These automated samples employ a continuous filter tape. The attenuation is measured through an unexposed portion of the filter tape. The tape is then exposed to the ambient sample flow where a deposit is accumulated. The beta attenuation is repeated, and the difference in attenuation between the blank filter and the deposit is a measure of the accumulated concentration. The weighing principle used in the TEOM method (https://tools.thermofisher.com/content/sfs/manuals/EPM-TEOM1405-Manual.pdf) is based on a mass change detected by the sensor as a result of the measurement of a change in frequency. The tapered element at the heart of the mass detection system is a hollow tube, clamped on one end and free to oscillate at the other. If additional mass is added, the frequency of the oscillation decreases and a precision electronic counter measures the oscillation frequency with a 10-second sampling period. An electronic control circuit senses oscillation and adds sufficient energy to the system to overcome lossesd while an automatic gain control circuit maintains the oscillation at a constant amplitude.

## 2.4 AERONET

AErosol RObotic NETwork (AERONET) is multiband photometer with an automatic sun tracking radiometer for direct sun measurements with a spectral range of 340 - 1640 nm wavelengths. The photometer measures the solar extinction in each wavelength to compute aerosol optical depth (Holben et al., 1998). In Israel, AERONET units type CE318-N (https://aeronet.gsfc.nasa.gov) operate in Sede Boker and the Weizmann Institute (Fig. 3). Unfortunately, the unit in Weizmann did not operate between 6-8 September 2015 due to power failure. For this study, data acquisition was comprised of AOD (500 nm wavelength) and Ångstrom exponent (440-870 nm wavelengths) based on AERONET Level 2.0 (cloud screened and quality assured for instrument calibration).

## 2.5 Global, direct and diffuse solar radiation measurements

Global solar radiation is measured by Kipp & Zonen pyranometer type CMP-11 in 22 sites (Fig. 20) operated by the IMS. The pyranometer produces 10 min measurements of the integrated radiation flux (W m$^{-2}$) between 300-3000 nm wavelengths. Diffuse and direct radiation are also measured in Beit Dagan (coastal region, 31 m ASL) and Beer Sheva (southern region, 71 m ASL). For diffuse radiation measurements, a ring is mounted over a pyranometer to avoid direct solar radiation. Direct radiation is measured by a sun tracker pyrheliometer.

## 2.6 Satellite imagery

### 2.6.1 SEVIRI (MSG satellite)

Meteosat Second Generation (MSG) is a new series of European geostationary satellites operated by EUMETSAT (European Organization for the Exploitation of Meteorological Satellites). On board the MSG is a 12-channel Spinning Enhanced Visible and Infrared Imager (SEVIRI) (Roebeling et al., 2006). The combination of red, blue and green (RGB) channels (12-10.8 μm, green:10.8-8.7 μm, blue:10.8 μm, respectively) produce imagery of dust in pink or magenta, dry land in pale blue at daytime and pale green at nighttime. Thick high-level clouds in red-brown tones and thin high-level clouds appear nearly black

(http://oiswww.eumetsat.int/). Access to EUMETSAT imagery is provided online by https://www.eumetsat.int/website/home/Images/RealTimeImages/index.html. Several studies compared AOD from MGS SEVIRI and AERONET measurements (Romano et al., 2013; Bennouna et al., 2009; Jolivet et al., 2008) showed the uncertainty of MSG SEVIRI AOD decreases as AOD rises. For continental aerosol type, errors do not exceed 10 % in viewing zenith angles between 20° and 50. The MSG SEVIRI AOD uncertainty it is expected to be under 15% (Mei et al, 2012) and even higher as the atmospheric AOD increases above 1.5 (EUMETSAT Scientific Validation Report SEVIRI Aerosol Optical Depth (23 Oct 2017). North Africa Sand storm survey (NASCube: http://nascube.univ-lille1.fr) obtains AOD by temperature anomalies based on SEVIRI RGB by evaluating the difference in the emissivity of dust and desert surfaces during daytime.

## 2.6.2 MODIS (Terra and Aqua satellites)

The MODerate resolution Imaging Spectrometer (MODIS) instrument is stationed aboard the Earth Observation System's (EOS) Terra and Aqua polar-orbiting satellites. Terra satellite is on a descending orbit (southward) over the equator at ~ 10:30 local sun time. The Aqua satellite is on an ascending orbit (northward) over the equator at ~ 13:30 local sun time. MODIS performs measurements by 36 channels between 412 -14200 nm whereas the aerosol retrieval makes use of seven channels (646, 855, 466, 553, 1243, 1632 and 2119 nm central wavelength) together with a number of other wavelength bands for screening procedures. Remer et al., (2006) revealed errors of 0.01 in the MODIS surface reflectance will lead to errors on the order of 0.1 in AOD retrieval. However, under conditions of high AOD (>1.5) the uncertainty is expected to rise.

## 2.6.3 CALIOP (CALIPSO satellite)

The Cloud-Aerosol Lidar with Orthogonal Polarization (CALIOP) is a two-wavelength polarization lidar (1064 and 532nm) aboard the Cloud-Aerosol Lidar and Infrared Pathfinder Satellite Observations (CALIPSO) that performs global profiling of aerosols and clouds in the troposphere and lower stratosphere. CALIOP measures signal returns in a large range, from aerosol-free region up to strong cloud returns. The CALIOP profiles are given below 40 km for the 532 nm channel and below 30 km for the 1064 nm channel. Data acquisition in this research was based on level 2 version 4-10 CALIPSO

product of 532 nm wavelength with a spatial resolution of 5 km (20N-50N, 20E-50E) and vertical
resolution of 60 m (limited up to 6 km).

# 3. Results and discussion


The following description of the dust event will proceed chronologically from 7 to 10 September and
include main findings from the different measuring instruments (Sect. 2). The order of the instruments
described follow the most interesting features revealed, not necessarily in the same order for each day.
We provide 2D ceilometer plots (height vs. time) presenting the extreme dust plume decent only from ~
1km ASL due to the ceilometer limitation to detect signals from higher levels (explained in Sect. 2.1).
Unlike the high resolution ceilometers, CALIPSO overpass above Israel was available only on 10
September 2015 revealing dust distribution in various levels up to 5 km ASL.

## 3.1   7 September 2015


On 7 September, images from MODIS Aqua (Fig. 4a) and MODIS Terra (Fig. 4e) taken between
07:20-12:10 UTC show that the dust plume progressed from northeast in a near-circular motion over the
Mediterranean Sea. The penetration of the dust plume to Israel was indicated by AERONET Sede Boker
site at ~ 05 UTC by an increase in AOD along a decrease in the Angström exponent (Fig. 5). The
connection between decreasing Angström exponent values and the dust plume was pointed out by
Mamouri et al., (2016) which presented values of linear depolarization ratio between 0.25-0.32 on 7 and
10 September, indicting the dominance of mineral dust. In addition, an increase in the PM concentration
started at ~ 05 UTC (not shown) reaching the highest hourly values of 107 μg m$^{-3}$ PM2.5 (Table 5) and
491 μg m$^{-3}$ PM10 (Table 6) only in the Jerusalem elevated sites (~ 800 m ASL) and only at 22 UTC. This
17-hour gap is shown by the ceilometers' plots (Fig. 6-12) as a downward motion of the dust plume from
~ 04 UTC in all measuring sites except for the elevated Mount Meron site (1150 m ASL, Fig. 13).
Following Gasch et al (2017) cold pool outflows concept, the exception of Mount Meron site is supported
by the MSG-SEVIRI picture (Fig. 14) showing that the first dust plume was fragmented (Fig.14, red
arrow) and the second dust plume (Fig.14, black arrow) had not passed over Israel before 12 UTC. The
deep blue scale evident in all Mount Meron ceilometer plots (Fig. 13) indicate total attenuation

distinctively from 7 September ~ 14 UTC to 8 September ~ 16 UTC. However, due to the complexity of the dust plume progress (further shown) and the weak signal counts shown up to 3.5 km ASL (before 7 September ~ 14 UTC and after 8 September ~ 16 UTC), the assumption of a total attenuation throughout the period analysed is uncertain. Unfortunately, we did not have auxiliary measurements from the Mount Meron region to justify our assumptions.

While the MSG-SEVIRI picture at 12 UTC shows AOD values to be under 1 in most parts of Israel (Fig. 15), the PM concentrations on ground level were found to be bearable (up to 105 µg m$^{-3}$ PM2.5 and 305 µg m$^{-3}$ PM10, mainly in the Jerusalem elevated sites). At 12 UTC, Beit Dagan radiosonde profiles show a characteristic MLH of 700 m ASL (Fig. 16). Moreover, at 23 UTC the formation of clouds was indicated by ceilometers' profiles (Fig. 17 a) at 400 m ASL in the shoreline (Tel Aviv, 5 m ASL) and up to ~700 m ASL in the elevated southern site (Nevatim, 400 m ASL). Clouds are identified by the peak shape of the ceilometer profiles (Uzan et al, 2016) and the high range corrected signal of 10 $^{-1}$ m$^{-1}$ sr$^{-1}$ which in this case was 4 orders of magnitude higher than the range corrected signal of the dust plume (shown in Fig 17 b-c). Hourly solar radiation measurements (Fig. 18, see 7 September daily plot) from Beit Dagan (central site) and Beer Sheva (southern site) show a significant effect of the dust plume by a decrease in direct radiation along with an increase of diffuse radiation.

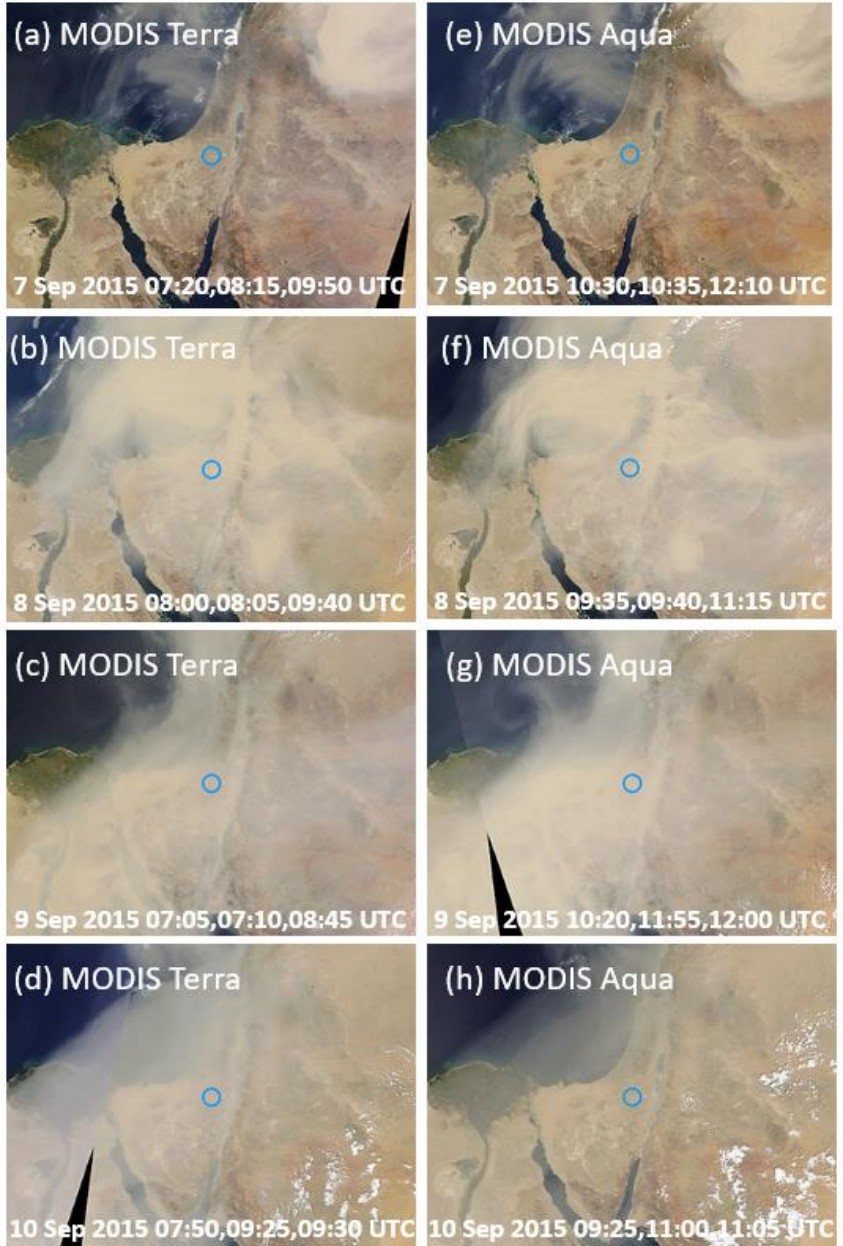

Figure 4. Pictures from MODIS terra (a-d) and MODIS Aqua (e-h). The date and time of overpass are indicated on each figure. The blue circle indicates the location of the AERONET Sede Boker site. Source: https://aeronet.gsfc.nasa.gov.

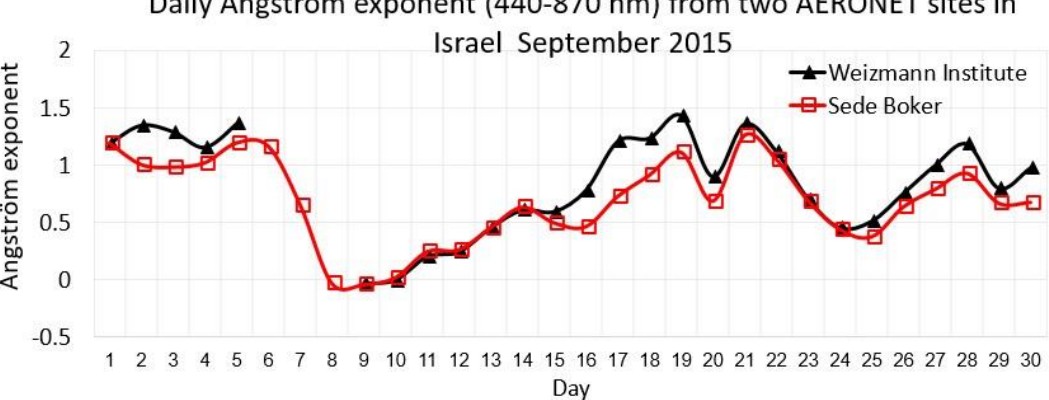

Figure 5. September 2015 daily average of AOD (top panel) and Angström exponent (bottom panel) from two AERONET sites in Israel (Sede Boker and Weizmann, see Fig.3). The Weizmann AERONET did not operate on 6-8 September due to power failure.

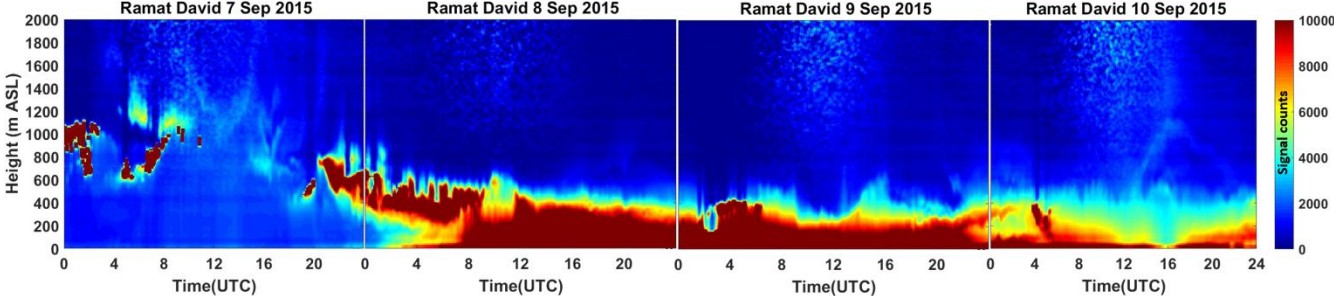

Figure 6. Ramat David ceilometer signal counts plots for 7-10 September 2015. Y-axis is the height up to 2000 m ASL, X-axis is the time in UTC, signal count scale range between 0-10,000.

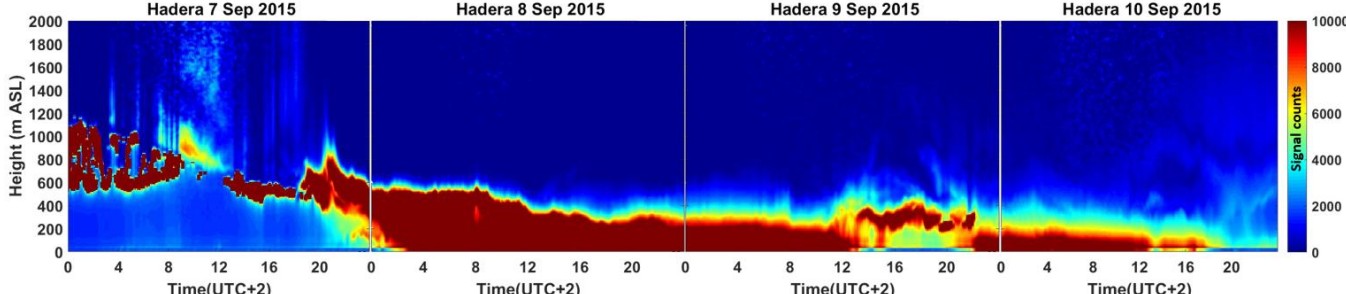

Figure 7. Hadera ceilometer signal counts plots for 7-10 September 2015. Y-axis is the height from site deployment to 2000 m ASL, X-axis is the time in LST (UTC+2), signal count scale range between 0-10,000.

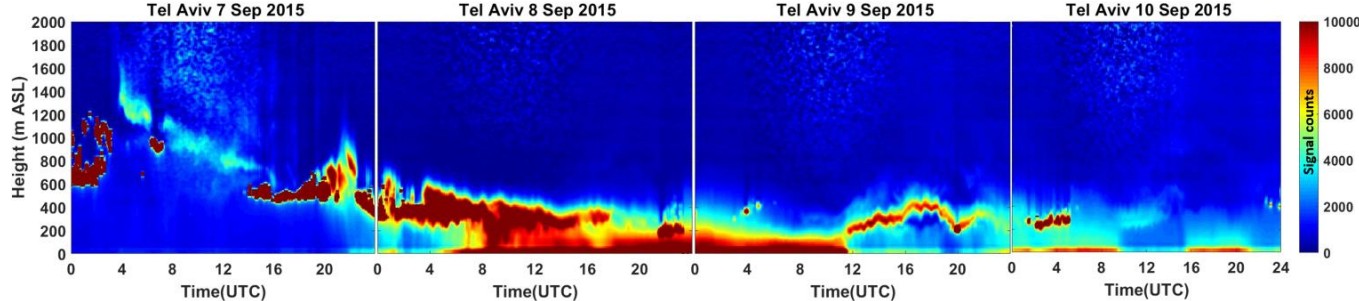

Figure 8. Tel Aviv ceilometer signal counts plots for 7-10 September 2015. Y-axis is the height from site deployment to 2000 m ASL, X-axis is the time in UTC, signal count scale range between 0-10,000.

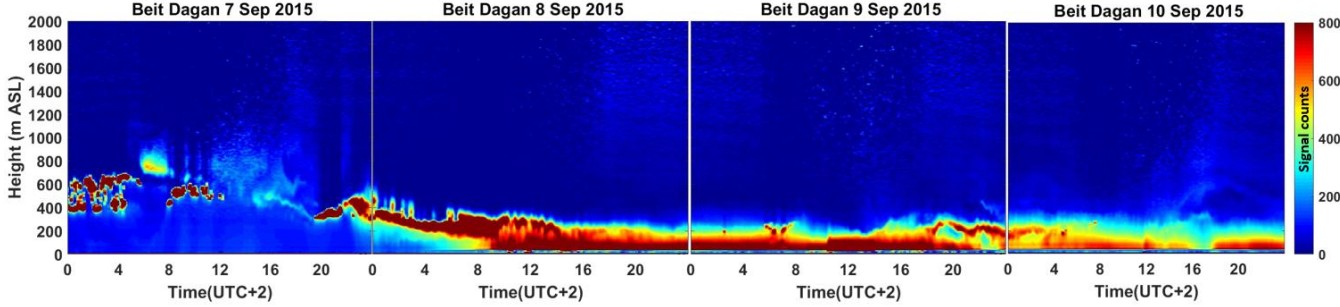

Figure 9. Beit Dagan ceilometer signal counts plots for 7-10 September 2015. Y-axis is the height from site deployment to 2000 m ASL, X-axis is in LST (UTC+2), signal count scale range between 0-800.

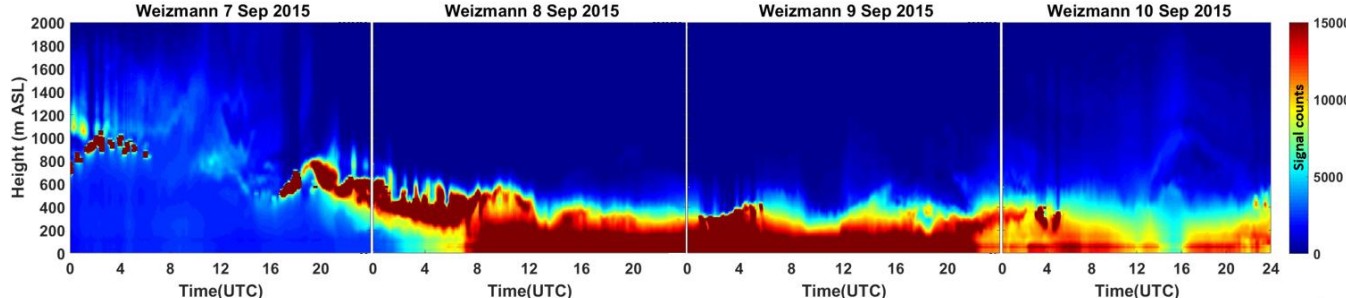

Figure 10. Weizmann ceilometer signal counts plots for 7-10 September 2015. Y-axis is the height from site deployment to 2000 m ASL, X-axis is in UTC, signal count scale range between 0-15,000.

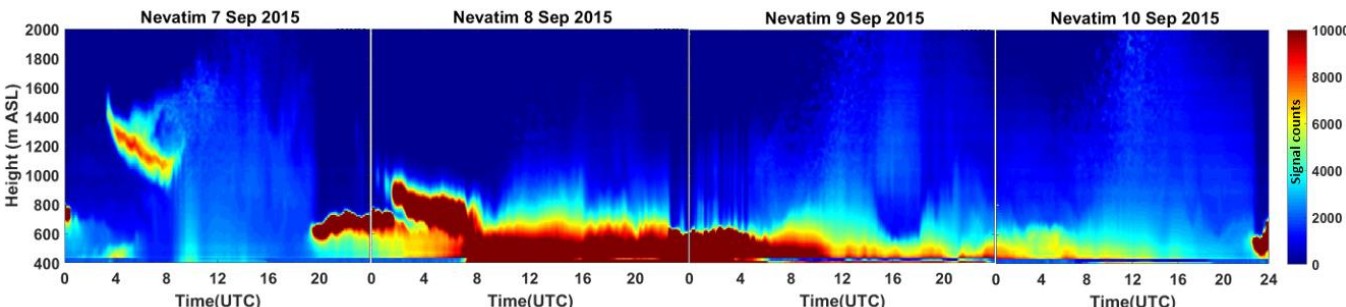

Figure 11. Nevatim ceilometer signal counts plots for 7-10 September 2015. Y-axis is the height from site deployment to 2000 m ASL, X-axis is the time in UTC, signal count scale range between 0-10,000.

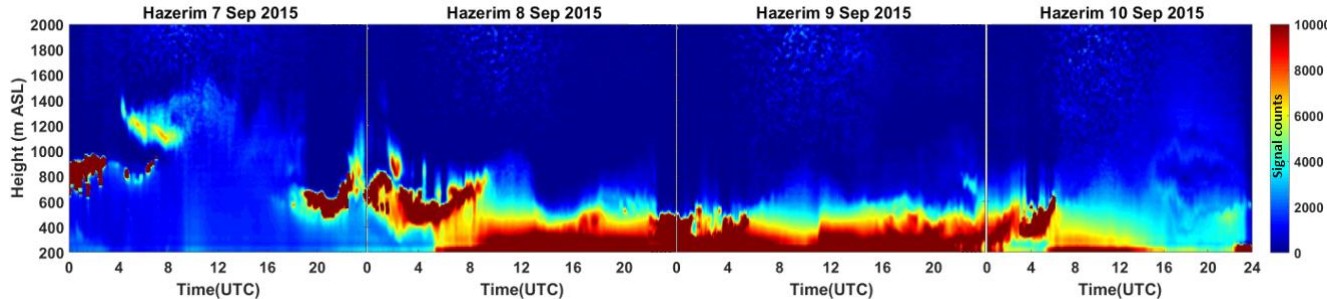

Figure 12. Hazerim ceilometer signal counts plots for 7-10 September 2015. Y-axis is the height from site deployment to 2000 m ASL, X-axis is the time in UTC, signal count scale range between 0-10,000.

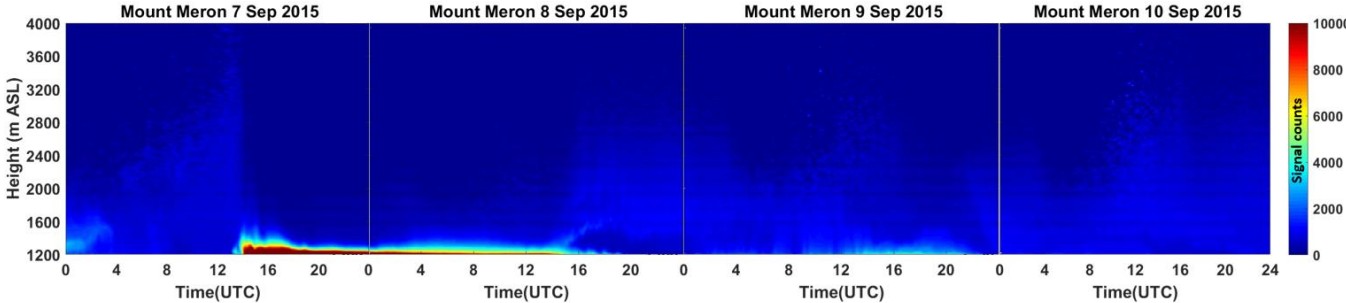

Figure 13. Mount Meron ceilometer signal counts plots for 7-10 September 2015. Y-axis is the height from site deployment to 4000 m ASL, X-axis is the time in UTC, signal count scale range between 0-10,000.

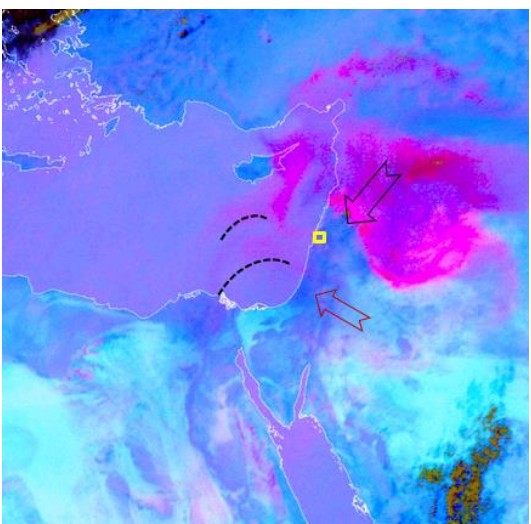

Figure 14. Picture from MSG-SEVIRI satellite of the dust RGB component (dust appears in pink colors) on 7 September 2015 12 UTC with indications of Mount Meron ceilometer site (yellow square, Lon 33.0˚, Lat 35.4˚) and the dust plumes progression from east to west (red arrow and dashed lines) and from the northeast to southwest (black arrow).

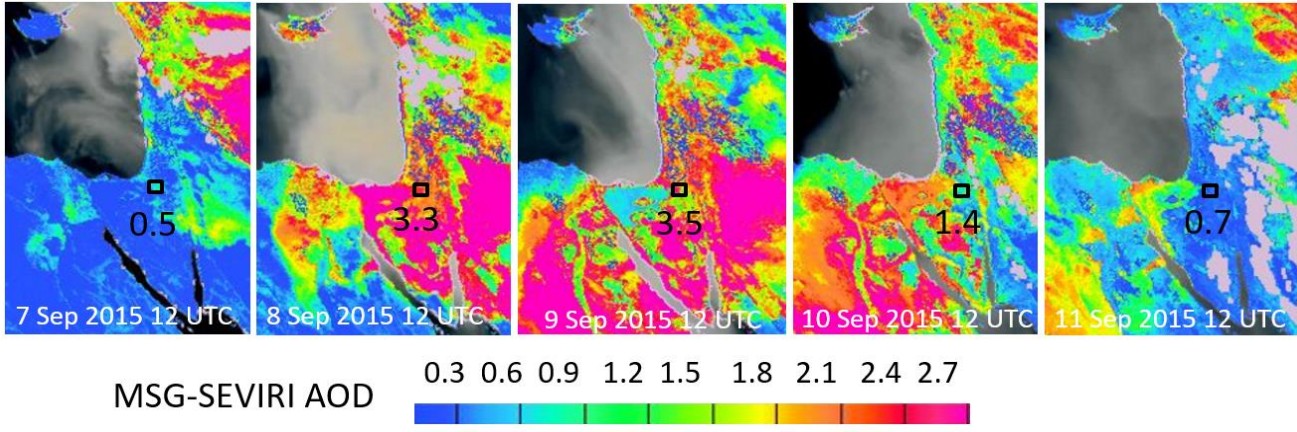

Figure 15. Aerosol Optical Depth (AOD) at 12 UTC 7-11 September 2015 analyzed by NASCube (Université de Lille) based on imagery from the MSG-SEVIRI satellite (by a combination of the SEVIRI channels IR8.7, IR10.8 and IR12.0). The map includes indication of the Sede Boker AERONET site (black square) and its AOD value at 12 UTC.

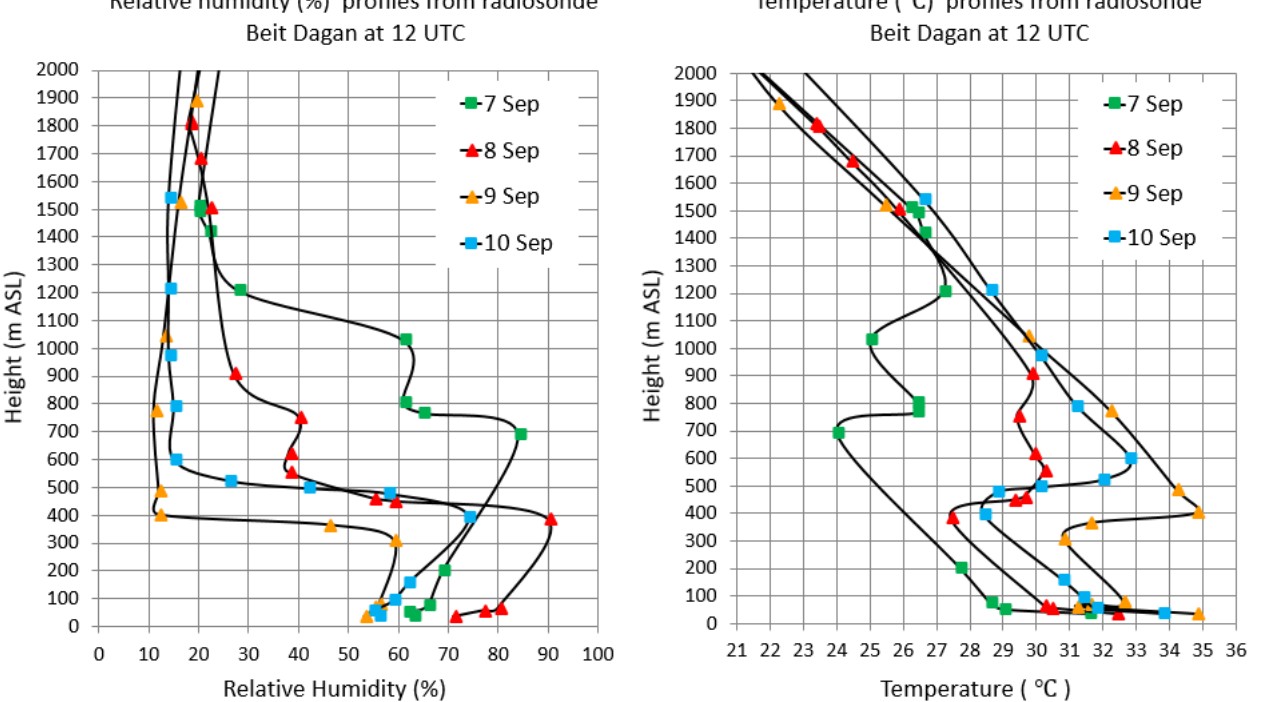

Figure 16. Radiosonde Beit Dagan profiles at 12 UTC between 7-10 September 2015 of relative humidity (left panel) and temperature (right panel).

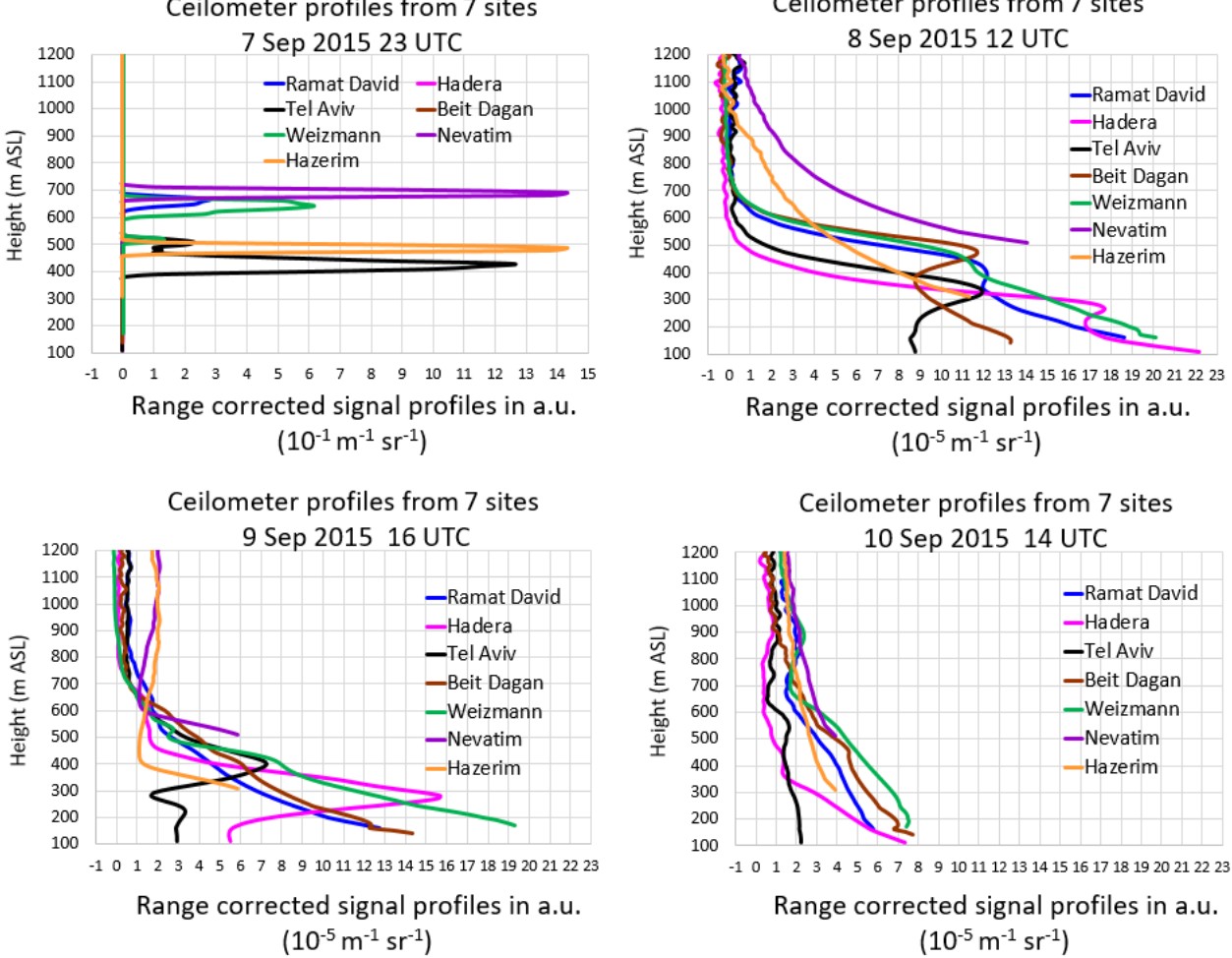

Figure 17. Ceilometer range corrected signal profiles (in arbitrary units) from 7 sites (Ramat David, Hadera, Tel Aviv, Beit Dagan, Weizmann, Nevatim and Hazerim, see locations in Fig. 3) on 7 Sep 2015 23 UTC (a), 8 September 2015 12 UTC (b), 9 September 2015 16 UTC (c) and 10 September 2015 14 UTC (d). Notice each profile begins relativily to the height of its' measuring site (ASL) including a deletion of data from the first 100 m AGL due to inaccuracies in the first range gates of the ceilometers (for details see Sect. 2.1). Fig (a) shows cloud detection therefore it is given in a different scale ($10^{-1}$ $m^{-1}$ $sr^{-1}$) and a different x-axis range.

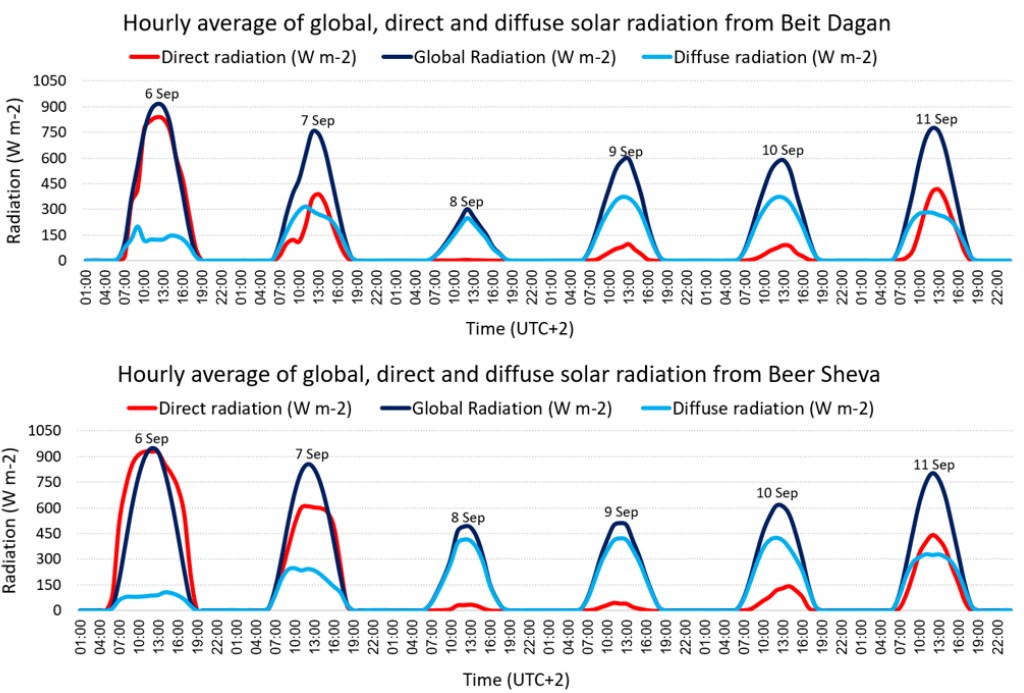

Figure 18. Hourly average of global, direct and diffuse solar radiation between 6-11 September 2015 from Beit Dagan and Beer Sheva.

## 3.2    8 September 2015

The main phase (the peak) of the dust storm occurred on 8 September. Images from MODIS Aqua (Fig. 4b) and MODIS Terra (Fig. 4f) taken between 08:00-11:15 UTC show the dust storm prevailed over Israel. Ceilometers' plots detect the descending motion of the dust plume reached ground level at ~ 08 UTC (Fig. 6-12). Simultaneously, Sede Boker AERONET AOD measurements increased up to ~4 along with a negative Angström exponent (not shown).

An hour later, at ~ 09 UTC, extreme maximum PM hourly values were measured in the elevated sites of Jerusalem Safra (10,280 µg m$^{-3}$ PM10) and Jerusalem Bar Ilan (3,063 µg m$^{-3}$ PM2.5). Whereby, in the coast and the lower northern regions, maximum PM values were measured only 14 hours later at ~23 UTC and were much lower (up to 3,459 µg m$^{-3}$ PM10 and 470 µg m$^{-3}$ PM2.5, see Tables 5-6). Fig. 19 illustrates the spatio-temporal variation of the PM10 extreme values, beginning at ~ 12 UTC in the elevated Jerusalem sites and ending at midnight in the shoreline.

At ~08 UTC ceilometer plots from Tel Aviv, Beit Dagan, Weizmann and Hadera (with a higher scale range of 0-15,000, not shown here) reveal an ununiformed dust layer, (beneath and above ~ 300 m

ASL) that eventually combined into one dense layer. This process may explain the spatial variation and time delay between the extreme PM measurements in the elevated vs. lower sites.

MSG-SEVIRI at 12 UTC estimated AOD to be 2.7 while Sede Boker AERONET measured a higher value of 3.3 (Fig. 15). Furthermore, MODIS images (Fig. 4a, 4b) show a dominant dust plume over Israel, whereas solar global radiation measurements (Fig.20a) present significant spatial variations as minimum values (down to 200 W m$^{-2}$) were measured mainly in northern Israel. Additionally, in spite the extreme PM10 values of 9,031 µg m$^{-3}$ measured in the elevated southern site (Negev Mizrahi 577 m ASL, Table 6), the maximum global radiation in southern Israel was still relatively high (~500 W m$^{-2}$). Generally, the radiative transfer analysis during heavy dust loads is complicated and relies on several parameters such as size, structure and composition of the aerosols (Bauer et al, 2011). Dense dust layers such as in this extreme dust storm definitely had an impact on the radiation budget hence changing weather patterns and air mass transport. The spatial variation of ground level measurements compared to the quit uniform picture revealed by the satellites may infer the complexity of the dust plume evolution.

Overall, 8 September shows the highest PM concentrations and the lowest solar radiation levels for this dust storm event. The solar radiation was composed mainly of diffuse radiation (Fig.18) emphasizing the immense atmospheric dust loads preventing direct insolation. Surprisingly, the low solar radiation was still capable to warm the ground and generate a late and weak sea breeze front (not shown). We assume the insufficient ground heating generated weak thermals that could not inflate a MLH. Therefore, we assume the low MLH (300 m ASL) revealed by radiosonde Beit Dagan profiles from 8 September (Fig.16) may indicate the dust plume base height. As a result, the ceilometers' plots were fully attenuated above ~300 m ASL.

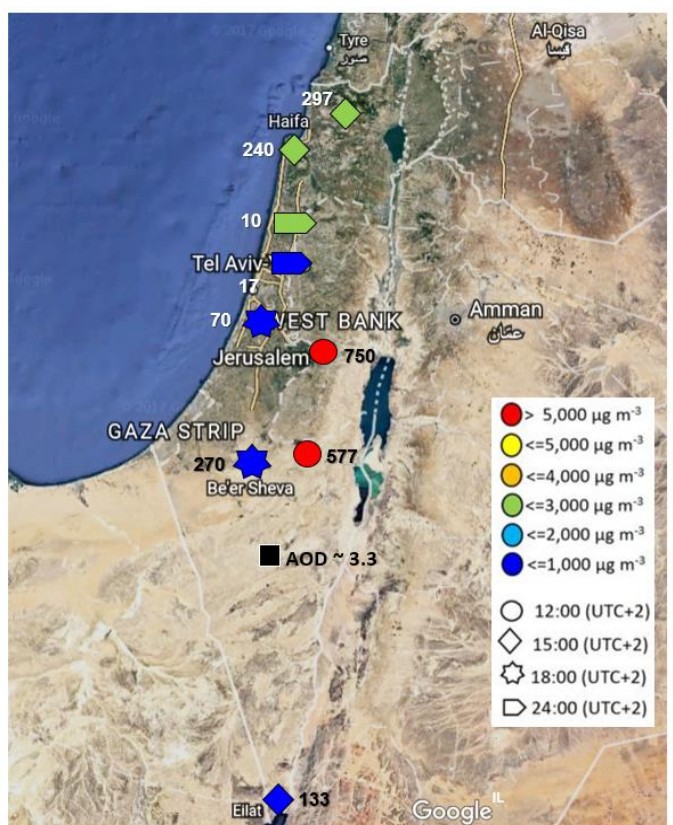

Figure 19. A map of PM10 maximum hourly concentrations from 9 sites measured on 8 September 2015 10 UTC (midday). The map includes indications of the time of measurement (symbol shape), concentration range (symbol color), height of measurement site (numbers on map) and AOD from AERONET Sede Boker site.

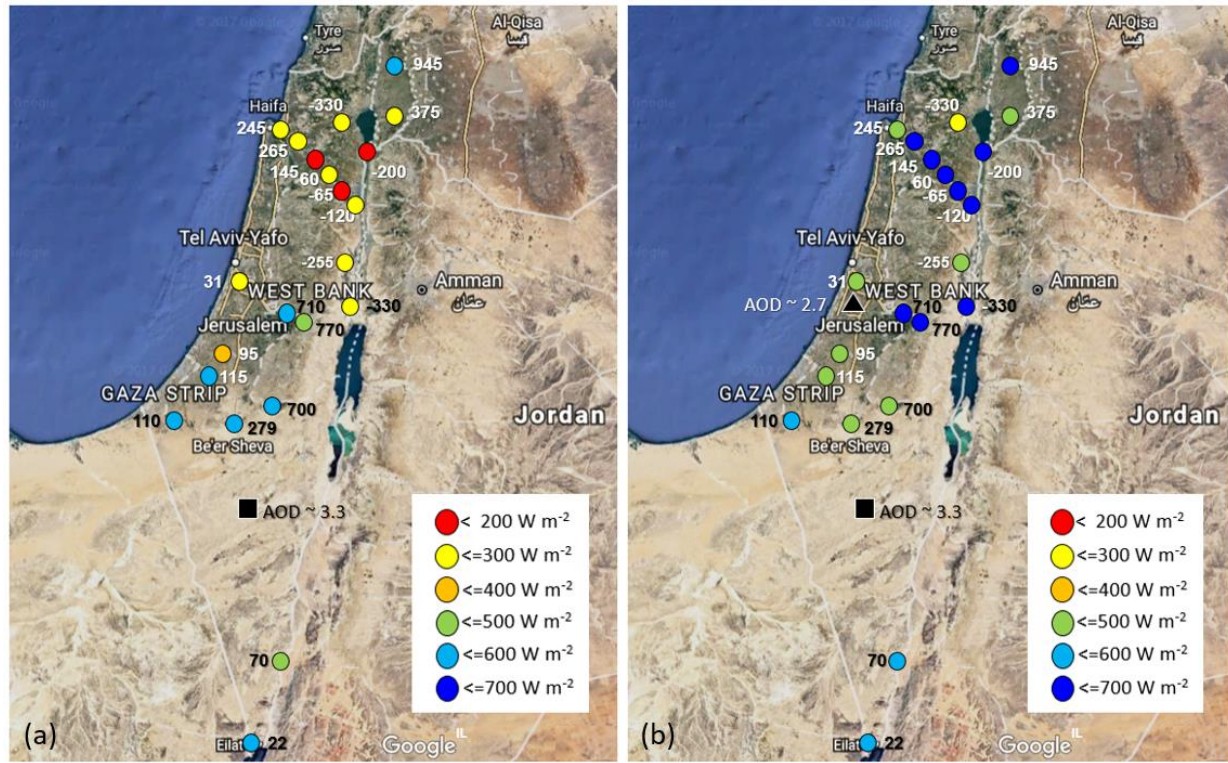

Figure 20. A map of maximum global solar radiation from 22 sites measured at 10 UTC (midday) on 8 September 2015 (a) and on 9 September 2015 (b). The map includes indications of radiation range (see legend), height of measurement site (numbers on the map) and AOD from AERONET Sede Boker site (black square) and Weizmann site (black triangle). On 8 September the AERONET in Weizmann site did not operate due to power failure.

## 3.3   9 September 2015

On 9 September, MODIS images (Fig.4c and 4g) taken between 07:05-12:00 UTC show the dust plume progression southward to Egypt (Fig. 4), indicated by Sede Boker AERONET AOD >3 along a negative Angström exponent (Fig. 5). At 12 UTC AOD from MSG-SEVIRI is~ 2.7 whereas Sede Boker AERONET AOD reached up to ~ 3.5 (Fig.15).  In contrary to the high AOD measurements, and the descend of the MLH down to ~ 350 m ASL (Fig.16), PM values did not increase but rather decrease below 900 µg m$^{-3}$ PM2.5 (Table 5) and 4050 µg m$^{-3}$ PM10 (Table 6). The drop in PM concentration gave rise to an increase of solar radiation up to 400 W m$^{-2}$ (Fig. 20 b). An increase in solar radiation enables significant ground heating to values measured prior to the initiation of the dust storm (not shown). Thus, allowing generation of thermals and the creation of a late the sea breeze cycle (Uzan et al., 2016). The entrance of the sea breeze front between 11- 12 UTC eventually produced a narrow dust layer ascent visible in mainly in the coastline -Tel Aviv and Hadera ceilometers (Fig.7-8). Interestingly, on 9

September, compared to the peak of the dust storm on the day before, we do not see a significant difference in solar radiation in southern Israel, which continued to be relatively high ~500 W m$^{-2}$ (Fig. 20b).

## 3.4  10 September 2015

On 10 September, MODIS pictures from 7:50 -11:05 UTC (Fig. 4d and 4h) show the dust plume over Israel transported southeast from Syria-Iraq to Sinai-Egypt. The CALIPSO single overpass Israel at 11:00-11:10 UTC revealed a dust layer up to 5 km ASL (Fig.21). This corresponds with the EARLINET lidar measurements in Limassol, Cyprus (Mamouri et al., 2016) detecting a dust plume between 1-3 km ASL. We assume the CALIOP lidar did not produce data beneath 2 km ASL due to total attenuation. Fortunately, the ceilometers complement the dust profile (beneath ~ 1 km ASL) showing a reduction both in signal counts (Fig. 6-12) and in range corrected signal profiles (Fig.17d) pointing out a reduction in atmospheric dust loads. AOD from MSG-SEVIRI and Sede Boker AEONET show a decrease down to ~1.5 and a low Angström exponent of ~0.5 indicating prevalence of mineral dust.

A profound reduction in PM values, down to a third of the values from the day before (Table 6), was evident mainly in southern Israel. Therefore, an increase in direct radiation was measured in southern site (Fig.18). The reduction of dust loads may also be denoted by the orange background color of the photograph taken on 8 September (Fig.1b) compared to the grey background visible on 10 September (Fig.1c). As the dust storm dissipated, cloud formation (indicated by brown spots and evaluated by ceilometer profiles -not shown) was visible from ~4 UTC by ceilometers Ramat David (Fig.6), Tel Aviv (Fig.8), Weizmann (Fig.10) and Hazerim (Fig.11). the clouds formation was not evident by MODIS imagery (Fig 4d, 4h).

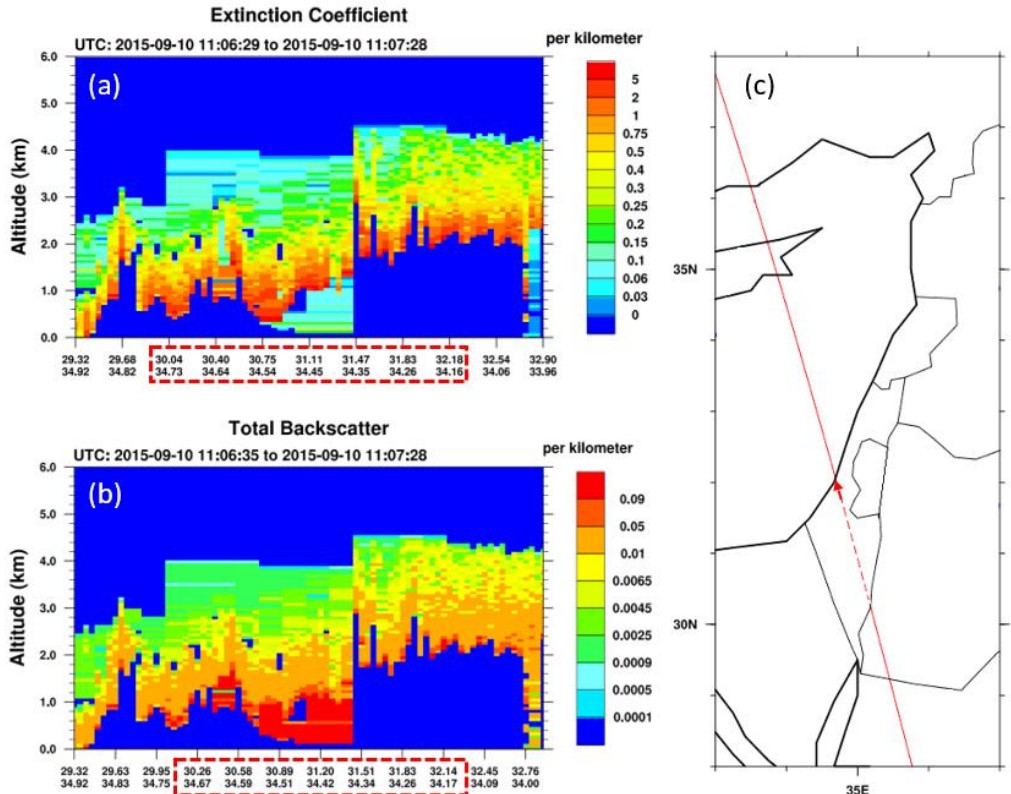


Figure 21. (c) A map of the EM centered on Israel with indication of CALIPSO satellite overpass
from south-east to north-west (red line) on 10 September 2015 11:00-11:10 UTC. CALIOP lidar
products (from 532 nm wavelength) of extinction coefficient (a) and total backscatter (b) are
given along the path over Israel (dashed red line). The ground elevation begins from -20 m ASL
in the southeastern point. A quarter of the way through ground level rises up to ~ 600 m ASL and
then gradually declines to sea level height as it reached shoreline in the northwestern point.

In the attempt to determine the "end" of the dust storm over Israel, we analyzed measurements
from all instruments (Sect. 2) seeking values that were measured prior to the dust storm penetration.
AERONET AOD values (Fig.5), solar radiation measurements (not shown) and satellite imagery from
MODIS and SEVIRI (not shown) indicate clearance of the dust storm on 17 September. On the other
hand, PM values and ceilometer profiles indicated the dust storm ended 4 days earlier on 13 September
(not shown). The difference between the measurements that include atmospheric layers aloft (satellite
imagery, solar radiation and AOD AERONET) compared to measurements limited to the lower
atmosphere (PM values and ceilometers) postulates a scheme of several dust layers or multiple sources
of the dust plumes, which may support similar conclusions from previous studies (Stavros et al, 2016;
Mamouri et al, 2016, Gasch et al, 2017).


# 4. Conclusions

A very severe dust storm struck the EM on September 2015. Previous investigations presented in-situ and remote sensing measurements, discussed the initiation of the dust storm in the Syrian-Iraqi border, aspects of its transport over the EM and the limitations of the models to forecast this unique event. The analysis concentrated mainly on the upper level of the atmosphere at specific time segments of the dust storm. The benefit of this study is the provision of continuous measurements of vertical dust profiles in the lower part of the troposphere from 8 ceilometer sites. The data presented here can be used as a tool to verify state of the art model simulations and provide a different point of view to the meteorological conditions governing the dust plume advection over the EM.

This study confirmed that the dust storm entered Israel on 7 September and showed the gradual downfall of the dust plume from ~1000m ASL on 7 September down to ~400 m ASL on 8 September. The detailed ceilometer profiles and auxiliary instruments enabled to separate the dust storm into separated dust layers (beneath and above 1 km). As the dust plume descended towards ground level on 8 September, PM concentration increased in the elevated stations (up to 10,280 $\mu$g m$^{-3}$ PM10) and radiation decreased down to ~200 W m$^{-2}$ mainly in the northern region.

On 9 September, in spite of the high AOD (above 3), the global radiation (mainly comprised of diffuse radiation) increased, thus enabling sufficient ground heating for the creation of a late sea breeze front (between 11-12 UTC). The sea breeze circulation generated a narrow dust layer detached from the ground in the coastal region (Tel Aviv, Hadera and Beit Dagan).

On 10 September, the dust plume motion continued southwest to Egypt, indicated by CALIPSO as dust layers up to 5 km. The end of the dust storm over Israel was indicated on 17 September by satellite imagery, solar radiation and AOD AERONET, while measurements limited to the lower atmosphere (PM values and ceilometers) indicated the dust storm ended on 13 September. The difference between the various instruments suggests a scheme of several dust layers or multiple sources of the dust plumes.

To conclude, Ceilometers have been found to be a crucial tool in the study of the September dust storm evolution over Israel. In general, ceilometers provide high resolution data base (temporal and spatial) that broaden the scope of the atmospheric measurements. Fortunately, as worldwide ceilometer deployment expands, ceilometers are realized as an essential tool in the analysis of meteorological phenomena and aerosol transport most valuable in the meso-scale.

# 5. Data availability



PM10 measurements- Israeli Environmental ministry air quality monthly reports:
http://www.svivaaqm.net.

Israeli Environmental ministry air quality monthly reports (in Hebrew):
http://www.sviva.gov.il/subjectsEnv/SvivaAir/AirQualityData/NationalAirMonitoing/Pages/AirMoritor
ingReports.aspx.

Weather reports- Israeli Meteorological Service September monthly report (in Hebrew):
http://www.ims.gov.il/IMS/CLIMATE/ClimateSummary/2015/hazesept+2015.htm

Radiosonde profiles –University of Wyoming: http://weather.uwyo.edu/upperair/sounding.html.

AERONET data- https://aeronet.gsfc.nasa.gov.

Meteosat Second Generation Spacecraft pictures:
http://nascube.univ-lille1.fr/cgi-bin/NAS3_v2.cgi.
https://www.eumetsat.int/website/home/Images/RealTimeImages/index.html

Ceilometer profiles- the data is owned by governmental offices. The data is not online and provided by
request.


# Author contribution

Leenes Uzan carried out the research and prepared the manuscript under the careful guidance of Smadar
Egert and Pinhas Alpert. The authors declare that they have no conflict of interest.





# **Acknowledgements**

For the provision of ceilometer data, we wish to thank the Israeli Meteorological Service (IMS), the Israeli Air Force (IDF), Association of towns for environmental protection (Sharon-Carmel) and Rafat Qubaj from the department of Earth and Planetary Science in the Weizmann institute of Science. Special thanks to Nir Stav (IMS) and Dr. Yoav Levy (IMS) for their fruitful advice, Anat Baharad (IMS) for computer assistance and Pavel Kunin (Tel Aviv university) for the CALIPSO images. We thank the principal investigators Prof. Arnon Karnieli and Prof. Yinon Rudich for their effort in establishing and maintaining Sede Boker and Weizmann AERONET sites. We wish to thank the institutes that provide open site data reduction: Université de Lille NASCube, Wyoming University Radiosonde and the Israeli ministry of Environmental protection for the PM data. Partial funding of this research was made by the Virtual Institute DESERVE (Dead Sea Research Venue).

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

Table 1. The publications on the September 2015 dust event

| Publications | Title | Main Tool | Main outcome |
|---|---|---|---|
| Pu, B. and Ginoux, P (2016) | The impact of the Pacific Decadal Oscillation on springtime dust activity in Syria | MODIS Terra MODIS Aqua DOD AOD, GFDL-AM3 model | Model underestimation in the EM due to inaccurate soil moisture |
| Parolari et al. (2016) | Climate, not conflict, explains extreme Middle East dust storm | WRF model | Unusual low level westerly wind spread to the EM, to reversely transport the previously eastward particles back to the EM. |
| Mamouri et al. (2016) | Extreme dust storm over the eastern Mediterranean in September 2015: satellite, lidar, and surface observations in the Cyprus region | MODIS, EARLINET profiles and PM10 | Dust plumes from Syria entered the EM in a double layer structure, pointing to multiple dust sources |
| Solomos et al. (2016) | Remote sensing and modeling analysis of the extreme dust storm hitting Middle East and Eastern Mediterranean in September 2015 | RAMS model EARLINET lidar, MSG and CALIPSO. | Low model ability to simulate the event, due to inaccuracies in model physical processes. |
| Jasim, F.H. (2016) | Investigation of the 6-9 September 2015 Dust Storm over Middle East | Satellite MSG-SEVIRI, Meteoinfo model | Two dust storms simultaneously, from northern Syria and Sinai desert created by two low pressure systems |
| Gasch et al. (2017) | An analysis of the September 2015 severe dust event in the Eastern Mediterranean | ICON-ART model | An unusual early active Red Sea Trough with meso-scale convective systems generating cold-pool outflows producing the dust storm. Model lacked development of a super critical flow to produce excessive wind speeds |







Table 2. Ceilometers locations

| Location | Site | Long/Lat | Distance from shoreline (km) | Height (m AGL) |
|---|---|---|---|---|
| Mount Meron | Northern | 33.0/35.4 | 31 | 1,150 |
| Ramat David | Northern | 32.7/35.2 | 24 | 50 |
| Hadera | Onshore | 32.5/34.9 | 3.5 | 10 |
| Tel Aviv | Onshore | 32.1/34.8 | 0.05 | 5 |
| Beit Dagan | Inland | 32.0/34.8 | 7.5 | 33 |
| Weizmann | Inland | 31.9/34.8 | 11.5 | 60 |
| Nevatim | Southern | 31.2/34.9 | 44 | 400 |
| Hazerim | Southern | 31.2/34.7 | 70 | 200 |

*Ceilometer Weizmann is a CL51





Table 3. Ceilometers configurations

| Location | Type | Time resolution(sec) | Height resolution (m) | *Height range (km) |
|---|---|---|---|---|
| Mount Meron | CL31 | 16 | 10 | 7.7 |
| Ramat David | CL31 | 16 | 10 | 7.7 |
| Hadera | CL31 | 16 | 10 | 7.7 |
| Tel Aviv | CL31 | 16 | 10 | 7.7 |
| Beit Dagan | CL31 | 15 | 10 | 7.7 |
| Weizmann | CL51 | 16 | 10 | 15.4 |
| Nevatim | CL31 | 16 | 10 | 7.7 |
| Hazerim | CL31 | 16 | 10 | 7.7 |

* Height range dependents on sky conditions and is limited as AOD increases.
* In all ceilometers but in Beit Dagan site, data acquisition was limited to 4.5 km based on the BLview firmware







Table 4. Ceilometer technical information

| Location | Type | Engine board | Receiver | Transmitter | Firmware |
|---|---|---|---|---|---|
| Beit Dagan | CL31 | CLE311 | CLR311 | CLT311 | 1.72 |
| Weizmann | CL51 | CLE321 | CLRE321 | CLT521 | 1.03 |



Table 5. Hourly maximum concentration of PM2.5, collected from 21 monitoring sites, between 7-10 September 2015. The values are ranked from low (dark green) to high (dark red) values.

| No. | Site | Height (m ASL) | Region | PM2.5 ($\mu$g m$^{-3}$) | | | |
|---|---|---|---|---|---|---|---|
| | | | | 7-Sep-15 | 8-Sep-15 | 9-Sep-15 | 10-Sep-15 |
| 1 | Kefar Masarik | 8 | North | 52 | 378 | 389 | 378 |
| 2 | Ahuza | 280 | North | 36 | 743 | 650 | 419 |
| 3 | Newe Shaanan | 240 | North | 43 | 400 | 466 | 525 |
| 4 | Nesher | 90 | North | 43 | 564 | 496 | 349 |
| 5 | Kiryat Biyalic | 25 | North | 53 | 424 | 703 | 447 |
| 6 | Kiryat Binyamin | 5 | North | 40 | 223 | 412 | 256 |
| 7 | Kiryat Tivon | 201 | North | 47 | 413 | 416 | 300 |
| 8 | Afula | 57 | North | 44 | 836 | 550 | 405 |
| 9 | Raanana | 54 | Coast | 38 | 173 | 291 | 229 |
| 10 | Antolonsky | 34 | Coast | 32 | 470 | 626 | 386 |
| 11 | Ashdod | 25 | Coast | 36 | 303 | 750 | 332 |
| 12 | Ironi D | 12 | Coast | 34 | 424 | 507 | 327 |
| 13 | Tel aviv Central Station | 29 | Coast | 41 | 716 | 803 | 451 |
| 14 | Ashkelon | 25 | Coast | 61 | 182 | 537 | 119 |
| 15 | Jerusalem Efrata | 749 | Mountain | 106 | 2285 | 434 | 403 |
| 16 | Jerusalem Bar Ilan | 770 | Mountain | 107 | 3063 | 641 | 518 |
| 17 | Gedera | 70 | South | 34 | 433 | 683 | 308 |
| 18 | Nir Israel | 30 | South | 25 | 363 | 638 | 228 |
| 19 | Kiryat Gvaram | 95 | South | 42 | 376 | 870 | 300 |
| 20 | Sede Yoav | 105 | South | 45 | 323 | 245 | 228 |
| 21 | Negev Mizrahi | 577 | South | 42 | 1748 | 526 | 317 |



Table 6. Hourly maximum concentration of PM10, collected from 31 monitoring sites, between 7-10
September 2015. The values are ranked from low (dark green) to high (dark red) values.

| No. | Site | Height (m ASL) | Region | PM10 ($\mu$g m$^{-3}$) | | | |
|---|---|---|---|---|---|---|---|
| | | | | 7-Sep-15 | 8-Sep-15 | 9-Sep-15 | 10-Sep-15 |
| 1 | Galil Maaravi | 297 | North | 114 | 3130 | 1987 | 1562 |
| 2 | Karmelia | 215 | North | 39 | 1120 | 1008 | 765 |
| 3 | Newe Shaanan | 240 | North | 104 | 3459 | 2471 | 1518 |
| 4 | Haifa Port | 0 | North | 78 | 1600 | 1965 | 1699 |
| 5 | Nesher | 90 | North | 117 | 3265 | 2746 | 1270 |
| 6 | Kiryat Haim | 0 | North | 82 | 1161 | 1625 | 1088 |
| 7 | Afula | 57 | North | 97 | 3239 | 2322 | 1961 |
| 8 | Um El Kotof | 0 | Coast | 99 | 2025 | 2028 | 1630 |
| 9 | Orot Rabin | 0 | Coast | 58 | 1152 | 1455 | 999 |
| 10 | Barta | 0 | Coast | 112 | 2540 | 2345 | 1612 |
| 11 | Qysaria | 19 | Coast | 54 | 1067 | 2116 | 1272 |
| 12 | Rehuvot | 70 | Coast | 88 | 2236 | 3045 | 1257 |
| 13 | Givataim | 0 | Coast | 112 | 1909 | 4014 | 1484 |
| 14 | Yad Avner | 77 | Coast | 61 | 1738 | 2902 | 1252 |
| 15 | Ameil | 20 | Coast | 96 | 2027 | 3472 | 1321 |
| 16 | Shikun Lamed | 17 | Coast | 51 | 1701 | 3244 | 1097 |
| 17 | Station | 29 | Coast | 87 | 1420 | 2176 | 998 |
| 18 | Ashkelon | 29 | Coast | 117 | 953 | 1692 | 551 |
| 19 | Ariel | 546 | Mountain | 128 | 2723 | 1481 | 1358 |
| 20 | Jerusalem Efrata | 770 | Mountain | 273 | 7820 | 1630 | 1437 |
| 21 | Jerusalem Bar Ilan | 749 | Mountain | 181 | 5588 | 1191 | 966 |
| 22 | Jerusalem Safra | 797 | Mountain | 491 | 10280 | 2389 | 1780 |
| 23 | Gush Ezion | 960 | Mountain | 310 | 6230 | 1679 | 1119 |
| 24 | Erez | 80 | South | 44 | 1000 | 1000 | 718 |
| 25 | Beit Shemesh | 350 | South | 115 | 2097 | 1943 | 1788 |
| 26 | Carmy Yosef | 260 | South | 85 | 1047 | 784 | 594 |
| 27 | Modiin | 267 | South | 185 | 2701 | 2245 | 1980 |
| 28 | Bat Hadar | 54 | South | 65 | 1342 | 2563 | 841 |
| 29 | Nir Galim | 0 | South | 94 | 1479 | 2292 | 1027 |
| 30 | Negev Mizrahi | 577 | South | 183 | 9031 | 2806 | 1730 |
| 31 | Eilat | 0 | South | 275 | 1867 | 1592 | 1684 |
