# Peer review of "New insights into the vertical structure of the September 2015 dust storm employing 8 ceilometers and auxiliary measurements over Israel"

_Atmospheric Chemistry and Physics, 2017_

## Referee Comment (RC1) · Anonymous Referee #1 · 21 Sep 2017

**New insights into the vertical structure of the September 2015 dust storm employing 8 ceilometers over Israel**

**by Leenes Uzan et al.**

The Eastern Mediterranean region including Israel experienced an extreme dust storm in September 2015. This gave rise to a number of papers investigating the development and characterization of this dust storm by means of numerical models and measurements. Uzan's paper belongs to the second category by considering the spatiotemporal distribution of the dust as measured by a network of 8 ceilometers in Israel. These measurements were supplemented by additional data sets from ground based in-situ measurements and passive remote sensing as well as satellite observation. As the ceilometers – the main focus of the paper – contribute the vertical distribution of dust in the lower part of the atmosphere with very high resolution they do provide valuable information. As a consequence the topic is worth to be published in ACP.

However, before publication can be considered the paper must undergo significant improvements: the discussion of the findings must be much more precise and convincing (and extended), and the organization of the text must be (partly) re-organized. Moreover, many expressions are unclear (the reader can imagine what is meant, but the text in its strict sense is not completely correct) and statements are inconsistent; it is impossible to mention all of them in the framework of a review (only a few examples are mentioned below). Several citations must be added. I strongly encourage all authors to carefully check the text in detail. According to the paper S.E. and P.A. were guiding the research, so it is expected that they can provide significant support.

For me as a reviewer it is really hard to help to improve the paper as many of the mandatory changes interact with changes somewhere else in the paper and many amendments may induce more changes and adaptations. As a consequence my comments below cannot be exhaustive. Moreover I will not comment on typos and linguistic deficiencies – this can be improved during copy editing (if the paper is accepted based on its scientific content).

Major concerns:

1. The description of the instruments and data sets on the one hand, and the presentation and discussion of scientific results on the other hand should be clearly separated. Presently e.g. the "radiosonde section" includes results (lines 211 ff.) that should be moved to Sect. 3. In the "results"-section the

discussion should be more elaborated and linked to the findings from the other data sets (provided in part when the MLH is discussed). Figs. 5a and 6 also belong to the results. A (short) section on the satellite data used in this study is missing and might be included. Note, that the titles of the subsections are inconsistent: "ceilometers" are instruments, but "PM10" is not.

Vice versa the role of the scaling factor (lines 282 ff) should be moved to the description of the instruments and data sets. In particular the ceilometer section must include more relevant information (see below).

2. The discussion of the scientific results must be improved and extended. In the present state the paper is sort of a collection of (useful) pieces of information, but their relationship and their interpretation is not sufficiently elaborated.

First of all the absolute values of the attenuated backscatter must be checked: some of the numbers are unrealistic and the corresponding figures are not clear. For example the labels of Fig. 7c (e.g. 0.000029 in units of $10^{-9}$ m$^{-1}$ sr$^{-1}$) are confusing. Moreover, there is a lot of information only shown in figures but not explained in words. The authors should be aware that a comparison of brown vs. brown color or brown vs. blue color (Figs. 10–16) is not suitable for a scientific publication. Please use quantitative numbers! Think about plotting coincident attenuated backscatter profiles of the 8 sites (similar to Fig. 7), this may help to see temporal delays in the arrival and decay of the plume.

Make sure that all figures are explained in detail (e.g. missing for Fig. 1 and Fig. 4) and their contribution to the scientific objectives is clear.

Below please find a few more detailed comments:

- Whole text: use a common format for dates: "8. September 2015" instead of "08.09.15" is certainly a better option.

- Whole text: Use "lidar" instead of "LIDAR".

- line 61: I recommend to add a short paragraph here introducing the scientific objectives and the benefit of the paper. Something like "One of the strongest events ever... It was investigated already by several studies... However, ... is missing. Therefore, in our investigation we focus on...". Then the short description of the event and the review of the existing publications can follow as is.

- line 62: Fig. 1 seems to be the justification that it is worthwhile to study the event. A detailed explanation of Fig. 1 is however missing.

- line 68: How does the visibility and the reference to AERONET fit together? Give a citation for AERONET.

- line 71 (figure caption): The AOD derived from MSG (mention the sensor!) is shown. Make clear that Fig. 6 shows the AOD from AERONET for comparison.

- lines 78 ff: Give citations for Meteoinfo, MODIS, EARLINET, ICON-ART.

- line 91: Avoid acronyms in cases when it is only used once or twice, e.g. RST and SBF.

- line 98: When referring to CALIPSO, mention that it was found that the top of the dust layer was at about 3–4 km (though the overpass was several hundreds of kilometers east of Israel). Even better: include a (short?) discussion of CALIPSO measurements (hopefully closer to Israel than in Gasch et al., 2017) into the results-section showing where the upper boundary of the dust layer has been. For this purpose, quicklooks from the CALIPSO-website could be sufficient. Then, the measurement range of the ceilometers can be highlighted (it is doubtful that the ceilometers can fully penetrate the dust layer at all times).

- line 115: Where was the lidar located?

- line 118: Were the model results compared to CALIPSO or EARLINET (both are lidars)?

- line 120: ”...but the ability to predict the details were partial..”: I don't understand this sentence.

- line 122: Lidars don't provide ”aerosol concentration”, but backscatter coefficients (or extinction coefficients, depended on the system).

- 124: Give citation to ”deep blue algorithm”.

- line 130: What is meant by ”scattering properties”? Is this relevant for this paper?

- line 139: ”So far, no attempt has been done to relate the models findings...”. This seems to be in contradiction to the previous review of publications. The benefit of this paper is the provision of vertical profiles of the dust in the

lower part of the troposphere and the continuous measurements at 8 sites. This helps to obtain a more complete picture (with high resolution) of the event over Israel: this is a valuable contribution.

"...spatial, temporal and vertical...": vertical can be omitted because it is "included" in spatial.

- line 140: "display" $\longrightarrow$ "discuss"

- line 144: "Section 3 presents the results": this is very general and does not help the reader. Specify the content of Sect. 3 a little bit!

- line 152: "... to the atmosphere above its measuring point". Must be "top of the atmosphere". Omit "above its measuring point": strictly it is a slanted column.

- line 159: "...to add the vertical aerosol distribution...". The distribution is not added, but information on the distribution. Check the whole paper for similar wordings that are not fully correct.

- line 165: To be more precise the first sentence and the corresponding citations should be modified: Start with a general statement on the potential of lidars to observe (dust) aerosols: here you can cite Mona et al. (already mentioned, but missing in the list of references), more papers from the EAR-LINET community such as Wiegner at al. (2011), Papayannis et al. (2008), Ansmann et al., (2003), all in J. Geophy. Res., or other papers. Then, you can state that with recent improvements of the ceilometers' hardware these eyesafe single-wavelength systems are getting more and more important in particular when implemented as network (move the citation of Wiegner et al., 2014, to this place). The Weitkamp-paper is also missing in the reference list – so I don't know why it is included here. Finally, Vaisala's CL31 can be introduced but note that the cited Münkel-paper from 2004 is on the CT25k-ceilometer, not on the CL31. So replace this citation by e.g. Münkel et al., (2011)[1]. Kotthaus et al. (2015) (cited later) is indeed on the CL31.

- Subsection "ceilometers": This paragraph needs major revisions. A few points: Make clear what you want to measure/determine: mixing layer heights, general structure of layers, heights of lower and upper boundaries, attenuated backscatter, particle backscatter coefficients or whatever. Depending on the desired output the requirements on the data evaluation are
* * *
[1] Münkel, C., Schäfer, K., and Emeis, S.: Adding confidence levels and error bars to mixing layer heights detected by ceilometer, Proc. SPIE, 8177, 817708-1817708-9, 2011.

different. Include the discussion of the "scale factor" here: it is given by the manufacturer and it is unknown to the user where it comes from and how accurate it is. What is the purpose of it? Is it just an "mean" conversion factor of counts (no unit) to attenuated backscatter in $m^{-1}$ $sr^{-1}$ ? When you mention the wavelength of 910 nm, you should also mention that the signals must be corrected for water vapor absorption whenever you want to quantitatively derive any aerosol related quantity (e.g. link to AOD); here the citation of Weigner and Gasteiger (already included in the reference list) should be added. As a consequence of Kotthaus et al. you should give the firmware version of your ceilometers. You should mention that the overlap correction is automatically performed by the proprietary software and that it is not disclosed to the user. If the overlap correction creates signal artefacts (at 50–100 m), different for each ceilometer, is this crucial for your scientific objectives (if yes, what are the consequences?)? Be clear with the temporal resolution (it can be selected by the user in the range from ... to ... s; in this paper 16 s was selected, is there a special reason for 15 s for some ceilometers?). 7.7 km applies to all ceilometers, not only that at Beit Dagan. Do you know the pulse energy of the CL31?

- lines 208 ff: Move all results to Sect. 3. "...disclosing the different meteorological conditions...": This is a too short statement: explain, what is shown. Explain the differences that can be seen from Fig. 4. What are the conclusions with respect to the overall topic of the paper?

- line 223: Give the manufacturer (and type) of the instruments.

- line 235: If the availability is given for the PM10-monitors it would be consistent to mention the availability of the other instruments as well (certainly a minor point).

- line 236: "(87% of the monitoring stations" can be omitted.

- lines 237/38: This is also a result and should be moved to the corresponding section. The same is true for Fig. 5a. Add a xy-grid to facilitate the interpretation of Fig. 5a.

- "...within a 1 min period": maybe it is better to remove this. From this statement the measurement cycle is not clear: 8 s measurements, 22 s break, 8 s measurements, 22 s break, 8 s measurements? This is more than 1 minute.

- line 262: As this is a result it should be moved to Sect. 3.

- Fig. 6: to be moved to Sect. 3. I don't understand the x-axis? Why is the length of the days different? Where are the night measurements of AERONET from (do you deploy a lunar photometer?)? Is there a specific reason for not choosing a line plot in Fig. 6a (as in Fig. 6b)?

- line 278: The results-section starts with the list of data sources used: AOD satellite pictures are mentioned but not explained in Sect. 2, whereas radiosondes are explained in Sect. 2 but not mentioned here.

- line 281 ff: Move lines 281–283 to Sect. 2. Check carefully the following numbers of the attenuated backscatter; some of them are unrealistic ($7 \cdot 10^{-1}$ $m^{-1}$ $sr^{-1}$). See also general remarks at the beginning. Add a xy-grid to Fig. 7.

- Figs. 8–16: What is the unit of the color code? It does not agree with the values given for attenuated backscatter. Are these numbers the "counts"?

- line 320: "the time corrected from local time to" can be removed. The labels of Fig. 8 should be the same as those of Figs. 10–16.

- lines 328 ff: Obviously attenuated backscatter is converted into particle extinction coefficient (with a relatively large uncertainty inherent in all CL31 measurements; see water vapor absorption mentioned above, unknown lidar ratio, unknown accuracy of the scaling factor) with an estimated lidar ratio. Then, the visibility is estimated according to Koschmider. Which altitude was selected for this conversion? The problem is, that if it is done for a large altitude (maybe 100 m or more) it is difficult to compare this visibility to independent ground based measurements. If it is done for the ground, then the overlap problem is critical. Nevertheless, an order of magnitude agreement should be possible, but please extend this paragraph by explaining all aspects of this comparison.

- lines 334 ff: "Ceilometers are not provided with an AOD...": I don't understand this paragraph. AOD and MLH are mentioned but it is unclear what the message is. The retrieval of the MLH from ceilometer data is completely different from a retrieval of the AOD (provided that can be determined at all). So, how is the validation ("verify ceilometer") of the ceilometer's performance achieved? Please rephrase and extend this part.

- line 381 ff: A short paragraph should be included here to prepare the reader: The event is divided into several phases (not necessarily split into single days,

one phase can be shorter, another can last for more than one day; development, main phase, decay can be an alternative) according to certain criteria and to highlight consistencies/inconsistencies of different data sets/models (by doing this "Next, we describe the decrease of aerosols aloft on mid-day" in line 299 can be omitted).

Then, each subsection such as "Entrance of dust into Israel – 7 Sep 2015" should include the full discussion and interpretation of all available data sets for the corresponding period, i.e., parts from Sect. 2 should be included here whenever applicable.

- line 401: What is meant by "decrease"? Concentration or altitude?

- line 405: "clearly shown in Fig. 13-16 between 08-16 UTC)": This is indeed hard to see. Can you explain this in a more quantitative way?

- line 408: Please clarify what "these model findings" are? The vertical distribution of dust (backscatter)?

- line 422: 2000 $\mu$g/m$^3$: is this in contradiction to 9800 $\mu$g/m$^3$ in line 416?

- line 437: "...limited radiative transmitted...": what does this mean?

- line 443: The figures don't support the "total clearance" statement.

- Fig. 17: This figure is misleading as the range of the color code is different from Fig. 14. This should be pointed out clearly. As long as the intercomparison with the AOD (AERONET) is qualitative only, it would be helpful to show the typical(?) background(?) values of the AOD and attenuated backscatter of the days before the event for comparison. From Fig. 6a it seems that after 11. September 2015 the AOD was still relatively large (by the way: can you give the annual mean AOD of the sites in Israel?). How "typical" are the ceilometer profiles between 11. and 14. September 2015?

- line 454: 250 m: is this the vertical extent or the altitude?

- line 486: The PM10 measurements are considered as in-situ measurements, not remote sensing.

- line 488: "...for the first time, such an event is vertically analyzed using an array of ceilometers...". On the one hand this is true, on the other hand it is slightly misleading as the vertical structure (by other means) has already been investigated. So it might be advisable to use a less strong statement in the next sentence (a note on the limited measurement range).

- lines 492, 497: "plum"!

- line 494: "mainly of mineral dust": where is this information coming from?

- line 507: "The complicated nature of the dust...". This is indeed a benefit of the ceilometer measurements: the temporal changes of the vertical distribution of aerosols can be monitored with very high resolution and thus help to better understand ground based measurements (even though there is the overlap-issue and the beam is likely fully attenuated somewhere in dense dust layers). It can be concluded that it makes no sense to "extrapolate" small scale vertical features in Israel from measurements in Cyprus. This should be emphasized as it is a strong argument for the publication of such a study.

- line 575: This citation must likely be updated.

---

## Referee Comment (RC2) · Anonymous Referee #2 · 17 Oct 2017

General

The paper is not in a good shape and needs significant improvement. My comments may help. Please do not feel criticized (as authors). I need to be critical to help you to improve the paper.

As the main goal ceilometer observations, continuously taken during a huge dust storm, are presented and discussed. However, the authors seem to have only minor experience how to handle ceilometer data properly and carefully. This is one of the most serious problems I have with this paper.

Ceilometers are made for cloud detection, and not for aerosol profiling. This is especially true for the Vaisala system, which operates at 910nm and is sensitively affected by water vapor absorption. In case of this severe and unique dust storm, the ceilometer obviously could be used to measure the lowest few hundred meters of the dust layers. But the main part, from 1 to about 4-5 km height remained undetectable.

It seems to me that the authors try to avoid to clearly state: The ceilometer is of rather limited use in dust plume tracking. We were not able to see the full dust layer. But such a statement is required! ..and will not disturb the main goal of the paper.

As reviewer I have to say: This is not acceptable and has to be improved! In cases with thick dust layer with optical depth >1 the transmitted (rather weak) radiation pulses of these ceilometers are immediately attenuated in the lowest part of the dust layers.

The other unacceptable issue is that the authors state that they present their ceilometer results in terms of 'attenuated backscatter coefficient'. However, this is simply wrong, very misleading, and will be confusing for many readers (especially aerosol lidar scientists).

By defintion: The attenuated backscatter coefficient is the Rayleigh-calibrated range-corrected backscatter signal! ... Such a calibration is usually impossible (for 910 nm backscattering), even at clear sky conditions.

Thus, the authors show color plots of the 'basic' range corrected ceilometer signals! All the plots have to be changed to meet this point. . ... as will be explained in detail below.

There are many other points that have to be clarified as explained below.

Major revisions are needed.

Details (P = page, L = line):

The abstract does not just summarize the paper properly. The abstract should be compact, i.e., as short as possible and just cover the contents of the paper: The main goal, the instruments and methods used, and main findings,. ... no outlook. . ., no

unnecessary (motivating) statements that can be given in the introduction. . .

P3, L78, reference for Meteoinfo or http. . ..

P3, L94, the model beeter explained. . ... compared to what?

P3, L94, ART, give reference

P4, L100, Please check the finally revised version of the Gasch paper. I asked these authors, the final version should be out soon.

P4, L115, L118, L122: lidar. . . not LIDAR

The introduction could be more straight forward, as follows: There was a huge dust storm in the Middle East, however, the dust forecast models failed. The question was then: Why? This question motivated Solomos et al. (2017) to run a cloud resolving model system. They found the most probable reasons. Please state their findings in the introduction! Afterwards, Gasch et al. (2017) used the new IKON/ART model system with rather high resolution (a global cloud-process-resolving model!!!) and also investigated this dust storm. . . and discussed the storm in even larger detail. . .. and concluded. . ... Please read the final version and present their final conclusions. This would be a nice introduction, very informative, so that the reader would learn a lot. And then you could provide the motivation for your own ceilometer study. . .. Because open points remained, and this historical dust storm must be documented for a variety of regions in the Middle East.

P5, L134, Is there no discussion in the litertaure on dust-radiation-dynamics relationship? I believe, the SAMUM group (Tellus 2011 special issue) investigated the relationship between dust (and smoke) and the radiation field and changes in the air flow (dynamics) as a result of the impact of dust and smoke on the radiative fluxes. Such dense dust layers as in September 2015 certainly had a huge impact on the radiation budget and significantly changed the weather pattern and thus air mass transport. This may explain why the routine dust forecast models failed because the forecasted

dust concentration was too low to produce a significant radiation effect on the weather pattern (and dynamics) and dust transport in the model.

P5, L138-L142: You must clearly say in the beginning how the ceilometer network can contribute: Ceilometers can detect the dust layer base and provide some information about the lower part of the dust layer and, very important, the downward transport towards the ground. This is a good and valuable contribution to atmospheric science. On the other hand, not more than that! But this is fine! Nevertheless, you need to provide the limits of such systems! Very clearly! At these high AODs, there is no chance to detect most of the dust and the dust layer top.

P6, L159-161: These statements are misleading. At these large dust AODs, satisfactory aerosol profiling with the Israeli ceilometers was impossible! Furthermore, I do not find Ansmann et al. and Mona et al. in the references.

P7, L173: Do you think that you would find the true mixing layer height (when applying the wavelet analysis) under such dense dust conditions? I am not sure! Usually you have the polluted mixing layer and the clear free troposphere on top. At these conditions, the wavelet technique works well. Now you have the opposite. And there was almost no solar heating of the ground (almost no convection), just a residual (less dust laden) layer below the dust layers. What you detect and interpret as mixing layer top is to my opinion just the other way around: the dust layer base height. One should state and discuss this point more clearly. At AODs of 2 and more there is no convection left to lead to well mixed conditions. The dust layer is warmer than the lowest, near-surface tropospheric layer and produces the temperature inversion observed with radiosonde.

P7, L178: . . .up to 7.7 km height AGL. Yes the ceilometer may measure up to 7.7 km height, but only for clear sky conditions with AOD of the order of 0.2, at least clearly below 1. One has to state that. In case of the dust storm, all delivered 'counts' above 1 km were more or less just background noise!

P7, L183: Now a critical issue: I checked the Kotthaus paper. According to this paper, and as it is well known, the ceilometer delivers range-corrected signals in arbitrary units. The measured counts are converted by using a conversion factor to obtain useful signal profiles, when the background is subtracted. We may denote these range corrected signals as level-0 data. Vaisala uses a 'conventional' conversion factor to transfer the background-corrected signals into lidar backscatter signals. But this is NOT the attenuated backscatter coefficient! To obtain the attenuated backscatter coefficient (something like the Rayleigh-calibrated range-corrected signal) you need an actual calibration (to obtain an actual conversion factor). This actual conversion factor however can only be derived under clear sky conditions (so that clear sky backscatter in the free troposphere becomes visible and an accompanying sunphotometer delivers the total OD (aerosol plus water vapor absorption) at around 910 nm, and the aerosol-related AOD at 910nm by using interpolation between the measured 870 and 1020nm AOD). At these favorable conditions, the range-corrected signal may be calibrated to deliver the attenuated (aerosol) backscatter profile, and by using the Klett method and adjustment of the lidar ratio (that has to take care of water vapor absorption in addition in cae of the Vaisala ceilometer) even the total particle backscatter coefficient. But all this was not possible under these dust storm conditions. In conclusion, you just present range-correct signals in arbitrary units. All the plots have to be changed accordingly. The paper is unacceptable if this important changes are not done.

P7, L188 ... The Beit Dagan Ceilometer measures up to 7.7 km. Yes, as mentioned, under clear sky conditions. At dust storm conditions with AODs from 2 to probably 5 and more, the ceilometer measures just noise from 1 to 7.7 km height. So this is an unacceptable statement. Please change.

P7, L198: Please remove the trivial statement about the radiosonde ascents. Please provide just information on radiosonde type and company, and meteorological parameters measured.

P8, Figure 4: I would suggest to use very contrasting colors, orange and blue, or such nice colors as in Figure 6, for the different sites. We need x-axis description (RH (%)

and Temperature (deg C)) and also y-axis text (Height (m.a.s.l.)). I am wondering. . ..: Do we need to show the Cyprus profiles? Further suggestion: To my opinion it would make sense to show the Israel radiosonde profiles up to 6-7 km height, to have a chance to identify the dust top height because above the dust layer the RH should increase again, at least should show changes, and there should be a temperature gradient change as well or even a temperature inversion at dust top height.

In the radiosonde plots there is always written: . . . luanching sites. . . please improve!

P9, Figure 5 caption: Please clearly state what the yellow curve shows. It took me some time, to see that it belongs to the southern region. The curves show the mean of all sites of a given region? Hard to see the ceilometer stations. Give them a yellow full circle and plot them last (after plotting everything else). Then one should clearly see the sites.

P10, L264. The Rehovot station did not work, the instrument was out of order? . . .. or did the station not allow useful data analysis because the AOD was too high? Please clarify and state what was the case. . .

P10, Figure 6: the time axis . . . the day scale (width) is changing from day to day. E.g., the 8 September is very narrow, the 7 September has a factor of 4 more space. . ., why?

P11, L282: The authors write: the ceilometer profiles are unitless and therefore divided by a scale factor of $10^{-9}$ 1 /(m sr) enabling quantitative analysis (Kotthaus et al., 2016).

This is simply wrong and unacceptable. Please improve! The basic ceilometer data are signals (let us say . . . in units. . . counts per second), and if they are then range-corrected. . . then you get the dimension 'counts per second times m**2'. So the values are not unitless, but usually given in arbitrary units. Next, by dividing these data by $10^{-9}$ . . . does not change anything.You still have just range-corrected signals. You

can only obtain a profile of the attenuated backscatter coefficient if you are able to cal-ibrate this range-corrected signal profile in the tropospheric region with pure Rayleigh backscattering or in the way as described above. So, you show range corrected sig-nals!!! And not attenuated backscatter!!! As mentioned already, you must change . . ..  to range corrected signals in all ceilometer plots!

P12, Figure 7, We need a clear statement, that the range-corrected signals shown in Figure 7 decrease rapidly and is close zero at about 700 to 750m in b,c,d because of the strong laser light attenuation in the dust layer! As long as such a statement is missing the reader may believe to see the full dust layer and the top is at 750-1000m height. To repeat: This is unacceptable.

Then, in Figure 8, the numbers for 'your'attenuated backscatter are suddenly up to 15000, compared to values of about 10ˆ(-14) in Fig 7 (b,c,d)? Then in Figs 10-12: up to 10000. And in Fig 13, suddenly only up to 800. . ., Fig. 14 up to 15000, and Fig15-16 up to 10000. So all this is rather strange. . .and only reasonable and understandable if we switched to range corrected signals (arbitrary units).

So, please change. . ... to range corrected signals.

P12. Figure 8,x-axis please show data always from 0-24 local time (or UTC). Again we need proper text for the x-axis and y-axis, as it is the case in Figure 7.

P12-15: All the Figures 8,10-16 have no x-axis and y-axis description. This is poor and unacceptable. And again, all these ceilometer color plots suggest that the dust layer was just a few hundred meter thick. This is dangerous! The reason is simply the almost total attenuation of the ceilometer radiation pulses by the rather dense dust layers. This must be made very clear.

P13, L324: again and again: you were able to track the dust layer base only, this must be clearly said.

P13, L330: A visibility of 200m (visual meteorological range is defined by an AOD

of 3, after Koschmieder for an AOD of 4) according to an AOD of 3 means that the particle extinction coefficient was 15 km-1 or 0.015 m-1 and the backscatter coefficient is then 0.0003 m-1 sr-1 if the dust lidar ratio is 50sr. All your 'numbers' are far far away from these value. This corroborates: It is impossible and dangerous (and thus not justified) to convert the range-corrected signals into optical properties just by taking 'some' conversion factor!

P13, L334-L337. I would remove this text on the ceilometer and the AOD upper limit. This is useless. The Vaisala ceilometers are not built for aerosol profiling. The wavelength is bad, the signals are corrupted by water vapor absorption.

P13, L347: Plots are given in different scales to highlight the dust features. This is ok, because range corrected signals are shown and the ceilometer performance changes from site to site (from ceilometer to ceilometer). So again, there is a clear need to work with range corrected signals.

P13, Fig. 9 shows the dust base height. To my opining it is misleading to denote the near-surface layer a 'mixed layer' at these conditions with no vertical exchange.

P13: The text on this page is poor and needs to be significantly improved.

P15, L3834: Please clearly state where the dust layer top was found by Solomos et al. (2017).

P17. . ... Figure 17 indicates that there was dust higher up. The AOD decreased towards 0.5-1 on 12-14 Sep. A perfect mixing layer could develop now up to 750 m, as seen on 13 and 14 Sep. Nice to see, that the aerosol dried in the PBL during the morning hours and thus the color of the range corrected signals changed from red to green and blue (for dry particles producing less backscatter later on).

P17, L435: At very high optical depth as on 9 Sep, I would assume that convective motions in the PBL as well as a sea breeze winds cannot develop. Are you sure that sea breeze developments were possible at these days with almost no sun and

differential sea/land heating? Please keep the discussion free of speculation.

P17, L444: You state: The ceilometer reveal total clearance on 10 Sep! But the Weizmann Institute AERONET shows AODs of 2 and more on 10 Sep! What is wrong, what is true? Please clarify?

P18, L454: ..as a dust layer of 250 m thickness (fig 11-13, 15-16) penetrationed Israel at a height of 1000-1500m. . .. How do you know the depth of the dust layer? The ceilometer fails to see higher up. . .. So, how do you know? I would leave out to mention any dust layer depth.

P18, L461: The AOD was >1.0 all the time on 9 and 10 Sep. . .until 12 Sep. What do you thus mean with dissipation of dust?

P19: The conclusions have to be rewritten compeletly after improving all the text before along the lines this review and the other review.

P19, L488: for the first time such an event is vertically analyzed. . .. . .. this is misleading because Mamouri et al. already used lidar to characterize the dust storm. You probably wanted to say, for the first time . . .. with a ceilometer network. However, you should mention that there were already lidar studies with Cyprus lidar and CALIOP lidar, and now you come with a ceilometer network study. . .. . .. Then this would be more clear, and of course this is a new aspect.

P19, L490: Again, you have to mention the limits of a ceilometer. It was too weak to see the layers higher up. No chance to see the main part of the dust layers and dust layer top.

P19, L494: As a result. . .. . .. of what?

P19, L498-499: When were the AE values high again? They were continuously <0.5 even on 14 Sep (Weizmann AERONET).

P19, L502-504: This is speculation, at least to my opinion. Be more save with your

statements.

P19, L506-511: Again, dangerous statements. I would remove. Otherwise, you need to check the CALIPSO overflight over Israel to corroborate your speculative suggestions. However, the modeling papers of Solomos et al and Gasch et al. (partly based on model plus CALIOP results) do not leave room for statements like . . . who knows to what height the dust plumes reached over Israel. To my opinion, in the Middle East dust layer top was up to 4-5 km height everywhere.

P21, L554: No authors..

P23, L621: TOASJ. . .?

Some more comments:

I asked colleagues from Israel to help me with. . . the following. . ., I am not familiar with Hebrew language. . . for example: http://www.svivaaqm.net/.

Technically, you have at http://www.svivaaqm.net/ 'reports from numerous sites'. . . The dust storm: at September 7, around 22:00 values above 100 are already recorded for two sites (cities): Kiriat Ata and Nesher (next to Haifa).

Further questions and recommendations, they had:

The authors must justify what new information they get using ceilometer more clearly than in Figure 18, what new insights they get about the extreme dust event? And number/summarize all "new insights" about the event that they discover.

What sources of errors do they have when using the ceilometers? They should critically state the limitations, disadvantages and advantages. Including comparison between different ceilometers that authors used for the analyses. Without this comprehensive critical discussion on authors findings, the outcome of the paper is doubtful.

Haifa region (Northern part of Israel) must be included for more comprehensive analyses. The authors stated that: "Unfortunately, there are no PM10 measurements in

[Figure]

northern Israel".(Line 233) During the dust event there were seven sites that measure PM10 that were available. Figure 5 is not correct therefore, it covers more area, than depicted in Figure 5, including the Haifa Bay area.

Figure 18: The major claim- the dust penetrates from the East. But- combining PM10 from the Haifa Bay area, there is a "jump" towards values of 2500-3000 micrograms/m3 at 8 of September, similarly to East, which means it has two entrances/sources. Do the authors see the "North region" dust entrance using ceilometer data?

Why did the authors not include PM2.5 sites data.

"The AERONET (Fig. 6) and ceilometer plots (Fig. 8, 11-16) reveal that the first dust plume penetrated Israel at approximately 04:00 UTC". What day? Sept. 7 or Sept 8?

Page 3, Fig.1: The shown AOD range... is that the range of trustworthy values? Because the dust AODs were much higher than 2.7. So one could enlarge the color scale... How large is the uncertainty in the MSG data, source for the data (http....) should be mentioned.

---

## Author Comment (AC1) · 30 Nov 2017

We wish to thank both referees for their comprehensive and elaborated comments. Despite some mistakes they have revealed in the manuscript, the referees took the time and effort to explain the source of the inaccuracies and offer good advice and practical solutions.

Overall, we have received extensive comments on all sections and figures by both referees. Therefore, we've decided to combine the referees' comments and write our point to point response along the manuscript sections: Abstract, Introduction, Instruments, Results, Conclusions and discussion, References.

[Figure]

We hope this structure is acceptable and easy to follow.

Sincerely, Leenes Uzan, Dr. Smadar Egert and Prof. Pinhas Alpert

Please also note the supplement to this comment:
https://www.atmos-chem-phys-discuss.net/acp-2017-634/acp-2017-634-AC1-supplement.pdf

---

## Author Comment (AC2) · 4 Dec 2017

**Author's Response:**

We wish to thank both referees for their comprehensive and elaborated comments. Despite some mistakes they have revealed in the manuscript, the referees took the time and effort to explain the source of the inaccuracies and offer good advice and practical solutions.

Overall, we have received extensive comments on all sections and figures by both referees. Therefore, we've decided to combine the referees' comments and write our point to point response along the manuscript sections:

- Abstract
- Introduction
- Instruments
- Results
- Conclusions and discussion
- References

We hope this structure is acceptable and easy to follow.

Sincerely,
Leenes Uzan, Dr. Smadar Egert and Prof. Pinhas Alpert

**Abstract: lines 39-51**

**Referees Comments:**

Referee #2: *The abstract does not just summarize the paper properly. The abstract should be compact, i.e., as short as possible and just cover the contents of the paper: The main goal, the instruments and methods used, and main findings…. no outlook..., no unnecessary (motivating) statements that can be given in the introduction....*

**Author's response:** Comments accepted.

**Author's changes in manuscript:** The abstract was rewritten as recommended.

**Introduction: lines 56-145**

**Referees Comments:**

Referee #1: *line 61: I recommend to add a short paragraph here introducing the scientific objectives and the benefit of the paper. Something like" One of the strongest events ever... It was investigated already by several studies... However, ... is missing. Therefore, in our investigation we focus on...". Then the short description of the event and the review of the existing publications can follow as is.*

Referee #2: *The introduction could be more straight forward, as follows: There was a huge dust storm in the Middle East, however, the dust forecast models failed. The question was then: Why? This question motivated Solomos et al. (2017) to run a cloud resolving model system. They found the most probable reasons. Please state their findings in the introduction! Afterwards, Gasch et al. (2017) used the new IKON/ART model system with rather high resolution (a global cloud-process-resolving model!!!!) and also investigated this dust storm... and discussed the storm in even larger detail.... and concluded.... Please read the final version and present their final conclusions. This would be a nice introduction, very informative, so that the reader would learn a lot. And then you could provide the motivation for your own ceilometer study.... Because open points remained, and this historical dust storm must be documented for a variety of regions in the Middle East.*

**Author's response:** Comments accepted.

**Author's changes in manuscript:** Keeping in thought this paper is supplementary and an important insight of the extreme September 2015 dust storm, we significantly enlarged the overview of previous studies describing the dust storm source, the analysis of observations and posteriori model runs. On our behalf, we added information describing the weather conditions and analyzed the dust event evolution over Israel in the lower atmosphere (from ground level to about 1 km). The data we present can support verification of state of the art model simulations (actually already been done by Gasch et al., (2016) which cited our presentation from the EGU conference).

**Referees Comments:**
Referee #1: *line 68: How does the visibility and the reference to AERONET fit together? Give a citation for AERONET.*
**Author's response:** Comments accepted.
**Author's changes in manuscript:** The referee is correct. AOD from AERONET could not be a reference to visibility. The AERONET reference was erased. We added a citation for the AERONET data.

**Referees Comments:**
Referee #1*: line 62: Fig. 1 seems to be the justification that it is worthwhile to study the event. A detailed explanation of Fig. 1 is however missing.*
*line 71 (figure caption): The AOD derived from MSG (mention the sensor!) is shown. Make clear that Fig. 6 shows the AOD from AERONET for comparison.*
Referee #2: *Page 3, Fig.1: The shown AOD range... is that the range of trustworthy values? Because the dust AODs were much higher than 2.7. So one could enlarge the color scale... How large is the uncertainty in the MSG data, source for the data (http....) should be mentioned.*
**Author's response:** Comments accepted.
**Author's changes in manuscript:** Fig 1. was moved to the " Results and discussion" section where it was discussed. The caption was updated with the MSG SEVIRI sensor and source of the MSG data. The uncertainty of AOD measurements from MSG SEVIRI can rise up to 15%. The Sede-Boker AERONET AOD values at 12 UTC were added to the figure above the spot indicating the Sede-Boker site.

**Referees Comments:**
Referee #1: *lines 78 ff: Give citations for Meteoinfo, MODIS, EARLINET, ICON-ART.*
Referee #2:  *P3, L78, reference for Meteoinfo or http.*
**Author's response:** Comments accepted.
**Author's changes in manuscript:** Citations were added.

**Referees Comments:**
Referee #1: *line 91: Avoid acronyms in cases when it is only used once or twice, e.g. RST and SBF.*
**Author's response:** Comment accepted.
**Author's changes in manuscript:** Acronyms used up to twice were deleted.

**Referees Comments:**

Referee #1: *Use a common format for dates: "8. September 2015" instead of "08.09.15" is certainly a better option.*

**Author's response:** Comment accepted.

**Author's changes in manuscript:** The format of all dates was changed to the suggested format.

**Referees Comments:**

Referee #1: *line 98: When referring to CALIPSO, mention that it was found that the top of the dust layer was at about 3–4 km (though the overpass was several hundreds of kilometers east of Israel). Even better: include a (short?) discussion of CALIPSO measurements (hopefully closer to Israel than in Gasch et al., 2017) into the results-section showing where the upper boundary of the dust layer has been. For this purpose, quick looks from the CALIPSO website could be sufficient. Then, the measurement range of the ceilometers can be highlighted (it is doubtful that the ceilometers can fully penetrate the dust layer at all times).*

**Author's response:** Comments accepted.

**Author's changes in manuscript:** We added two images (total backscatter and extinction coefficient) from the only passage of CALIPSO over Israel, on the 10 September 2015. The images reveal a dust layer between 2-4 km ASL (see Fig.X below). A discussion on the CALIPSO measurements was added to the "Results and discussion" section.

**Fig. X**

[Figure]

(a)The CALIPSO passage over Israel (red line) on the 10 September 2015.

(b+c) Imagery based on CALIOP total backscatter and extinction coefficient above Israel ( lon\lat indicated by the black rectangle)

**Referees Comments (overview of Gasch et al.):**

Referee #2:

*P3, L94, the model better explained..... compared to what?*

*P3, L94, ART, give reference.*

*P4, L100, please check the finally revised version of the Gasch paper. I asked these authors, the final version should be out soon.".*

**Author's response:** Comments accepted.

**Author's changes in manuscript:** We elaborated the overview of the Gasch et al. (2016) paper, explaining the model set up, ART reference and the concluded process of the dust storm generation. The final version of the Gasch et al (2016) was approved on the 15 November 2017 and will be cited accordingly.

**Referees Comments (overview of Stavros et al.):**

Referee #1:

*line 115: Where was the lidar located?*

*line 118: Were the model results compared to CALIPSO or EARLINET (both are lidars)?*

*line 120: "...but the ability to predict the details were partial..": I don't understand this sentence. Is this relevant for this paper?*

*line 122: Lidars don't provide" aerosol concentration", but backscatter coefficients (or extinction coefficients, depended on the system).*

**Author's response:** Comments accepted.

**Author's changes in manuscript:** We have rearranged the paragraph written on the Stavros et al., research. We referred to the lidar located (Limassol Cyprus), comparison of the EARLINET lidar to the model results and the estimation of the aerosol concentration based on the lidar ability to reproduce the strength of the dust event.

**Referees Comments:**

Referee #1: *Use "lidar" instead of "LIDAR".*

Referee #2: *P4, L115, L118, L122: lidar... not LIDAR*

**Author's response:** Comments accepted.

**Author's changes in manuscript:** "LIDAR" was changed to "lidar" throughout the text.

**Referees Comments:**

Referee #1:

*line 130: What is meant by" scattering properties"?*

**Author's response:** Comment accepted.

**Author's changes in manuscript:** " Scattering properties " was corrected to "optical properties".

**Referees Comments:**
Referee #1:
124: *Give citation to " deep blue algorithm".*
**Author's response:** Comment accepted.
**Author's changes in manuscript:** A citation for deep blue algorithm was added (Hsu et al., 2013).

**Referees Comments:**
Referee #2*: P5, L134, Is there no discussion in the literature on dust-radiation-dynamics relationship? I believe, the SAMUM group (Tellus 2011 special issue) investigated the relationship between dust (and smoke) and the radiation field and changes in the air flow (dynamics) as a result of the impact of dust and smoke on the radiative fluxes. Such dense dust layers as in September 2015 certainly had a huge impact on the radiation budget and significantly changed the weather pattern and thus air mass transport. This may explain why the routine dust forecast models failed because the forecasted dust concentration was too low to produce a significant radiation effect on the weather pattern (and dynamics) and dust transport in the model.*
**Author's response:** Comments accepted.
**Author's changes in manuscript:** The cited reference of Pu et al.(2016) did not discuss the effect of dust particles on the thermal radiation. Following the referees' remark, an elaborated discussion based on ground level radiation was added to "Results" section.

**Referees Comments:**
Referee #1*: line 139:  "So far, no attempt has been done to relate the models findings...".* *This seems to be in contradiction to the previous review of publications. The benefit of this paper is the provision of vertical profiles of the dust in the lower part of the troposphere and the continuous measurements at 8 sites. This helps to obtain a more complete picture (with high resolution) of the event over Israel: this is a valuable contribution. "...spatial, temporal and vertical...": vertical can be omitted because it is "included" in spatial.*
*line 140: "display" - "discuss".*
Referee #2: *P5, L138-L142: You must clearly say in the beginning how the ceilometer network can contribute: Ceilometers can detect the dust layer base and provide some information about the lower part of the dust layer and, very important, the downward transport towards the ground. This is a good and valuable contribution to atmospheric science. On the other hand, not more than that! But this is fine! Nevertheless, you need to provide the limits of such systems! Very clearly! At these high AODs, there is no chance to detect most of the dust and the dust layer top.*
**Author's response:** Comments accepted.
**Author's changes in manuscript:**  We rephrased the paragraph referring to the ceilometers' contribution (lines 138-142) and listed operation the limitation of the ceilometers in general and under high AOD conditions.

Referee #1*:*
*line 140: "display" - "discuss".*
**Author's response:** Comment accepted.
**Author's changes in manuscript:**  The paragraph was rewritten. Among other changes, "display" was changed to "discuss".

**Manuscript lines 148-273 (Sect.2 Instruments):**

**Referees Comments:**
Referee #1: *A (short) section on the satellite data used in this study is missing and might be included.*
**Author's response:** Comments accepted.
**Author's changes in manuscript:** A section of satellite data regarding imagery from MSG, MODIS terra, MODIS aqua and CALIPSO satellites was added to the "Instruments" section.

**Referees Comments:**
Referee #2: *The description of the instruments and data sets on the one hand, and the presentation and discussion of scientific results on the other hand should be clearly separated. Presently e.g. the" radiosonde section" includes results (lines 211 ff.) that should be moved to Sect. 3*
**Author's response:** Comments accepted.
**Author's changes in manuscript:** Presentation and discussion of scientific results were moved to the "Results" section.

**Referees Comments:**
Referee #1: *line 152: "... to the atmosphere above its measuring point". Must be" top of the atmosphere". Omit" above its measuring point": strictly it is a slanted column.*
*line 159: "...to add the vertical aerosol distribution...". The distribution is not added, but information on the distribution. Check the whole paper for similar wordings that are not fully correct.*
**Author's response:** Comments accepted.
**Author's changes in manuscript:** we rephrased these sentences as recommended.

**Referees Comments:**

Referee #1: *The role of the scaling factor (lines 282 ff) should be moved to the description of the instruments and data sets. In particular, the ceilometer section must include more relevant information.*

*line 165: To be more precise the first sentence and the corresponding citations should be modified: Start with a general statement on the potential of lidars to observe (dust) aerosols: here you can cite Mona et al. (already mentioned, but missing in the list of references), more papers from the EARLINET community such as Wiegner at al. (2011), Papayannis et al. (2008), Ansmann et al., (2003), all in J. Geophy. Res., or other papers. Then, you can state that with recent improvements of the ceilometers' hardware these eyesafe single-wavelength systems are getting more and more important in particular when implemented as network (move the citation of Wiegner et al., 2014, to this place). The Weitkamp-paper is also missing in the reference list – so I don't know why it is included here. Finally, Vaisala's CL31 can be introduced but note that the cited Mu̇nkel-paper from 2004 is on the CT25k-ceilometer, not on the CL31. So replace this citation by e.g. Mu̇nkel et al., (2011)1. Kotthaus et al. (2015) (cited later) is indeed on the CL31. Subsection"* ceilometers*": This paragraph needs major revisions. A few points: Make clear what you want to measure/determine: mixing layer heights, general structure of layers, heights of lower and upper boundaries, attenuated backscatter, particle backscatter coefficients or whatever. Depending on the desired output the requirements on the data evaluation are different.*

Referee #2: *It seems to me that the authors try to avoid to clearly state: The ceilometer is of rather limited use in dust plume tracking. We were not able to see the full dust layer. But such a statement is required! ..and will not disturb the main goal of the paper. As reviewer I have to say: This is not acceptable and has to be improved! In cases with thick dust layer with optical depth >1 the transmitted (rather weak) radiation pulses of these ceilometers are immediately attenuated in the lowest part of the dust layers".*

**Author's response:** Comments accepted.

**Author's changes in manuscript:** The ceilometer paragraph was changed. We described the ceilometers limitations including the instrument specific characteristics which affect the quality and availability of the attenuated backscatter profiles, the calibration methods, the rage detection limitations by the overlap function and determination of the sensitivity of the attenuated backscatter signal to relative humidity. Nevertheless, we portrayed the contribution of the ceilometers in the analysis of dust plume development in the lower part of the troposphere (up to 1.5 km AGL).

**Instruments - Ceilometers:  lines 164-188**

**Referees Comments:**

Referee #1: *Include the discussion of the" scale factor" here: it is given by the manufacturer and it is unknown to the user where it comes from and how accurate it is. What is the purpose of it? Is it just a" mean" conversion factor of counts (no unit) to attenuated backscatter in m−1 sr−1?*

**Author's response:** The ceilometer attenuated backscatter profiles are automatically corrected by:

*An internal calibration to convert signal counts to attenuated backscatter multiplying by $10^{-9}$.  The internal algorithm (resulting in $10^{-9}$), unknown to the user, is suitable for al ceilometer types (C. Muenkel, private communication, September 2017).

*A cosmetic shift of the backscatter signal to better visualize the clouds base.

*An obstruction correction when the ceilometers' window is blocked (by an obstacle).

*An overlap correction to the height where the receiver field of view reaches complete overlap with the emitted laser beam.

**Author's changes in manuscript:** Data referring to the automatic correction of the ceilometer output profiles was added to the "Instruments" section.

**Referees Comments:**

Referee #1: *When you mention the wavelength of 910 nm, you should also mention that the signals must be corrected for water vapor absorption whenever you want to quantitatively derive any aerosol related quantity (e.g. link to AOD); here the citation of Weigner and Gasteiger (already included in the reference list) should be added.*

**Author's response:** Following Wiegner et al., (2014, 2015), the ceilometer wavelength range (given as 905 ±3 nm) is influenced by water vapor absorption. However, in the case of aerosol layer detection, the water vapor distribution has a small effect on the signal change (indicating the MLH or an elevated mixed layer) because the aerosol backscatter itself remains unchanged. Consequently, except for a case of a dry layer in a humid mixed layer height, the water vapor is unlikely to lead misinterpretation of the aerosol stratification.

**Author's changes in manuscript:** The aforementioned explanation was added to the "Ceilometers" subsection. In our study we did not attempt to derive aerosol quantity as our ceilometers were not calibrated. We have rephrased all sentences that could mislead the reader in this manner.

**Referees Comments:**

Referee #1: *As a consequence of Kotthaus et al. you should give the firmware version of your ceilometers.*

**Author's response:** Unfortunately, 6 out of the 8 ceilometers belong to a governmental office which does not allow to publicize their firmware data. Yet, we have been confirmed by

the exclusive supplier of the ceilometers in Israel that the combination of firmware and hardware had been done according to Kotthaus recommendations.

Moreover, the title of Table 2 refers to all ceilometers as CL31 type, while ceilometer Weizmann (Rehovot) is a CL51. Therefore, we added Table X as follows:

Table X. Ceilometer configuration

| Location | Type | Time resolution(sec) | Height resolution (m) | *Height range (km) |
|---|---|---|---|---|
| Mount Meron | CL31 | 16 | 10 | 7.7 |
| Ramat David | CL31 | 16 | 10 | 7.7 |
| Hadera | CL31 | 16 | 10 | 7.7 |
| Tel Aviv | CL31 | 16 | 10 | 7.7 |
| Beit Dagan | CL31 | 15 | 10 | 7.7 |
| Rehovot (Weizmann) | CL51 | 16 | 10 | 15.4 |
| Nevatim | CL31 | 16 | 10 | 7.7 |
| Hazerim | CL31 | 16 | 10 | 7.7 |

*Height range is dependent on sky conditions and is limited as AOD increases.

* In all ceilometers but Beit Dagan site, data acquisition was up to 4.5 km downloaded from BLview firmware.

The firmware and hardware of the two remaining ceilometers:

Weizmann: engine board ( CLE321 ),  receiver( CLRE321 ), transmitter type( CLT521 ) firm ware version(1.03).

Beit Dagan: engine board ( CLE311 ),  receiver( CLR311 ), transmitter type( CLT311 ) firm ware version(1.72).

**Author's changes in manuscript:** We added the information given above.

**Referees Comments:**

Referee #1: *You should mention that the overlap correction is automatically performed by the proprietary software and that it is not disclosed to the user. If the overlap correction creates signal artefacts (at 50–100 m), different for each ceilometer, is this crucial for your scientific objectives (if yes, what are the consequences?).*

**Author's response:** In our study we focused on the downward motion of the dust plume and its effect on the creation of the mixed layer height which did not reach below 200 m AGL. Therefore, artefacts up to 100 m AGL were not crucial.

**Author's changes in manuscript:** An explanation of the automatic overlap correction was added to the text.

**Referees Comments:**

Referee #1: *Be clear with the temporal resolution (it can be selected by the user in the range from ... to ... s; in this paper 16 s was selected, is there a special reason for 15 s for some ceilometers?). 7.7 km applies to all ceilometers, not only that at Beit Dagan. Do you know the pulse energy of the CL31?*

Referee #2: *P7, L178: ...up to 7.7 km height AGL. Yes the ceilometer may measure up to 7.7 km height, but only for clear sky conditions with AOD of the order of 0.2, at least clearly below 1. One has to state that. In case of the dust storm, all delivered 'counts' above 1 km were more or less just background noise!.*

*P7, L188 ... The Beit Dagan Ceilometer measures up to 7.7 km. Yes, as mentioned, under clear sky conditions. At dust storm conditions with AODs from 2 to probably 5 and more, the ceilometer measures just noise from 1 to 7.7 km height. So this is an unacceptable statement. Please change.*

**Author's response:** All ceilometers were capable to measure up to 7.7 km with a range gate of 10 m with the limitations of sky condition and a decrease in SNR with height. The data acquisition was limited to 4.5 km by BLview default (except for ceilometer Beit Dagan). Only ceilometer Beit Dagan has temporal resolution of 15 s set by the manufacturer. The pulse energy of the CL31 1.2 μWs ± 20%.

**Author's changes in manuscript:** We rephrased the sentence and pointed out the limitations of the ceilometers to actually measure up to the maximum height range declared by the manufacturer.

**Referees Comments:**

Referee #2: *In the radiosonde plots there is always written: ... luanching sites... please improve!*

**Author's response:** Comment accepted.

**Author's changes in manuscript:** "luanching" was replaced by "launching".

**Referees Comments:**

Referee #2: *P7, L173: Do you think that you would find the true mixing layer height (when applying the wavelet analysis) under such dense dust conditions? I am not sure! Usually you have the polluted mixing layer and the clear free troposphere on top. At these conditions, the wavelet technique works well. Now you have the opposite. And there was almost no solar heating of the ground (almost no convection), just a residual (less dust laden) layer below the dust layers. What you detect and interpret as mixing layer top is to my opinion just the other way around: the dust layer base height. One should state and discuss this point more clearly. At AODs of 2 and more there is no convection left to lead to well mixed conditions. The dust layer is warmer than the lowest, near-surface tropospheric layer and produces the temperature inversion observed with radiosonde.*

**Author's response:** We recognize the referees' logic on the creation of thermals under high AOD conditions. Ground level measurements of global radiation, ground temperature (actually named " grass temperature" as it measures 10 cm AGL) reveal the process of ground heating on the 9 September was possible as the maximum global radiation measured at 12 (UTC+2) reached 500-700 (W m-2) in 19 out of 23 sites measured (Fig X1). In comparison, on the 8 September, maximum global radiation reached 200-300 (W m-2) in 10 out of 23 sites (Fig. X2).

**Fig. X1** Maximum global solar radiation 9 September 2015 at 12 (UTC+2) measured across Israel in 23 sites. Indication of height of each measuring site (ASL) is given upon the map.

[Figure]

**Fig. X2** Maximum global solar radiation 8 September 2015 at 12 (UTC+2) measured across Israel in 23 sites. Indication of height of each measuring site (ASL) is given upon the map

[Figure]

Moreover, 33 sites of ground temperature measurements were separated to 5 regions- north, south, center, coast and mountains:

**Fig. X3** Ground temperature (10 cm AGL) from northern Israel

Hourly average of ground Temperature (10 cm AGL) northern Israel ( 8 sites)  6-11 September 2015

Legend:
- Avney Eitan (375 m ASL)
- Ayerlet Hashar (10 m ASL)
- Dafna (135 m ASL)
- Elon (300 m ASL)
- Kefar Blum (75 m ASL)
- Newe Yaar (115 m ASL)
- Afula (60 m ASL)
- Galed (180 m ASL)

**Fig. X4** Ground temperature (10 cm AGL) from mountain sites in Israel

Hourly average of ground temperature (10 cm AGL) in two mountain sites in Israel 6-11 September 2015

Legend:
- North- Pichman Mount Meron (945 m ASL)
- South- Shani (700 m ASL)

**Fig. X5** Ground temperature (10 cm AGL) from valley sites in Israel

Hourly average of ground temperature (10 cm AGL) from the Israel valley (7 sites) 6-11 September 2015

Legend:
- Beit Zayda (-200 m ASL)
- Eden Farm (-120 m ASL)
- Gilgel (-255 m ASL)
- Hazeva (-135 m ASL)
- Mesada (-200 m ASL)
- Yavniel (0 m ASL)
- Zemah (-200 m ASL)

**Fig. X6** Ground temperature (10 cm AGL) along the coast of Israel

Hourly average of ground temperature (10 cm AGL) along the coast of Israel (7 sites) 6-11 September 2015

Legend:
- Beit Dagan (31 m ASL)
- Besor Farm (110 m ASL)
- Dorot (155 m ASL)
- Ein Hahoresh (15 m ASL)
- Ein Carmel (25 m ASL)
- Kevuztat Yavne (50 m ASL)
- Shavey Ziyon (8 m ASL)

**Fig. X7** Ground temperature (10 cm AGL) from southern Israel

[Figure]

The Fig. X3-X7 reveal the ground temperature on 9 September reached the values measured on 6 September. Furthermore, comparing the global radiation (FIG. X1-X2) and ground temperature figures (Fig. X3-X7), we find in southern Israel (the region of the Sede-Boker AERONET site) the maximum global radiation on the 8 September was the highest among all sites (400- 600 W m-2) and barely changed on 9 September.  Hence, we do believe the creation of thermals was possible on the 9 September and the thermal creation on the 8 September was rather weak. Thus, the WCT method (seeking the derivative which is the peak value) on the 8 September was probably a result of subsidence of the dust plume as suggested by the referee.

**Author's changes in manuscript:** If the referees agree with our findings, we will add the aforementioned data to the text and edit the plots according to the authors' submission instructions.

**Referees Comments:**

Referee #2:  *P7, L183: Now a critical issue: I checked the Kotthaus paper. According to this paper and as it is well known, the ceilometer delivers range-corrected signals in arbitrary units. The measured counts are converted by using a conversion factor to obtain useful signal profiles, when the background is subtracted. We may denote these range corrected signals as level-0 data. Vaisala uses a 'conventional' conversion factor to transfer the background-corrected signals into lidar backscatter signals. But this is NOT the attenuated backscatter coefficient! To obtain the attenuated backscatter coefficient (something like the Rayleigh-calibrated range-corrected signal) you need an actual calibration (to obtain an actual conversion factor). This actual conversion factor however can only be derived under clear sky conditions (so that clear sky backscatter in the free troposphere becomes visible and an accompanying sunphotometer delivers the total OD (aerosol plus water vapor absorption) at around 910 nm, and the aerosol related AOD at 910nm by using interpolation between the measured 870 and 1020nm AOD). At these favorable conditions, the range-corrected signal may be calibrated to deliver the attenuated (aerosol) backscatter profile, and by using the Klett method and adjustment of the lidar ratio (that has to take care of water vapor absorption in addition in case of the Vaisala ceilometer) even the total particle backscatter coefficient. But all this was not possible under these dust storm conditions. In*

*conclusion, you just present range-correct signals in arbitrary units. All the plots have to be changed accordingly. The paper is unacceptable if this important changes are not done.*
**Author's response:** Comments accepted.
**Author's changes in manuscript:** The plots present signal counts. The term "backscatter coefficient" was deleted from the text. We apologize for the mistake. Thank you for the detailed explanation.

**Instruments- Radiosonde:  lines 197-213**

**Referees Comments:**
Referee #1: *"...disclosing the different meteorological conditions...": This is a too short statement: explain, what is shown. Explain the differences that can be seen from Fig. 4. What are the conclusions with respect to the overall topic of the paper?*
Referee #2:  *P7, L198: Please remove the trivial statement about the radiosonde ascents. Please provide just information on radiosonde type and company, and meteorological parameters measured.*
*lines 208 ff: Move all results to Sect. 3.*
**Author's response:** Comments accepted.
**Author's changes in manuscript:** The discussion and figures of the radiosondes profiles were moved to the "Results" section. An elaborated discussion on the radiosondes profiles was added. The radiosonde is a Vaisala type RS41-SG producing profiles of humidity, temperature, pressure, and wind measurement.

**Instruments- Particulate matter measurements: lines 221-247:**

**Referees Comments:**
Referee #1: *Note, that the titles of the subsections are inconsistent: "ceilometers" are instruments, but "PM10" is not.*
**Author's response:** Comment accepted.
**Author's changes in manuscript:** The title of Sect. 2.3 was changed from "PM10" to " Particulate matter monitoring". In the text, PM10 was mentioned as the type of measurement and not as the name of the instrument itself.

**Referees Comments:**
Referee #1:
*line 223: Give the manufacturer (and type) of the instruments.*
**Author's response:** The particulate matter measurements were taken from the air monitoring network directed by the Israel ministry for environmental protection. Unfortunately, the ministry does not publicize in their internet site the type of instrument

they use. Nevertheless, all monitoring stations were granted ISO17025 meaning the calibration method of measurement is approved by European standards. Based on private communication with the ministry, the manufacturer of the majority of particulate matter instruments is Thermo Fisher Scientific. Two main types are used:  FH 62 C14 (beta attenuation method) and 1405 TEOM (tapered element oscillating microbalance method). **Author's changes in manuscript:** The instrument type and manufacturer were added to the text.

**Referees Comments:**

Referee #1:

*line 235: If the availability is given for the PM10-monitors it would be consistent to mention the availability of the other instruments as well (certainly a minor point).*

*line 236:" (87% of the monitoring stations" can be omitted.*

**Author's response:** The PM10 continuous measurements are 5 min average based on instantaneous measurements of 1 min. Data is automatically defined "invalid" if among the average based used (e.g.,30 min, 1 hr) more than 25% of the data is missing. Therefore, the availability of the PM10 concentrations we have presented was possible to evaluate.  In the other instruments (AERONET, ceilometers), there isn't a defined method to objectively evaluate the availability of the data, thus it was not stated. Overall, the ceilometers data (not referring to the quality of the data along the profile) was 100% but for Weizmann (Rehovot) site with 6 hours missing on the 6 September (shown in the figure) due to a local power failure. The AERONET sites availability is dependent on the daylight hours and the degree of AOD (at high AODs the instruments shuts down). Fig. 6 was an attempt to present the different availability of the AODs as the extent of available data changed from day to day. **Author's changes in manuscript:**  We added comments referring to the availability of the data from each instrument but avoided presenting absolute values.

**Referees Comments:**

Referee #1:

*lines 237/38: This is also a result and should be moved to the corresponding section. The same is true for Fig. 5a.Add a xy-grid to facilitate the interpretation of Fig. 5a. Figs. 5a … also belong to the results.*

Referee #2: *P9, Figure 5 caption: Please clearly state what the yellow curve shows. It took me some time, to see that it belongs to the southern region. The curves show the mean of all sites of a given region? Hard to see the ceilometer stations. Give them a yellow full circle and plot them last (after plotting everything else). Then one should clearly see the sites.*

**Author's response:** Following the comments we decided to present the data from specific monitoring stations (29) rather than an average of a set stations representing a region. A set of draft figures is given (Fig. X8-X11). If the referees approve, these plots will be organized according to editing instructions.

**Fig. X8** PM10 measurements in the mountain region of Israel between 7-10 September 2015. The list of monitoring stations and their measuring height is given in the legend. The area of monitoring is denoted in the map on the left by a dashed black circle.

[Figure]

**Fig. X9** PM10 measurements in the northern region of Israel between 7-10 September 2015. The list of monitoring stations and their measuring height is given in the legend. The area of monitoring is denoted in the map on the left by a dashed black circle.

[Figure]

**Fig. X10** PM10 measurements in the coastal region of Israel between 7-10 September 2015. The list of monitoring stations and their measuring height is given in the legend. The area of monitoring is denoted in the map on the left by a dashed black circle.

[Figure]

**Fig. X11** PM10 measurements in the southern region of Israel between 7-10 September 2015. The list of monitoring stations and their measuring height is given in the legend. The area of monitoring is denoted in the map on the left by a dashed black circle.

[Figure]

**Author's changes in manuscript:** New PM10 measurement plots from 29 monitoring stations were moved to the "Results" section.

**Referees Comments:**

Referee #2: *I asked colleagues from Israel to help me with... the following..., I am not familiar with Hebrew language... for example: http://www.svivaaqm.net/. Technically, you have at http://www.svivaaqm.net/ 'reports from numerous sites'... The dust storm: at September 7, around 22:00 values above 100 are already recorded for two sites (cities): Kiriat Ata and Nesher (next to Haifa). Haifa region (Northern part of Israel) must be included for more comprehensive analyses. The authors stated that: "Unfortunately, there are no PM10 measurements in northern Israel".(Line 233) During the dust event there were seven sites that measure PM10 that were available. Figure 5 is not correct therefore, it covers more area, than depicted in Figure 5, including the Haifa Bay area.*

**Author's response:** All PM10 measurements, including the Haifa region, were included in our calculations (Fig . X9.) Nevertheless, these stations do not cover what we define as northest region in Israel (denoted by a yellow circle in Fig. X12) named "Golan heights" (also referred by Gasch et al.).

**Fig. X12** The northern region (yellow circle) where no PM10 measurements are held.

[Figure]

Measurements in the Golan heights would have given us a perspective whether the dust plume prevalence over Israel on 8 September 2015, was indeed from the northeast or rather unified but measured mainly by the highest monitoring stations since the dust plume hadn't thus far subsided.

**Author's changes in manuscript:** New plots presenting 3-hr concentration of PM10 from 29 stations instead of an average concentration per region.

**Referees Comments:**

Referee #2: *Why did the authors not include PM2.5 sites data.*

**Author's response:** We did not include the PM2.5 measurements although we've already analyzed them) since the intensity of the dust storm was less profound in PM2.5 measurements (see Fig. X13) and less monitoring stations were available compared to PM10 measurements. We decided PM10 would be sufficient.

**Fig. X13** An example of daily PM2.5 concentrations between 4-15 September 2015 from of 20 monitoring stations with available data during the dust event. The monitoring stations were separated to 6 regions by the combination of north/south/center/inland/shoreline. The measurements of the stations were denoted by the colors of the different regions (see legend). The amount of stations in each region is given inside the legend.

[Figure]

**Author's changes in manuscript:** We can add a figure of PM2.5 measurements in a format preferred by the referee (e.g., parallel to the PM10 measurements). Fig. X12 is just an example.

**Instruments- AERONET : lines 254-273**

**Referees Comments:**
Referee #1: *"...within a 1 min period": maybe it is better to remove this. From this statement the measurement cycle is not clear: 8 s measurements, 22 s break, 8 s measurements, 22 s break, 8 s measurements? This is more than 1 minute.*
**Author's response:** Comment accepted.
**Author's changes in manuscript:** The sentence was omitted.

**Referees Comments:**
Referee #1: *Figs.. 6 also belong to the results.*
*line 262: As this is a result it should be moved to Sect. 3. Fig. 6: to be moved to Sect. 3.*
**Author's response:** Comments accepted.
**Author's changes in manuscript:** AERONET figure was moved to the "Results" section.

**Referees Comments:**
Referee #1: *I don't understand the x-axis of Fig.6 ? Why is the length of the days different? Where are the night measurements of AERONET from (do you deploy a lunar photometer?)? Is there a specific reason for not choosing a line plot in Fig. 6a (as in Fig. 6b)?*
Referee #2: *P10, Figure 6: the time axis ... the day scale (width) is changing from day to day. E.g., the 8 September is very narrow, the 7 September has a factor of 4 more space..., why?*
**Author's response:** All the data available from the Israeli AERONET sites was delineated in Fig. 6. The length of the days is different due to missing data. Fig. 6 was an attempt to present the availability of the AODs as the extent of available data changed from day to day. We found the presentation of Fig. 6 more informative than a line plot.

**Fig. X14** Daily AOD and Angstöm coefficient from two AERONET sites in Israel

[Figure]

**Author's changes in manuscript:** we changed the plot to daily average (Fig. X14).

**Referees Comments:**

Referee #2: *P10, L264. The Rehovot station did not work, the instrument was out of order? .... or did the station not allow useful data analysis because the AOD was too high? Please clarify and state what was the case...*

**Author's response:** The instrument was out of order the due to power failure in the Weizmann institute.

**Author's changes in manuscript:** We mentioned the instrument was out of order and did not automatically shut down as a result of high AODs.

**Results: lines 276-472**

**Referees Comments:**

Referee #1: *In the "results" -section the discussion should be more elaborated and linked to the findings from the other data sets (provided in part when the MLH is discussed). The discussion of the scientific results must be improved and extended. In the present state the*

*paper is sort of a collection of (useful) pieces of information, but their relationship and their interpretation is not sufficiently elaborated.*

**Author's response:** Comments accepted.

**Author's changes in manuscript:** The "Results" paragraph was rewritten according to the overall comments of the referees.

**Referees Comments:**

Referee #1: *From Fig. 6a it seems that after 11. September 2015 the AOD was still relatively large (by the way: can you give the annual mean AOD of the sites in Israel?). How "typical" are the ceilometer profiles between 11. and 14. September 2015?*

**Author's response:** Annual mean of 2015 of AOD from Sede-Boker : 0.22 . AERONET Weizmann started operating on 6 June 2015, thus the half year mean is: 0.25. Although Sede Boker is situated in the southern Negev desert of Israel, the Average AOD of the "dusty and windy" season on Oct- Nov remained 0.25. Therefore, the AODs above 3 are rare. Between 11-14 September 2015, AODs are large although the PM10 measurements and ceilometer signals infer differently since they do not include the whole air column.  This finding may justify the results of Stavros et al. (2016) and the CALIPSO measurements on the 10 September of an elevated layer at about 3-4 km of dust plume.

**Author's changes in manuscript:** We added a discussion on the comparison between measurements (AERONET, particulate matter, global radiation, ceilometers, Satellite pictures) after 11 September 2015.

**Referees Comments:**

Referee #1: *line 281 ff: Move lines 281–283 to Sect. 2.*

**Author's response:** Comment accepted.

**Author's changes in manuscript:** The sentences were moved to the ceilometer discussion in the "Instruments" section.

**Referees Comments:**

Referee #1: *The absolute values of the attenuated backscatter must be checked: some of the numbers are unrealistic and the corresponding figures are not clear. For example, the labels of Fig. 7c (e.g. 0.000029 in units of $10^{-9}$ $m^{-1}$ $sr^{-1}$) are confusing. lines 281–283: Check carefully the following numbers of the attenuated backscatter; some of them are unrealistic ($7 \cdot 10^{-1}$ $m^{-1}$ $sr^{-1}$).*

*lines 281–283: See also general remarks at the beginning. Add a xy-grid to Fig. 7.Figs. 8–16: What is the unit of the color code? It does not agree with the values given for attenuated backscatter. Are these numbers the "counts"?*

Referee #2: *P12, Figure 7, We need a clear statement, that the range-corrected signals shown in Figure 7 decrease rapidly and is close zero at about 700 to 750m in b,c,d because of the strong laser light attenuation in the dust layer! As long as such a statement is missing*

*the reader may believe to see the full dust layer and the top is at 750-1000m height. To repeat: This is unacceptable. This is simply wrong and unacceptable. Please improve! The basic ceilometer data are signals (let us say ... in units... counts per second), and if they are then range corrected... then you get the dimension 'counts per second times m\*\*2'. So the values are not unitless, but usually given in arbitrary units. Next, by dividing these data by 10ˆ(-9) ... does not change anything. You still have just range-corrected signals. You can only obtain a profile of the attenuated backscatter coefficient if you are able to calibrate this range-corrected signal profile in the tropospheric region with pure Rayleigh backscattering or in the way as described above. So, you show range corrected signals!!! And not attenuated backscatter!!! As mentioned already, you must change .... to range corrected signals in all ceilometer plots!*

**Author's response:** A fundamental mistake. The attenuated backscatter presented in Fig. 7 had already been divided by $10^9$, therefore, $10^{-9}$ should have been omitted from the x-axis label. The numbers shown in the ceilometer plots are signal counts**.**

**Author's changes in manuscript:** Ceilometer profiles were correct to units of attenuated backscatter in the order of $10^{-5}$ $m^{-1}$ $sr^{-1}$. A title of "counts" was added to the legend in all ceilometer daily plots.

**Referees Comments:**

Referee #1: *The authors should be aware that a comparison of brown vs. brown color or brown vs. blue color (Figs. 10–16) is not suitable for a scientific publication. Please use quantitative numbers! Think about plotting coincident attenuated backscatter profiles of the 8 sites (similar to Fig. 7), this may help to see temporal delays in the arrival and decay of the plume.*

Referee #2: *P12. Figure 8,x-axis please show data always from 0-24 local time (or UTC). Again we need proper text for the x-axis and y-axis, as it is the case in Figure 7. P12-15: All the Figures 8,10-16 have no x-axis and y-axis description. This is poor and unacceptable. And again, all these ceilometer color plots suggest that the dust layer was just a few hundred meter thick. This is dangerous! The reason is simply the almost total attenuation of the ceilometer radiation pulses by the rather dense dust layers. This must be made very clear.*

*line 320: "the time corrected from local time to" can be removed.*

**Author's response:** After several trials we found the current plot contours are the most informative. Therefore, we took your advice and added profiles (as in Fig. 7) for each of the ceilometer plots at 4 times of interest: the dust storm penetration (7.9.17, 14:00-15:30 UTC), downward dust transport towards ground level (8.9.17, 11:00-12:30 UTC), dust ascent (9.9.17, 12:00-13 UTC) and the time of CALIPSO passage over Israel (10.9.17, 10:30-12:00 UTC) (Fig. X15).

**Fig. X14** Tel Aviv ceilometer daily plots (top panel) and attenuated backscatter signals (bottom panel) at time of the dust storm penetration to Israel (7 Sep 2015), downward dust transport (8 Sept 2015), dust ascent (9 Sept 2015) and time of CALIPSO passage over Israel (10 Sept 2015). The period of the profiles is denoted by white dashed lines upon the ceilometer plots.

[Figure]

**Author's changes in manuscript:** Updated the ceilometer plots as Fig.X14.

**Referees Comments:**

Referee #2: *P13, L324: again and again: you were able to track the dust layer base only, this must be clearly said. In Figure 8, the numbers for 'your' attenuated backscatter are suddenly up to 15000, compared to values of about 10ˆ(-14) in Fig 7 (b,c,d)? Then in Figs 10-12: up to 10000. And in Fig 13, suddenly only up to 800..., Fig. 14 up to 15000, and Fig15-16 up to 10000. So all this is rather strange...and only reasonable and understandable if we switched to range corrected signals (arbitrary units). So, please change..... to range corrected signals.*

P13, L347: *Plots are given in different scales to highlight the dust features. This is ok, because range corrected signals are shown and the ceilometer performance changes from site to site (from ceilometer to ceilometer). So again, there is a clear need to work with range corrected signals.*

**Author's response:** 6 Out of the 8 of the ceilometers belong to a governmental office. We were not informed whether the "message profile noise h2" was off or on therefore whether an automated range correction was done.

**Author's changes in manuscript:** No changes.

**Referees Comments:**

Referee #1: *lines 328 ff: Obviously attenuated backscatter is converted into particle extinction coefficient (with a relatively large uncertainty inherent in all CL31 measurements; see water vapor absorption mentioned above, unknown lidar ratio, unknown accuracy of the scaling factor) with an estimated lidar ratio. Then, the visibility is estimated according to Koschmider. Which altitude was selected for this conversion? The problem is, that if it is done for a large altitude (maybe 100 m or more) it is difficult to compare this visibility to independent ground based measurements. If it is done for the ground, then the overlap problem is critical. Nevertheless, an order of magnitude agreement should be possible, but please extend this paragraph by explaining all aspects of this comparison.*

Referee #2: *P13, L330: A visibility of 200m (visual meteorological range is defined by an AOD of 3, after Koschmieder for an AOD of 4) according to an AOD of 3 means that the particle extinction coefficient was 15 km-1 or 0.015 m-1 and the backscatter coefficient is then 0.0003 m-1 sr-1 if the dust lidar ratio is 50sr. All your 'numbers' are far far away from these value. This corroborates: It is impossible and dangerous (and thus not justified) to convert the range-corrected signals into optical properties just by taking 'some' conversion factor!*

**Author's response:** We took the visibility observations under 100 m AGL and compared to the attenuated backscatter received at ~500 m AGL. We understand it is misleading and therefore we decided to omit this paragraph.

**Author's changes in manuscript:** The paragraph was deleted.

**Referees Comments:**

Referee #1: *lines 334 ff: "Ceilometers are not provided with an AOD...": I don't understand this paragraph. AOD and MLH are mentioned but it is unclear what the message is. The retrieval of the MLH from ceilometer data is completely different from a retrieval of the AOD (provided that can be determined at all). So, how is the validation ("verify ceilometer") of the ceilometer's performance achieved? Please rephrase and extend this part.*

Referee #2: *P13, L334-L337. I would remove this text on the ceilometer and the AOD upper limit. This is useless. The Vaisala ceilometers are not built for aerosol profiling. The wavelength is bad, the signals are corrupted by water vapor absorption.*

**Author's response:** We intended to explain that we do not know the detection limit of the attenuated backscatter signals at high AODs. Therefore, we cannot put a direct limit to the ceilometer threshold. Instead we tried to use other methods to verify the first layer height. In this subsection we attempted to retrieve the inversion height from the ceilometers and radiosonde. Since aerosols are usually trapped under the inversion height and this height is similar in the radiosonde and WCT calculation as a tool to define the height of the first dust layer.

**Author's changes in manuscript:** We rephrased the sentences.

**Referees Comments:**

Referee #2: *P19, L490: Again, you have to mention the limits of a ceilometer. It was too weak to see the layers higher up. No chance to see the main part of the dust layers and dust layer top.*

**Author's response:** Comment accepted.

**Author's changes in manuscript:** We have listed the ceilometers limits in the Instruments section and referred to these limits in the results.

**Referees Comments:**

Referee #2: *P13, Fig. 9 shows the dust base height. To my opining it is misleading to denote the near-surface layer a 'mixed layer' at these conditions with no vertical exchange P13: The text on this page is poor and needs to be significantly improved.*

**Author's response:** We accept the referee's opinion on the 8 September since the global radiation (Fig. X2) was marginal for a significant thermal creation. However the trend of the wind speed in Beit Dagan, Jerusalem and Haifa sites (Fig. X16 a-c) indicates the intrusion of the sea breeze had occurred (Uzan et al., 2012, Fig. X16,d). Wind speed measurements of 8 September indicate lower values and a delay in the wind speed intensification. As stated in the paper, we assume the change in wind speed is owed to the low radiation (below 300 W m$^{-2}$ in the three sites mentioned above, Fig. X2). Therefore, the thermals were not significant in the formation of the mixed layer height.

**Author's changes in manuscript:** The "Results" paragraph was rewritten according to the overall comments of the referees.

**Fig. X16**

[Figure]

[Figure]

[Figure]

Fig. (8). PBL averaged wind speed diurnal variations in days with an inversion aloft for the following three synoptic systems, PT-W, PT-M and H-W. A total of three profiles based on 347 days for PT-W, 232 days for PT-M, and 198 days for H-W, during June-October 1997-1999, 2002-2005. The profiles were measured by the LAP-3000 profiler (Sec. 2). Also indicated are the time period of sunrise and sunset, the period of maximum solar radiation and the SBF times of entrance and weakening. See text for the method of calculation.

**Referees Comments:**

Referee #1: *line 381 ff: A short paragraph should be included here to prepare the reader: The event is divided into several phases (not necessarily split into single days, one phase can be shorter, another can last for more than one day; development, main phase, decay can be an alternative) according to certain criteria and to highlight consistencies/inconsistencies of different data sets/models (by doing this "Next, we describe the decrease of aerosols aloft on mid-day" in line 299 can be omitted). Then, each subsection such as" Entrance of dust into Israel – 7 Sep 2015" should include the full discussion and interpretation of all available data sets for the corresponding period, i.e., parts from Sect. 2 should be included here whenever applicable.*

**Author's response:** Comments accepted.

**Author's changes in manuscript:** We changed the results section and described the dust storm evolution along it's generation rather than by dates.

**Referees Comments:**

Referee #2: *P15, L383: Please clearly state where the dust layer top was found by Solomos et al. (2017).*

**Author's response:** Stavros et al (not Solomos, my mistake) analyzed by CALIPSO overpass two layers, 2 km (total attenuation was reported up to 1.5 km) and between 3-4 km.

**Author's changes in manuscript**: The overview of Stavros et al. was extended.

**Referees Comments:**

Referee #2: *"The AERONET (Fig. 6) and ceilometer plots (Fig. 8, 11-16) reveal that the first dust plume penetrated Israel at approximately 04:00 UTC". What day? Sept. 7 or Sept 8?*

**Author's response:** *7* September 2015.

**Author's changes in manuscript:** The "Results" paragraph was rewritten according to the overall comments of the referees.

**Referees Comments:**

Referee #1: *line 401: What is meant by "decrease"? Concentration or altitude?*

**Author's response:** We meant the subsidence of the dust plume.

**Author's changes in manuscript:** The "Results " paragraph was rewritten according to the overall comments of the referees.

**Referees Comments:**

Referee #1: line 405:" clearly shown in Fig. 13-16 between 08-16 UTC)": This is indeed hard to see. Can you explain this in a more quantitative way?

**Author's response:** We based our conclusions on the wind direction profiles from radiosonde launches in Beit Dagan. Apart the radiosonde, we do not have auxiliary measurements to prove our assumption. Therefore, if the referee could not relate to our conclusions, we decided to omit this hypothesis.

**Author's changes in manuscript:** The sentence was erased.

**Referees Comments:**

Referee #1: *line 408: Please clarify what" these model findings" are? The vertical distribution of dust (backscatter)?*

**Author's response:** We were referring to the hydraulic jump upstream the dead sea rift valley explained by Gasch et al., as the dust plume entered from east and progressed southwest. On the other, the ceilometers plots revealed the penetration of the dust plume occurred straightaway from north (Ramat David site) to south (Hazerim and Nevatim sites) and to west (Hadera, Tel Aviv, Beit Dagan and Weizmann sites).

**Author's changes in manuscript:** we revised the discussion of the "Results" section and the comparison to previous studies.

**Referees Comments:**

Referee #1: *line 422: 2000 µg/m3: is this in contradiction to 9800 µg/m3 in line 416*?

**Author's response:** We were referring to the second "jump" in PM10 measurements which occurred simultaneously in all monitoring stations on the 8 September at~17:00 (UTC+2) after the extreme values of 9800 µg/m3 were measured only in the high altitude stations on the 8 September at ~ 12:00 (UTC +2).

**Author's changes in manuscript:** We rephrased the sentence in the scope of the "Results" section.

**Referees Comments:**

Referee #1: *line 437: "...limited radiative transmitted...": what does this mean?*

**Author's response:** We meant the decrease in the global solar radiation.

**Author's changes in manuscript:** The sentence was corrected and updated with additional data of global radiation measurements in Israel (23 sites across Israel) focusing on the 8 and 9 September 2015.

**Referees Comments:**

Referee #2: *P17, L435: At very high optical depth as on 9 Sep, I would assume that convective motions in the PBL as well as a sea breeze winds cannot develop. Are you sure that sea breeze developments were possible at these days with almost no sun and differential sea/land heating? Please keep the discussion free of speculation.*

**Author's response:** On the 8 September, the maximum wind speed was below 3m/s and maximum global solar radiation up to 264 W m-2. The ability to generation of thermals under these would be rather weak. on the next day, on the 9 of September the maximum global radiation more than doubled reaching 621 (w m-2) and wind speed increased to a maximum of 5 m s-1 at 13:00 (UTC+2). At these conditions, convection and creation of thermals is possible.

**Author's changes in manuscript:** We've added an explanation and figures regarding the possibility of thermals creation on the 9 September.

**Referees Comments:**

Referee #1: *Fig. 17: This figure is misleading as the range of the color code is different from Fig. 14. This should be pointed out clearly. As long as the inter-comparison with the AOD (AERONET) is qualitative only, it would be helpful to show the typical(?) background(?) values of the AOD and attenuated backscatter of the days before the event for comparison.*

Referee #2: *P17.... Figure17 indicates that there was dust higher up. The AOD decreased towards 0.5-1 on 12-14 Sep. A perfect mixing layer could develop now up to 750 m, as seen on 13 and 14 Sep. Nice to see, that the aerosol dried in the PBL during the morning hours and thus the color of the range corrected signals changed from red to green and blue (for dry particles producing less backscatter later on).*

**Author's response:** Comments accepted.

**Author's changes in manuscript**: We added an underlined the explanation of the different signal counts (up to 3000) of figure 17 in the manuscript compared to the rest of the ceilometers plots (up to 15,000) and daily AOD measurements from both AERONET sites in Israel, Sede-Boker and Weizmann, for the whole month of September 2015 (Fig. X13). For a comparison of a "typical background", we created ceilometer Tel Aviv plots for 1-8 September based on a scale of 3000 counts (Fig. X17). If the referees find the plots contribute to the paper, we will delineate similar plots for the rest of the ceilometers.

**Fig. X17**

[Figure]

Ceilometer Tel Aviv plots
(scale of 0-3000 counts)
from 1 to 8 September 2015

**Referees Comments:**

Referee #2: *P17, L444: You state: The ceilometer reveals total clearance on 10 Sep! But the Weizmann Institute AERONET shows AODs of 2 and more on 10 Sep! What is wrong, what is true? Please clarify?*

*P18, L461: The AOD was >1.0 all the time on 9 and 10 Sep...until 12 Sep. What do you thus mean with dissipation of dust?*

*P19, L498-499: When were the AE values high again? They were continuously <0.5 even on 14 Sep (Weizmann AERONET).*

**Author's response:** The "clearance" we mentioned was referred to the lower part of the atmosphere (up to ~1 km), as the PM10 values decreased considerably from the 11 September (Fig X18). By 11 September the amount of signal counts from all 8 ceilometers declined (Fig X19 for example). The possibility of a profound decrease in the ceilometer signal counts while the AE is still high is owed to the fact that AERONET measures the whole atmospheric column, including the second dust layer indicated by CALIPSO (Fig. X) and Mamouri et al., (2016) to be at ~ 2-4.5 km. The ceilometer on the other hand was capable to detect only the first km. This comparison may point out residual of dust plume aloft even at 15 September, in contrary to MODIS –aqua imagery (Mamouri et al., 2016) and AOD<1 from MSG SEVIRI (Fig. X20).

**Author's changes in manuscript:** We have added a discussion on the dust ascent in the form mentioned above.

**Fig. X18**

The mixed layer height based on 00Z and 12Z profiles from radiosonde Beit-Dagan vs $PM_{10}$ daily average concentration from the Ceilometer Hadera site

[Figure]

**Fig. X19** Daily attenuated backscatter plots from ceilometer CL51 Weizmann-Rehovot (central Israel). Y-axis : Height (m AGL), X-xais: Time (UTC) , Scale : 0-15,000 signal counts. Ceilometer profiles were averaged by 15 minutes running average for better SNR.

[Figure]

**Fig. X20** NAScube optical thickness based on MSG SEVIRI from 15 September 2015
(source: http://nascube.univ-lille1.fr/cgi-bin/NAS3_v2.cgi)

[Figure]

**Referees Comments:**

Referee #1: *line 454: 250 m: is this the vertical extent or the altitude?*

Referee #2: *P18, L454:  ...as a dust layer of 250 m thickness (fig 11-13, 15-16) penetrated Israel at a height of 1000-1500m.... How do you know the depth of the dust layer? The ceilometer fails to see higher up.... So, how do you know? I would leave out to mention any dust layer depth.*

**Author's response:** Comments accepted.

**Author's changes in manuscript:** Considering the ceilometers limitations to detect attenuated backscatter signals above a dense dust layer, we omitted the assumptions regarding the vertical extent of the dust layer.

**Referees Comments:**

Referee #1: *line 486: The PM10 measurements are considered as in-situ measurements, not remote sensing.*

**Author's response:** Comments accepted.

**Author's changes in manuscript:** We corrected the sentence referring to PM10 measurements as in situ.

**Conclusions and discussion: lines 480-514**

**Referees Comments:**
Referee #2: P19: The conclusions have to be rewritten completely after improving all the text before along the lines this review and the other review.
**Author's response:** Comment accepted.
**Author's changes in manuscript:** We rephrased the "Conclusions and discussion" section according to the referees' comments.

**Referees Comments:**
Referee #2: What sources of errors do they have when using the ceilometers? They should critically state the limitations, disadvantages and advantages. Including comparison between different ceilometers that authors used for the analyses. Without this comprehensive critical discussion on authors findings, the outcome of the paper is doubtful.
**Author's response:** Comments accepted.
**Author's changes in manuscript:** We listed the ceilometers limitations in the Instruments sections. In the discussion section we referred to these limitations in the as part of the process of evaluation.

**Referees Comments:**
Referee #2: *Figure 18: The major claim - the dust penetrates from the East. But- combining PM10 from the Haifa Bay area, there is a "jump" towards values of 2500-3000 micrograms/m3 at 8 of September, similarly to East, which means it has two entrances/sources. Do the authors see the "North region" dust entrance using ceilometer data? The authors must justify what new information they get using ceilometer more clearly than in Figure 18, what new insights they get about the extreme dust event? And number/summarize all "new insights" about the event that they discover*
**Author's response:** We analyzed PM10 and PM2.5 measurements from all available monitoring stations in Israel. Unfortunately, we did not recognize a second jump from northern region compared to the rest of the country. To emphasize our conclusions, we prepared Fig. X21 presenting PM10 3 hr concentrations, including monitoring stations the referee mentioned: Haifa Bay, and the northest station in Israel (Karmiel). We added three more stations from the central shoreline (Hadera), eastern Israel (Jerusalem) and southeast (Arad). To our opinion, it is difficult to declare a second "jump" on the 8 September exclusively in the northern sites.

**Fig. X21**

[Figure]

We estimate the dust plume penetrated Israel was disclosed firstly by the global radiation measurements (Fig. X2). After the dust subsided (based on personal knowledge in environmental dispersion models, the physics behind the mathematical assumptions treat PM10 by the characteristics of gaseous dispersion), PM10 measurements delineated the dust plume dispersion on ground level (Fig. X22). On the other hand, TSP (Total suspended particles, larger particles than PM10 but below 45 µm aerodynamic diameter) maximum concentrations were measured on the night between 8-9 September (Fig. X23). This may indicate of local meteorological conditions generating resuspension not rather a consequence of the dust plume descent.

**Fig. X22** PM10 maximum concentration 8 September 2015 measured across Israel in 9 sites. Indication of height of each measuring site (ASL) is given upon the map.

[Figure]

**Fig. X23** TSP measurements from three sites adjacent to the Dead sea

[Figure]

**Author's changes in manuscript:** We suggest to exclude Fig. 18. Instead we my insert the discussion disclosed above with auxiliary data presenting the global, direct and diffused radiation from several sites cross Israel (Fig. X25-X30):

**Fig. 25** Global, direct and diffused radiation from the northest site

[Figure]

**Fig. 26** Global, direct and diffused radiation adjacent to ceilometer Beit Dagan

[Figure]

**Fig. 27** Global, direct and diffused radiation from the highest measuring point**

[Figure]

**Fig. 28** Global, direct and diffused radiation from southern Israel**

[Figure]

**Fig. 29** Global, direct and diffused radiation from southern Israel**

[Figure]

**Fig. 30** Global, direct and diffused radiation from southern Israel

[Figure]

Hourly average solar radiation measurements from Yotvata site (70 m ASL) 6-11 September 2015

━━ Direct radiation (W m-2)  ━━ Global Radiation (W m-2)  ━━ Diffused radiation (W m-2)

[Figure]

**Referees Comments:**

Referee #1: *line 488:" ...for the first time, such an event is vertically analyzed using an array of ceilometers...". On the one hand this is true, on the other hand it is slightly misleading as the vertical structure (by other means) has already been investigated. So it might be advisable to use a less strong statement in the next sentence (a note on the limited measurement range).*

Referee #2: *P19, L488: for the first time such an event is vertically analyzed...... this is misleading because Mamouri et al. already used lidar to characterize the dust storm. You probably wanted to say, for the first time .... with a ceilometer network. However, you should mention that there were already lidar studies with Cyprus lidar and CALIOP lidar, and now you come with a ceilometer network study...... Then this would be more clear, and of course this is a new aspect.*

**Author's response:** Comments accepted.

**Author's changes in manuscript:** We rephrased the sentence and emphasized the contribution of the ceilometer measurements to the analysis of the lower part of the atmosphere (from ground level up to ~1 km) as a completion to previous studies concentrating on the generation and propagation of the dust plume down to ~1.5 km.

**Referees Comments:**

Referee #1: lines 492, 497:" plume"!

**Author's response:** Comment accepted.

**Author's changes in manuscript:** Typing mistakes were corrected.

**Referees Comments:**

Referee #1: *line 494:" mainly of mineral dust": where is this information coming from?*

**Author's response:** We based our conclusions on Sede-Boker AERONET Angstöm coefficient measurements (Fig. X13)**.** Mamouri et al., (2016) studied the dust layer particle linear depolarization by an EARLINET lidar stationed in Limassol Cyprus. They concluded the linear depolarization ratio of 0.25-0.32 on 7 and 10 September, indicated the dominance of mineral dust (the lidar was inoperative on the 8 September).

**Author's changes in manuscript:** We added the citation to the AERONET Angstöm coefficient measurements in Israel and referred to the conclusions from Mamouri et al., (2016).

**Referees Comments:**

Referee #2: *P19, L494: As a result, ...... of what?*

**Author's response:** As a result of the low boundary layer.

**Author's changes in manuscript:** We rephrased the "Conclusions and discussion" section according aforementioned overall comments of the referees.

**Referees Comments:**

Referee #2: *P19, L502-504: This is speculation, at least to my opinion. Be more save with your statements.*

**Author's response:** Comment accepted.

**Author's changes in manuscript:** The sentence was omitted and a comprehensive analysis of the meteorological measurements (global radiation, direct radiation, diffused radiation, ground temperature, wind speed) and environmental measurements (PM10, PM2.5, TSP) were added in the attempt to explain and reveal the meteorological conditions held as the dust storm prevailed in Israel.

**Referees Comments:**

Referee #2: P19, L506-511: Again, dangerous statements. I would remove. Otherwise, you need to check the CALIPSO overflight over Israel to corroborate your speculative suggestions. However, the modeling papers of Solomos et al and Gasch et al. (partly based on model plus CALIOP results) do not leave room for statements like ... who knows to what height the dust plumes reached over Israel. To my opinion, in the Middle East dust layer top was up to 4-5 km height everywhere.

**Author's response:** We accept the comment. Additional data from the CALIPSO passage over Israel on the 10 September (given here in Fig. X) indeed shows a dust plume between ~2.5-4.5 km.

**Author's changes in manuscript:** We omitted the statement and refereed to the ceilometers data in the context of the evolution of the dust plume at the lowest level of the troposphere. We stated the ceilometers' limited ability to detect attenuated backscatter signals above the dust layer detected at ~ 1 km.

**References: lines 546-637**

**Referees Comments:**
Referee #2: *P21, L554: No authors.*
**Author's response:** The reference is a report edited and distributed by the U.S Environmental protection Agency with no specific authors mentioned upon the report.
**Author's changes in manuscript:** No change.

**Referees Comments:**
Referee #2: *P23, L621: TOASJ...?*
**Author's response:** Acronym for The open atmospheric science journal.
**Author's changes in manuscript:** The acronym was converted to the full name of the journal.

---

## Author Response (AR1)

**Author's Response:**

We wish to thank both referees for their comprehensive and elaborated comments. Despite some mistakes they have revealed in the manuscript, the referees took the time and effort to explain the source of the inaccuracies and offer good advice and practical solutions.

Overall, we have received extensive comments on all sections and figures by both referees. Therefore, we've decided to combine the referees' comments and write our point to point response along the manuscript sections:

- Abstract
- Introduction
- Instruments
- Results
- Conclusions and discussion
- References

We hope this structure is acceptable and easy to follow.

Sincerely,
Leenes Uzan, Dr. Smadar Egert and Prof. Pinhas Alpert

**Abstract: lines 39-51**

**Referees Comments:**

Referee #2: *The abstract does not just summarize the paper properly. The abstract should be compact, i.e., as short as possible and just cover the contents of the paper: The main goal, the instruments and methods used, and main findings…. no outlook..., no unnecessary (motivating) statements that can be given in the introduction....*

**Author's response:** Comments accepted.

**Author's changes in manuscript:** The abstract was rewritten as recommended.

**Introduction: lines 56-145**

**Referees Comments:**

Referee #1: *line 61: I recommend to add a short paragraph here introducing the scientific objectives and the benefit of the paper. Something like" One of the strongest events ever... It was investigated already by several studies... However, ... is missing. Therefore, in our investigation we focus on...". Then the short description of the event and the review of the existing publications can follow as is.*

Referee #2: *The introduction could be more straight forward, as follows: There was a huge dust storm in the Middle East, however, the dust forecast models failed. The question was then: Why? This question motivated Solomos et al. (2017) to run a cloud resolving model system. They found the most probable reasons. Please state their findings in the introduction! Afterwards, Gasch et al. (2017) used the new IKON/ART model system with rather high resolution (a global cloud-process-resolving model!!!!) and also investigated this dust storm... and discussed the storm in even larger detail.... and concluded.... Please read the final version and present their final conclusions. This would be a nice introduction, very informative, so that the reader would learn a lot. And then you could provide the motivation for your own ceilometer study.... Because open points remained, and this historical dust storm must be documented for a variety of regions in the Middle East.*

**Author's response:** Comments accepted.

**Author's changes in manuscript:** Keeping in thought this paper is supplementary and an important insight of the extreme September 2015 dust storm, we significantly expanded the overview of previous studies describing the origin of the dust storm, analysis of observations and posteriori model runs. On our behalf, we added information describing the weather conditions and analyzed the dust event evolution over Israel in the lower atmosphere (up to 1 km). The data we present can support verification of state of the art model simulations (actually already been done by Gasch et al., which cited our presentation from the EGU conference).

**Referees Comments:**

Referee #1: *line 62: Fig. 1 seems to be the justification that it is worthwhile to study the event. A detailed explanation of Fig. 1 is however missing.*

*line 71 (figure caption): The AOD derived from MSG (mention the sensor!) is shown. Make clear that Fig. 6 shows the AOD from AERONET for comparison.*

Referee #2: *Page 3, Fig.1: The shown AOD range... is that the range of trustworthy values? Because the dust AODs were much higher than 2.7. So one could enlarge the color scale... How large is the uncertainty in the MSG data, source for the data (http....) should be mentioned.*

**Author's response:** Comments accepted.

**Author's changes in manuscript:** Fig 1. was moved to the " Results and discussion" section where it was discussed. The caption was updated with the MSG SEVIRI sensor and source of the MSG data. The uncertainty of AOD measurements from MSG SEVIRI can rise up to 15% (Mei et al, 2012). The Sede-Boker AERONET AOD values at 12 UTC were added upon Fig.1 in the location of the Sede-Boker site.

[Figure]

Figure 1. Aerosol Optical Depth (AOD) at 12 UTC analyzed by NAScube (Université de Lille) based on imagery from the Meteosat Second Generation (MSG) Satellite (by a combination of the SEVIRI IR8.7, IR10.8 and IR12.0 channels). The map includes indication of the Sede Boker AERONET site (black square) and its AOD value at 12 UTC.

**Referees Comments:**

Referee #1: *line 68: How does the visibility and the reference to AERONET fit together? Give a citation for AERONET.*

**Author's response:** Comments accepted.

**Author's changes in manuscript:** The referee is correct. AOD from AERONET could not be a reference to visibility. The AERONET reference was erased. We added a citation for the AERONET data.

**Referees Comments:**

Referee #1: *lines 78 ff: Give citations for Meteoinfo, MODIS, EARLINET, ICON-ART.*

Referee #2: *P3, L78, reference for Meteoinfo or http.*

**Author's response:** Comments accepted.

**Author's changes in manuscript:** Citations were added.

**Referees Comments:**

Referee #1: *line 91: Avoid acronyms in cases when it is only used once or twice, e.g. RST and SBF.*

**Author's response:** Comment accepted.

**Author's changes in manuscript:** Acronyms used up to twice were deleted.

**Referees Comments:**

Referee #1: *Use a common format for dates: "8. September 2015" instead of "08.09.15" is certainly a better option.*

**Author's response:** Comment accepted.

**Author's changes in manuscript:** The format of all dates was changed in to the suggested format.

**Referees Comments:**

Referee #1: *line 98: When referring to CALIPSO, mention that it was found that the top of the dust layer was at about 3–4 km (though the overpass was several hundreds of kilometers east of Israel). Even better: include a (short?) discussion of CALIPSO measurements (hopefully closer to Israel than in Gasch et al., 2017) into the results-section showing where the upper boundary of the dust layer has been. For this purpose, quick looks from the CALIPSO website could be sufficient. Then, the measurement range of the ceilometers can be highlighted (it is doubtful that the ceilometers can fully penetrate the dust layer at all times).*

**Author's response:** Comments accepted.

**Author's changes in manuscript:** We added two images (total backscatter and extinction coefficient) from the only passage of CALIPSO over Israel, on 10 September 2015. The images reveal a dust layer between 2-4 km ASL (see Fig.X below).

[Figure]

Figure X. A map of CALIPSO satellite overpasses (right panel). The only overpass above Israel was on 10 September 2015 at 11:00-11:10 UTC (indicated by a red line). On the left panel are the CALIOP lidar products of total backscatter and extinction coefficient. Indications of the overpass over Israel is given by a black rectangle.

**Referees Comments (overview of Gasch et al.):**

Referee #2:

*P3, L94, the model better explained..... compared to what?*

*P3, L94, ART, give reference.*

*P4, L100, please check the finally revised version of the Gasch paper. I asked these authors, the final version should be out soon.".*

**Author's response:** Comments accepted.

**Author's changes in manuscript:** We expanded the summary of the Gasch et al. (2016) paper, explaining the model set up, ART model reference and the conclusions of the dust storm process. The final version of the Gasch et al (2016) was approved on the 15 November 2017 and will be cited accordingly.

**Referees Comments (overview of Stavros et al.):**

Referee #1:

*line 115: Where was the lidar located?*

*line 118: Were the model results compared to CALIPSO or EARLINET (both are lidars)?*

*line 120: "...but the ability to predict the details were partial..": I don't understand this sentence. Is this relevant for this paper?*

*line 122: Lidars don't provide" aerosol concentration", but backscatter coefficients (or extinction coefficients, depended on the system).*

**Author's response:** Comments accepted.

**Author's changes in manuscript:** We extended the summary of the Stavros et al., research and referred to the lidar located (Limassol Cyprus), the comparison of the EARLINET lidar to the model results and the estimation of the aerosol concentration based on the lidar ability to reproduce the strength of the dust event.

**Referees Comments:**

Referee #1: *Use "lidar" instead of "LIDAR".*

Referee #2: *P4, L115, L118, L122: lidar... not LIDAR*

**Author's response:** Comments accepted.

**Author's changes in manuscript:** "LIDAR" was changed to "lidar" throughout the text.

**Referees Comments:**

Referee #1:

*line 130: What is meant by" scattering properties"?*

**Author's response:** Comment accepted.

**Author's changes in manuscript:** " Scattering properties" was corrected to "optical properties".

**Referees Comments:**

Referee #1:

124: *Give citation to " deep blue algorithm".*

**Author's response:** Comment accepted.

**Author's changes in manuscript:** A citation for deep blue algorithm was added (Hsu et al., 2013).

**Referees Comments:**

Referee #2: *P5, L134, Is there no discussion in the literature on dust-radiation-dynamics relationship? I believe, the SAMUM group (Tellus 2011 special issue) investigated the relationship between dust (and smoke) and the radiation field and changes in the air flow (dynamics) as a result of the impact of dust and smoke on the radiative fluxes. Such dense dust layers as in September 2015 certainly had a huge impact on the radiation budget and significantly changed the weather pattern and thus air mass transport. This may explain why the routine dust forecast models failed because the forecasted dust concentration was too low to produce a significant radiation effect on the weather pattern (and dynamics) and dust transport in the model.*

**Author's response:** Comments accepted.

**Author's changes in manuscript:** The cited reference of Pu et al., (2016) did not discuss the effect of dust particles on the thermal radiation. Following the referees' remark, an elaborated discussion based on ground level radiation was added to "Results and discussion" section.

**Referees Comments:**

Referee #1: *line 139: "So far, no attempt has been done to relate the models findings...". This seems to be in contradiction to the previous review of publications. The benefit of this paper is the provision of vertical profiles of the dust in the lower part of the troposphere and the continuous measurements at 8 sites. This helps to obtain a more complete picture (with high resolution) of the event over Israel: this is a valuable contribution. "...spatial, temporal and vertical...": vertical can be omitted because it is "included" in spatial.*
*line 140: "display" - "discuss".*

Referee #2: *P5, L138-L142: You must clearly say in the beginning how the ceilometer network can contribute: Ceilometers can detect the dust layer base and provide some information about the lower part of the dust layer and, very important, the downward transport towards the ground. This is a good and valuable contribution to atmospheric science. On the other hand, not more than that! But this is fine! Nevertheless, you need to provide the limits of such systems! Very clearly! At these high AODs, there is no chance to detect most of the dust and the dust layer top.*

**Author's response:** Comments accepted.

**Author's changes in manuscript:** We rephrased the paragraph referring to the ceilometers' contribution (lines 138-142) and listed the limitation of the ceilometers in general and under high AOD conditions.

**Manuscript lines 148-273 (Sect.2 Instruments):**

**Referees Comments:**
Referee #1: *A (short) section on the satellite data used in this study is missing and might be included.*
**Author's response:** Comments accepted.
**Author's changes in manuscript:** A section of satellite data regarding imagery from MSG (SEVIRI), Terra (MODIS), Aqua (MODIS) and CALIPSO (CALIOP) satellites was added to the "Instruments" section.

**Referees Comments:**
Referee #2: *The description of the instruments and data sets on the one hand, and the presentation and discussion of scientific results on the other hand should be clearly separated. Presently e.g. the" radiosonde section" includes results (lines 211 ff.) that should be moved to Sect. 3*
**Author's response:** Comments accepted.
**Author's changes in manuscript:** Presentation and discussion of scientific results were moved to the "Results and discussion" section.

**Referees Comments:**
Referee #1: *line 152: "... to the atmosphere above its measuring point". Must be" top of the atmosphere". Omit" above its measuring point": strictly it is a slanted column.*
*line 159: "...to add the vertical aerosol distribution...". The distribution is not added, but information on the distribution. Check the whole paper for similar wordings that are not fully correct.*
**Author's response:** Comments accepted.
**Author's changes in manuscript:** We rephrased these sentences as recommended.

**Referees Comments:**
Referee #1: *The role of the scaling factor (lines 282 ff) should be moved to the description of the instruments and data sets. In particular, the ceilometer section must include more relevant information.*
*line 165: To be more precise the first sentence and the corresponding citations should be modified: Start with a general statement on the potential of lidars to observe (dust) aerosols: here you can cite Mona et al. (already mentioned, but missing in the list of references), more papers from the EARLINET community such as Wiegner at al. (2011), Papayannis et al. (2008), Ansmann et al., (2003), all in J. Geophy. Res., or other papers. Then, you can state that with recent improvements of the ceilometers' hardware these eyesafe single-wavelength systems are getting more and more important in particular when implemented as network (move the citation of Wiegner et al., 2014, to this place). The*

*Weitkamp-paper is also missing in the reference list – so I don't know why it is included here. Finally, Vaisala's CL31 can be introduced but note that the cited Mü̈nkel-paper from 2004 is on the CT25k-ceilometer, not on the CL31. So replace this citation by e.g. Mü̈nkel et al., (2011)1. Kotthaus et al. (2015) (cited later) is indeed on the CL31. Subsection" ceilometers": This paragraph needs major revisions. A few points: Make clear what you want to measure/determine: mixing layer heights, general structure of layers, heights of lower and upper boundaries, attenuated backscatter, particle backscatter coefficients or whatever. Depending on the desired output the requirements on the data evaluation are different.*

Referee #2: *It seems to me that the authors try to avoid to clearly state: The ceilometer is of rather limited use in dust plume tracking. We were not able to see the full dust layer. But such a statement is required! ..and will not disturb the main goal of the paper. As reviewer I have to say: This is not acceptable and has to be improved! In cases with thick dust layer with optical depth >1 the transmitted (rather weak) radiation pulses of these ceilometers are immediately attenuated in the lowest part of the dust layers".*

**Author's response:** Comments accepted.

**Author's changes in manuscript:** The ceilometer paragraph was changed.  We described the ceilometers limitations including the instrument specific characteristics which affect the quality and availability of the attenuated backscatter profiles, calibration methods, rage detection limitations by the overlap function and determination of the sensitivity of the attenuated backscatter signal to relative humidity. Nevertheless, we portrayed the contribution of the ceilometers in the analysis of dust plume development in the lower part of the troposphere (up to 1 km AGL).

**Instruments - Ceilometers:  lines 164-188**

**Referees Comments:**

Referee #1: *Include the discussion of the" scale factor" here: it is given by the manufacturer and it is unknown to the user where it comes from and how accurate it is. What is the purpose of it? Is it just a" mean" conversion factor of counts (no unit) to attenuated backscatter in $m^{-1}\,sr^{-1}$?*

**Author's response:** The ceilometer attenuated backscatter profiles are automatically corrected by:

*An internal calibration to convert signal counts to attenuated backscatter multiplying by $10^{-9}$.  The internal algorithm (resulting in $10^{-9}$), unknown to the user, is suitable for al ceilometer types (C. Muenkel, private communication, September 2017).

*A cosmetic shift of the backscatter signal to better visualize the clouds base.

*An obstruction correction when the ceilometers' window is blocked (by an obstacle).

*An overlap correction to the height where the receiver field of view reaches complete overlap with the emitted laser beam.

**Author's changes in manuscript:** Data referring to the automatic correction of the ceilometer output profiles was added to the "Instruments" section.

**Referees Comments:**

Referee #1: *When you mention the wavelength of 910 nm, you should also mention that the signals must be corrected for water vapor absorption whenever you want to quantitatively derive any aerosol related quantity (e.g. link to AOD); here the citation of Weigner and Gasteiger (already included in the reference list) should be added.*

**Author's response:** Following Wiegner et al., (2014, 2015), the ceilometer wavelength range (given as 905 ±3 nm) is influenced by water vapor absorption. However, in the case of aerosol layer detection, the water vapor distribution has a small effect on the signal change (indicating the MLH or an elevated mixed layer) because the aerosol backscatter itself remains unchanged. Consequently, except for a case of a dry layer in a humid mixed layer height, the water vapor is unlikely to lead misinterpretation of the aerosol stratification.

**Author's changes in manuscript:** The aforementioned explanation was added to the "Ceilometers" subsection. In our study we did not attempt to derive aerosol quantity as our ceilometers were not calibrated. We have rephrased all sentences that could have mislead the reader in this manner.

**Referees Comments:**

Referee #1: *As a consequence of Kotthaus et al. you should give the firmware version of your ceilometers.*

**Author's response:** Unfortunately, 6 out of the 8 ceilometers belong to a governmental office which does not allow to publicize their firmware data. Yet, we have been confirmed by the exclusive supplier of the ceilometers in Israel that the combination of firmware and hardware was done according to Kotthaus et al., recommendations.

**Author's changes in manuscript:** For the remaining 2 ceilometers we have added technical data summarized in Table X:

Table X. Ceilometer technical information

| Location | Type | Engine board | Receiver | Transmitter | Firmware |
|----------|------|--------------|----------|-------------|----------|
| Beit Dagan | CL31 | CLE311 | CLR311 | CLT311 | 1.72 |
| Weizmann | CL51 | CLE321 | CLRE321 | CLT521 | 1.03 |

Moreover, the title of Table 2 in the manuscript refered to all ceilometers as CL31 type, while ceilometer Weizmann (Rehovot) is a CL51. Therefore, we added Table X1 as follows:

Table X1. Ceilometer configuration

| Location | Type | Time resolution(sec) | Height resolution (m) | *Height range (km) |
|---|---|---|---|---|
| Mount Meron | CL31 | 16 | 10 | 7.7 |
| Ramat David | CL31 | 16 | 10 | 7.7 |
| Hadera | CL31 | 16 | 10 | 7.7 |
| Tel Aviv | CL31 | 16 | 10 | 7.7 |
| Beit Dagan | CL31 | 15 | 10 | 7.7 |
| Rehovot (Weizmann) | CL51 | 16 | 10 | 15.4 |
| Nevatim | CL31 | 16 | 10 | 7.7 |
| Hazerim | CL31 | 16 | 10 | 7.7 |

* Height range dependents on sky conditions and is limited as AOD increases.

* In all ceilometers but in Beit Dagan site, data acquisition was limited to 4.5 km based on the BLview firmware.

**Referees Comments:**

Referee #1: *You should mention that the overlap correction is automatically performed by the proprietary software and that it is not disclosed to the user. If the overlap correction creates signal artefacts (at 50–100 m), different for each ceilometer, is this crucial for your scientific objectives (if yes, what are the consequences?).*

**Author's response:** In our study we focused on the downward motion of the dust plume and its effect on the creation of the mixed layer height which did not reach below 200 m AGL. Therefore, artefacts up to 100 m AGL were not crucial.

**Author's changes in manuscript:** An explanation of the automatic overlap correction was added to the text.

**Referees Comments:**

Referee #1: *Be clear with the temporal resolution (it can be selected by the user in the range from ... to ... s; in this paper 16 s was selected, is there a special reason for 15 s for some ceilometers?). 7.7 km applies to all ceilometers, not only that at Beit Dagan. Do you know the pulse energy of the CL31?*

Referee #2: *P7, L178: ...up to 7.7 km height AGL. Yes the ceilometer may measure up to 7.7 km height, but only for clear sky conditions with AOD of the order of 0.2, at least clearly below 1. One has to state that. In case of the dust storm, all delivered 'counts' above 1 km were more or less just background noise!.*

*P7, L188 ... The Beit Dagan Ceilometer measures up to 7.7 km. Yes, as mentioned, under clear sky conditions. At dust storm conditions with AODs from 2 to probably 5 and more, the ceilometer measures just noise from 1 to 7.7 km height. So this is an unacceptable statement. Please change.*

**Author's response:** All ceilometers were capable to measure up to 7.7 km with a range gate of 10 m under the limitations of sky condition and a decrease in SNR with height. The data acquisition was limited to 4.5 km by BLview default except for ceilometer Beit Dagan.

Ceilometer Beit Dagan has a temporal resolution of 15 s set by the manufacturer. The pulse energy of the CL31 1.2 µWs ± 20%.

**Author's changes in manuscript:** We rephrased the sentence and pointed out the limitations of the ceilometers to measure up to the maximum height range declared by the manufacturer.

**Referees Comments:**

Referee #2: *In the radiosonde plots there is always written: ... launching sites... please improve!*

**Author's response:** Comment accepted.

**Author's changes in manuscript:** The radiosonde plots were changed. We have omitted the profiles from the Cyprus site and concentrated on the low level (up to 1200 m ASL) of the temperature and relative humidity profiles from Beit Dagan in order to better visualize the MLH descent once the dust storm prevailed.

[Figure]

Figure X2. Radiosonde Beit Dagan profiles at 12 UTC between 7-10 September 2015 of relative humidity (left panel) and temperature (right panel).

**Referees Comments:**

Referee #2: *P7, L173: Do you think that you would find the true mixing layer height (when applying the wavelet analysis) under such dense dust conditions? I am not sure! Usually you have the polluted mixing layer and the clear free troposphere on top. At these conditions, the wavelet technique works well. Now you have the opposite. And there was almost no solar heating of the ground (almost no convection), just a residual (less dust laden) layer below the dust layers. What you detect and interpret as mixing layer top is to my opinion just the other way around: the dust layer base height. One should state and discuss this point more clearly. At AODs of 2 and more there is no convection left to lead to well mixed conditions. The dust layer is warmer than the lowest, near-surface tropospheric layer and produces the temperature inversion observed with radiosonde.*

**Author's response:** Ground level measurements of global radiation and ground temperature (measurements 10 cm AGL) from 33 sites (Fig. X3-X7) revealed the process of ground heating on 9 September was possible as the maximum global radiation measured at 12 (UTC+2) reached 500-700 (W m-2) in 19 out of 22 sites (Fig. X8) and the ground temperature reached the values measured prior to the dust storm. Nevertheless, WCT method (seeking the derivative which is the peak value) on the 8 September was probably a result of subsidence of the dust plume as suggested by the referee.

**Fig. X3** Ground temperature (10 cm AGL) from northern Israel.

[Figure]

**Fig. X4** Ground temperature (10 cm AGL) from mountain sites in Israel.

[Figure]

**Fig. X5** Ground temperature (10 cm AGL) from valley sites in Israel.

[Figure]

**Fig. X6** Ground temperature (10 cm AGL) along the coast of Israel.

[Figure]

**Fig. X7** Ground temperature (10 cm AGL) from southern Israel.

[Figure]

**Author's changes in manuscript:** We have omitted the figure comparing the MLH as identified by the the ceilometer and radiosonde from Beit Dagan site. Instead we dedicated an anlysis referring to the spatial and temporal variations in solar radiation supplemented Fig. X8-X9:

[Figure]

Figure X8. A map maximum global solar radiation from 22 sites measured at 10 UTC (midday) on 8 September 2015 (a) and 9 September 2015 (b). The map includes indications of radiation range (see legend), height of measurement site (numbers in black) and AERONET AOD from Sede Boker site (black square) and Weizmann site (black triangle). On 8 September Weizmann AERONET did not operate.

[Figure]

Figure X9. Hourly average of global, direct and diffuse solar radiation between 6-11 September 2015 from Beit Dagan and Beer Sheva.

**Referees Comments:**

Referee #2:  *P7, L183: Now a critical issue: I checked the Kotthaus paper. According to this paper and as it is well known, the ceilometer delivers range-corrected signals in arbitrary units. The measured counts are converted by using a conversion factor to obtain useful signal profiles, when the background is subtracted. We may denote these range corrected signals as level-0 data. Vaisala uses a 'conventional' conversion factor to transfer the background-corrected signals into lidar backscatter signals. But this is NOT the attenuated backscatter coefficient! To obtain the attenuated backscatter coefficient (something like the Rayleigh-calibrated range-corrected signal) you need an actual calibration (to obtain an actual conversion factor). This actual conversion factor however can only be derived under clear sky conditions (so that clear sky backscatter in the free troposphere becomes visible and an accompanying sunphotometer delivers the total OD (aerosol plus water vapor absorption) at around 910 nm, and the aerosol related AOD at 910nm by using interpolation between the measured 870 and 1020nm AOD). At these favorable conditions, the range-corrected signal may be calibrated to deliver the attenuated (aerosol) backscatter profile, and by using the Klett method and adjustment of the lidar ratio (that has to take care of water vapor absorption in addition in case of the Vaisala ceilometer) even the total particle backscatter coefficient. But all this was not possible under these dust storm conditions. In conclusion, you just present range-correct signals in arbitrary units. All the plots have to be changed accordingly. The paper is unacceptable if this important changes are not done.*

**Author's response:** Comments accepted.

**Author's changes in manuscript:** The plots present signal counts. The term "backscatter coefficient" was deleted from the text. We apologize for the mistake. Thank you for the detailed explanation.

**Instruments- Radiosonde:  lines 197-213**

**Referees Comments:**

Referee #1: *"...disclosing the different meteorological conditions...": This is a too short statement: explain, what is shown. Explain the differences that can be seen from Fig. 4. What are the conclusions with respect to the overall topic of the paper?*

Referee #2:  *P7, L198: Please remove the trivial statement about the radiosonde ascents. Please provide just information on radiosonde type and company, and meteorological parameters measured.*

*lines 208 ff: Move all results to Sect. 3.*

**Author's response:** Comments accepted.

**Author's changes in manuscript:** A discussion on radiosondes profiles was moved to the "Results and discussion" section. The radiosonde is a Vaisala type RS41-SG producing profiles of humidity, temperature, pressure, and wind measurement.

**Instruments- Particulate matter measurements: lines 221-247:**

**Referees Comments:**

Referee #1: *Note, that the titles of the subsections are inconsistent: "ceilometers" are instruments, but "PM10" is not.*

**Author's response:** Comment accepted.

**Author's changes in manuscript:** The title of Sect. 2.3 was changed from "PM10" to " Particulate matter monitoring (PM10, PM2.5)". In the text, PM10 was mentioned as the type of measurement and not as the name of the instrument itself.

**Referees Comments:**

Referee #1:

*line 223: Give the manufacturer (and type) of the instruments.*

**Author's response:** The particulate matter measurements were taken from the air monitoring network directed by the Israel ministry for environmental protection. Unfortunately, the ministry does not publicize in their internet site the type of instrument they use. Nevertheless, all monitoring stations were granted ISO17025 meaning the calibration method of measurement is approved by European standards. Based on private communication with the ministry, the manufacturer of the majority of particulate matter instruments is Thermo Fisher Scientific. Two main types are used:  FH 62 C14 (beta attenuation method) and 1405 TEOM (tapered element oscillating microbalance method).

**Author's changes in manuscript:** The instrument type and manufacturer were added to the text.

**Referees Comments:**

Referee #1:

*line 235: If the availability is given for the PM10-monitors it would be consistent to mention the availability of the other instruments as well (certainly a minor point).*

*line 236:" (87% of the monitoring stations" can be omitted.*

**Author's response:** The PM10 continuous measurements are 5 min average based on instantaneous measurements of 1 min. Data is automatically defined "invalid" if among the average time based (e.g.,30 min, 1 hr) more than 25% of the data is missing. Therefore, it was possible to evaluate the availability of the PM10 measurements.  In the other instruments (AERONET, ceilometers), there isn't a defined method to objectively evaluate the availability of the data, thus it was not stated. For instance, AERONET also availability depends on the daylight hours and the degree of AOD (at high AODs the instruments shuts down). Overall, the ceilometers data (not referring to the quality of the data along the profile) was 100% except for the Weizmann (Rehovot) site with 6 hours missing on 7 September (shown in the figure) due to a local power failure. Fig. 6 presented instantaneous data from the AERONET sites, was an attempt to show the extent of the daily availability as it changed from day to day.

**Author's changes in manuscript:** We added comments referring to the availability of the data from each instrument but avoided presenting absolute values. The AERONET figure was changed to daily values shown in Fig X10:

[Figure]

Figure X10. September 2015 daily average of AOD (top panel) and Angström exponent (bottom panel) from two AERONET sites in Israel (Sede Boker and Weizmann). The Weizmann AERONET did not operate on 6-8 September due to power failure.

**Referees Comments:**

Referee #1:

*lines 237/38: This is also a result and should be moved to the corresponding section. The same is true for Fig. 5a. Add a xy-grid to facilitate the interpretation of Fig. 5a. Figs. 5a … also belong to the results.*

Referee #2: *P9, Figure 5 caption: Please clearly state what the yellow curve shows. It took me some time, to see that it belongs to the southern region. The curves show the mean of all sites of a given region? Hard to see the ceilometer stations. Give them a yellow full circle and plot them last (after plotting everything else). Then one should clearly see the sites.*

**Author's response:** Following the comments we decided to present the data from 31 monitoring sites (Table X3) and present the spatial and temporal PM10 dispersion by Fig. X11.

**Author's changes in manuscript:** We replaced the PM10 plots to Table X4 and Fig X11:

Table X4. Hourly maximum concentration of PM10, collected from 31 monitoring sites, between 7-10 September 2015. The values are ranked from low (dark green) to high (dark red) values.

| No. | Site | Height (m ASL) | Region | PM10 ($\mu g\ m^{-3}$) | | | |
|---|---|---|---|---|---|---|---|
| | | | | 7-Sep-15 | 8-Sep-15 | 9-Sep-15 | 10-Sep-15 |
| 1 | Galil Maaravi | 297 | North | 114 | 3130 | 1987 | 1562 |
| 2 | Karmelia | 215 | North | 39 | 1120 | 1008 | 765 |
| 3 | Newe Shaanan | 240 | North | 104 | 3459 | 2471 | 1518 |
| 4 | Haifa Port | 0 | North | 78 | 1600 | 1965 | 1699 |
| 5 | Nesher | 90 | North | 117 | 3265 | 2746 | 1270 |
| 6 | Kiryat Haim | 0 | North | 82 | 1161 | 1625 | 1088 |
| 7 | Afula | 57 | North | 97 | 3239 | 2322 | 1961 |
| 8 | Um El Kotof | 0 | Coast | 99 | 2025 | 2028 | 1630 |
| 9 | Orot Rabin | 0 | Coast | 58 | 1152 | 1455 | 999 |
| 10 | Barta | 0 | Coast | 112 | 2540 | 2345 | 1612 |
| 11 | Qysaria | 19 | Coast | 54 | 1067 | 2116 | 1272 |
| 12 | Rehuvot | 70 | Coast | 88 | 2236 | 3045 | 1257 |
| 13 | Givataim | 0 | Coast | 112 | 1909 | 4014 | 1484 |
| 14 | Yad Avner | 77 | Coast | 61 | 1738 | 2902 | 1252 |
| 15 | Ameil | 20 | Coast | 96 | 2027 | 3472 | 1321 |
| 16 | Shikun Lamed | 17 | Coast | 51 | 1701 | 3244 | 1097 |
| 17 | Station | 29 | Coast | 87 | 1420 | 2176 | 998 |
| 18 | Ashkelon | 29 | Coast | 117 | 953 | 1692 | 551 |
| 19 | Ariel | 546 | Mountain | 128 | 2723 | 1481 | 1358 |
| 20 | Jerusalem Efrata | 770 | Mountain | 273 | 7820 | 1630 | 1437 |
| 21 | Jerusalem Bar Ilan | 749 | Mountain | 181 | 5588 | 1191 | 966 |
| 22 | Jerusalem Safra | 797 | Mountain | 491 | 10280 | 2389 | 1780 |
| 23 | Gush Ezion | 960 | Mountain | 310 | 6230 | 1679 | 1119 |
| 24 | Erez | 80 | South | 44 | 1000 | 1000 | 718 |
| 25 | Beit Shemesh | 350 | South | 115 | 2097 | 1943 | 1788 |
| 26 | Carmy Yosef | 260 | South | 85 | 1047 | 784 | 594 |
| 27 | Modiin | 267 | South | 185 | 2701 | 2245 | 1980 |
| 28 | Bat Hadar | 54 | South | 65 | 1342 | 2563 | 841 |
| 29 | Nir Galim | 0 | South | 94 | 1479 | 2292 | 1027 |
| 30 | Negev Mizrahi | 577 | South | 183 | 9031 | 2806 | 1730 |
| 31 | Eilat | 0 | South | 275 | 1867 | 1592 | 1684 |

[Figure]

Figure X11. A map of PM10 maximum hourly concentration from 9 sites measured at 10 UTC on 8 September 2015. The map includes indications of the time of measurement (symbol shape), concentration range (symbol color), height of measurement site (numbers in black) and AERONET AOD from Sede Boker site.

**Referees Comments:**

Referee #2: *I asked colleagues from Israel to help me with... the following..., I am not familiar with Hebrew language... for example: http://www.svivaaqm.net/. Technically, you have at http://www.svivaaqm.net/ 'reports from numerous sites'... The dust storm: at September 7, around 22:00 values above 100 are already recorded for two sites (cities): Kiriat Ata and Nesher (next to Haifa). Haifa region (Northern part of Israel) must be included for more comprehensive analyses. The authors stated that: "Unfortunately, there are no PM10 measurements in northern Israel".(Line 233) During the dust event there were seven sites that measure PM10 that were available. Figure 5 is not correct therefore, it covers more area, than depicted in Figure 5, including the Haifa Bay area.*

**Author's response:** All PM10 measurements, including the Haifa region, were included in our calculations. Nevertheless, these stations do not cover what we define as the northest region in Israel named "Golan heights" (also referred by Gasch et al.). Measurements from the Golan heights would have given us a perspective whether the dust plume prevalence over Israel on 8 September 2015, had gradually progressed inland from northeast, or rather unified but measured mainly by the elevated monitoring sites since the dust plume hadn't thus far subsided.

**Author's changes in manuscript:** Following the comments we decided to present the data from 31 monitoring sites (Table X3) and present the spatial and temporal PM10 dispersion by Fig. X11.

**Referees Comments:**

Referee #2: *Why did the authors not include PM2.5 sites data.*

**Author's response:** We did not include the PM2.5 measurements although we've already analyzed them) since the intensity of the dust storm was less profound in PM2.5 measurements (see Fig. X13) an with less monitoring sites compared to PM10 measurements.

**Author's changes in manuscript:** We can add a summary (Table X5) of PM2.5 maximum hourly averages from all sites available between 1-10 September.

Table X5. Hourly maximum concentration of PM2.5, collected from 21 monitoring sites, between 7-10 September 2015. The values are ranked from low (dark green) to high (dark red) values.

| No. | Site | Height (m ASL) | Region | PM2.5 ($\mu$g m$^{-3}$) | | | |
|---|---|---|---|---|---|---|---|
| | | | | 7-Sep-15 | 8-Sep-15 | 9-Sep-15 | 10-Sep-15 |
| 1 | Kefar Masarik | 8 | North | 52 | 378 | 389 | 378 |
| 2 | Ahuza | 280 | North | 36 | 743 | 650 | 419 |
| 3 | Newe Shaanan | 240 | North | 43 | 400 | 466 | 525 |
| 4 | Nesher | 90 | North | 43 | 564 | 496 | 349 |
| 5 | Kiryat Biyalic | 25 | North | 53 | 424 | 703 | 447 |
| 6 | Kiryat Binyamin | 5 | North | 40 | 223 | 412 | 256 |
| 7 | Kiryat Tivon | 201 | North | 47 | 413 | 416 | 300 |
| 8 | Afula | 57 | North | 44 | 836 | 550 | 405 |
| 9 | Raanana | 54 | Coast | 38 | 173 | 291 | 229 |
| 10 | Antolonsky | 34 | Coast | 32 | 470 | 626 | 386 |
| 11 | Ashdod | 25 | Coast | 36 | 303 | 750 | 332 |
| 12 | Ironi D | 12 | Coast | 34 | 424 | 507 | 327 |
| 13 | Tel aviv Central Station | 29 | Coast | 41 | 716 | 803 | 451 |
| 14 | Ashkelon | 25 | Coast | 61 | 182 | 537 | 119 |
| 15 | Jerusalem Efrata | 749 | Mountain | 106 | 2285 | 434 | 403 |
| 16 | Jerusalem Bar Ilan | 770 | Mountain | 107 | 3063 | 641 | 518 |
| 17 | Gedera | 70 | South | 34 | 433 | 683 | 308 |
| 18 | Nir Israel | 30 | South | 25 | 363 | 638 | 228 |
| 19 | Kiryat Gvaram | 95 | South | 42 | 376 | 870 | 300 |
| 20 | Sede Yoav | 105 | South | 45 | 323 | 245 | 228 |
| 21 | Negev Mizrahi | 577 | South | 42 | 1748 | 526 | 317 |

**Instruments- AERONET : lines 254-273**

**Referees Comments:**
Referee #1: "...within a 1 min period": maybe it is better to remove this. From this statement the measurement cycle is not clear: 8 s measurements, 22 s break, 8 s measurements, 22 s break, 8 s measurements? This is more than 1 minute.
**Author's response:** Comment accepted.
**Author's changes in manuscript:** The sentence was omitted.

**Referees Comments:**
Referee #1: Figs.. 6 also belong to the results.
line 262: As this is a result it should be moved to Sect. 3. Fig. 6: to be moved to Sect. 3.
**Author's response:** Comments accepted.
**Author's changes in manuscript:** AERONET figure was moved to the "Results and discussion" section.

**Referees Comments:**
Referee #1: I don't understand the x-axis of Fig.6 ? Why is the length of the days different? Where are the night measurements of AERONET from (do you deploy a lunar photometer?)? Is there a specific reason for not choosing a line plot in Fig. 6a (as in Fig. 6b)?
Referee #2: P10, Figure 6: the time axis ... the day scale (width) is changing from day to day. E.g., the 8 September is very narrow, the 7 September has a factor of 4 more space..., why?
**Author's response:** All the data available from the Israeli AERONET sites was delineated in Fig. 6. The length of the days is different due to missing data. Fig. 6 was an attempt to present the availability of the AODs as the extent of available data changed from day to day. We found the presentation of Fig. 6 more informative than a line plot.
**Author's changes in manuscript:** The AERONET figure was changed to daily values shown in the aforementioned Fig X10.

**Referees Comments:**
Referee #2: P10, L264. The Rehovot station did not work, the instrument was out of order? .... or did the station not allow useful data analysis because the AOD was too high? Please clarify and state what was the case...
**Author's response:** The instrument was out of order the due to power failure in the Weizmann institute.
**Author's changes in manuscript:** We mentioned the instrument was out of order.

**Results: lines 276-472**

**Referees Comments:**

Referee #1: *In the "results" -section the discussion should be more elaborated and linked to the findings from the other data sets (provided in part when the MLH is discussed). The discussion of the scientific results must be improved and extended. In the present state the paper is sort of a collection of (useful) pieces of information, but their relationship and their interpretation is not sufficiently elaborated.*

**Author's response:** Comments accepted.

**Author's changes in manuscript:** The "Results" paragraph was rewritten according to the overall comments of the referees.

**Referees Comments:**

Referee #1: *From Fig. 6a it seems that after 11. September 2015 the AOD was still relatively large (by the way: can you give the annual mean AOD of the sites in Israel?). How "typical" are the ceilometer profiles between 11. and 14. September 2015?*

**Author's response:** Annual mean of 2015 of AOD from Sede-Boker : 0.22 . AERONET Weizmann started operating on 6 June 2015, thus the half year mean is: 0.25. Although Sede Boker is situated in the southern Negev desert of Israel, the Average AOD of the dusty and windy season (Oct- Nov) remained 0.25. Therefore, the AODs above 3 are rare. Between 11-14 September 2015, AODs are large although the PM10 measurements and ceilometer signals infer differently since they do not include the whole air column.  This finding may justify the results of Stavros et al. (2016) and the CALIPSO measurements on the 10 September of an elevated layer of the dust plume at about 3-4 km.

**Author's changes in manuscript:** We added a discussion on the comparison between measurements (AERONET, particulate matter, global radiation, ceilometers, Satellite pictures) after 10 September 2015.

**Referees Comments:**

Referee #1: *line 281 ff: Move lines 281–283 to Sect. 2.*

**Author's response:** Comment accepted.

**Author's changes in manuscript:** The sentences were moved to the ceilometer discussion in the "Instruments" section.

**Referees Comments:**

Referee #1: *The absolute values of the attenuated backscatter must be checked: some of the numbers are unrealistic and the corresponding figures are not clear. For example, the labels of Fig. 7c (e.g. 0.000029 in units of $10^{-9}$ m$^{-1}$ sr$^{-1}$) are confusing. lines 281–283: Check carefully the following numbers of the attenuated backscatter; some of them are unrealistic ($7 \cdot 10^{-1}$ m$^{-1}$ sr$^{-1}$).*

*lines 281–283: See also general remarks at the beginning. Add a xy-grid to Fig. 7.Figs. 8–16: What is the unit of the color code? It does not agree with the values given for attenuated backscatter. Are these numbers the "counts"?*

Referee #2: *P12, Figure 7, We need a clear statement, that the range-corrected signals shown in Figure 7 decrease rapidly and is close zero at about 700 to 750m in b,c,d because of the strong laser light attenuation in the dust layer! As long as such a statement is missing the reader may believe to see the full dust layer and the top is at 750-1000m height. To repeat: This is unacceptable. This is simply wrong and unacceptable. Please improve! The basic ceilometer data are signals (let us say ... in units... counts per second), and if they are then range corrected... then you get the dimension 'counts per second times m\*\*2'. So the values are not unitless, but usually given in arbitrary units. Next, by dividing these data by 10ˆ(-9) ... does not change anything. You still have just range-corrected signals. You can only obtain a profile of the attenuated backscatter coefficient if you are able to calibrate this range-corrected signal profile in the tropospheric region with pure Rayleigh backscattering or in the way as described above. So, you show range corrected signals!!! And not attenuated backscatter!!! As mentioned already, you must change .... to range corrected signals in all ceilometer plots!*

**Author's response:** A fundamental mistake. The attenuated backscatter presented in Fig. 7 had already been divided by $10^9$, therefore, $10^{-9}$ should have been omitted from the x-axis label. The numbers shown in the ceilometer plots are signal counts**.**

**Author's changes in manuscript:** Ceilometer profiles were correct to units of attenuated backscatter in the order of $10^{-5}$ $m^{-1}$ $sr^{-1}$. the title "Signal counts" was added to the legend in all ceilometer plots.

**Referees Comments:**

Referee #1: *The authors should be aware that a comparison of brown vs. brown color or brown vs. blue color (Figs. 10–16) is not suitable for a scientific publication. Please use quantitative numbers! Think about plotting coincident attenuated backscatter profiles of the 8 sites (similar to Fig. 7), this may help to see temporal delays in the arrival and decay of the plume.*

Referee #2: *P12. Figure 8,x-axis please show data always from 0-24 local time (or UTC). Again we need proper text for the x-axis and y-axis, as it is the case in Figure 7. P12-15: All the Figures 8,10-16 have no x-axis and y-axis description. This is poor and unacceptable. And again, all these ceilometer color plots suggest that the dust layer was just a few hundred meter thick. This is dangerous! The reason is simply the almost total attenuation of the ceilometer radiation pulses by the rather dense dust layers. This must be made very clear.*

*line 320: "the time corrected from local time to" can be removed.*

**Author's response:** After several trials we found the current plot contours are the most informative. However, supplemented plots of attenuated backscatter profiles from all ceilometers at specific hours given in Fig. X12.

**Author's changes in manuscript:** Supplement figure X12:

[Figure]

Figure X12. Ceilometer attenuated backscatter profiles from 7 sites (Ramat David, Hadera, Tel Aviv, Beit Dagan, Weizmann, Nevatim and Hazerim, Fig. 3) at 23 UTC 7 Sep 2015 (a), 8 September 2015 at 12 UTC (b), 9 September 2015 at 16 UTC (c) and 10 September 2015 at 14 UTC (d). Notice each profile begins relative to the height of its' measuring site (ASL) including a deletion of the first 100 m AGL due to inaccuracies in the first range gates of the CL31 ceilometers (for details see Sect. 2.1). Fig (a) shows cloud detection therefore it has a different scale ($10^{-1}$ $m^{-1}$ $sr^{-1}$) and a different x-axis range.

**Referees Comments:**

Referee #2: *P13, L324: again and again: you were able to track the dust layer base only, this must be clearly said. In Figure 8, the numbers for 'your' attenuated backscatter are suddenly up to 15000, compared to values of about 10^(-14) in Fig 7 (b,c,d)? Then in Figs 10-12: up to 10000. And in Fig 13, suddenly only up to 800..., Fig. 14 up to 15000, and Fig15-16 up to 10000. So all this is rather strange...and only reasonable and understandable if we switched to range corrected signals (arbitrary units). So, please change..... to range corrected signals.*

P13, L347: *Plots are given in different scales to highlight the dust features. This is ok, because range corrected signals are shown and the ceilometer performance changes from site to site (from ceilometer to ceilometer). So again, there is a clear need to work with range corrected signals.*

**Author's response:** 6 Out of the 8 of the ceilometers belong to a governmental office. We were not informed whether the "message profile noise h2" was off or on therefore whether an automated range correction was done.

**Author's changes in manuscript:** No changes.

**Referees Comments:**

Referee #1: *lines 328 ff: Obviously attenuated backscatter is converted into particle extinction coefficient (with a relatively large uncertainty inherent in all CL31 measurements; see water vapor absorption mentioned above, unknown lidar ratio, unknown accuracy of the scaling factor) with an estimated lidar ratio. Then, the visibility is estimated according to Koschmider. Which altitude was selected for this conversion? The problem is, that if it is done for a large altitude (maybe 100 m or more) it is difficult to compare this visibility to independent ground based measurements. If it is done for the ground, then the overlap problem is critical. Nevertheless, an order of magnitude agreement should be possible, but please extend this paragraph by explaining all aspects of this comparison.*

Referee #2: *P13, L330: A visibility of 200m (visual meteorological range is defined by an AOD of 3, after Koschmieder for an AOD of 4) according to an AOD of 3 means that the particle extinction coefficient was 15 km-1 or 0.015 m-1 and the backscatter coefficient is then 0.0003 m-1 sr-1 if the dust lidar ratio is 50sr. All your 'numbers' are far far away from these value. This corroborates: It is impossible and dangerous (and thus not justified) to convert the range-corrected signals into optical properties just by taking 'some' conversion factor!*

**Author's response:** We took the visibility observations under 100 m AGL and compared to the attenuated backscatter received at ~500 m AGL. We understand it is misleading and therefore we decided to omit this paragraph.

**Author's changes in manuscript:** The paragraph was deleted.

**Referees Comments:**

Referee #1: *lines 334 ff: "Ceilometers are not provided with an AOD...": I don't understand this paragraph. AOD and MLH are mentioned but it is unclear what the message is. The retrieval of the MLH from ceilometer data is completely different from a retrieval of the AOD (provided that can be determined at all). So, how is the validation ("verify ceilometer") of the ceilometer's performance achieved? Please rephrase and extend this part.*

Referee #2: *P13, L334-L337. I would remove this text on the ceilometer and the AOD upper limit. This is useless. The Vaisala ceilometers are not built for aerosol profiling. The wavelength is bad, the signals are corrupted by water vapor absorption.*

**Author's response:** We intended to explain that we do not know the detection limit of the attenuated backscatter signals at high AODs. Therefore, we cannot put a direct limit to the ceilometer threshold. Instead we tried to use other methods to verify the first layer height. In this subsection we attempted to retrieve the inversion height from the ceilometers and radiosonde. Since aerosols are usually trapped under the inversion height and this height is similar in the radiosonde and WCT calculation as a tool to define the height of the first dust layer.

**Author's changes in manuscript:** We rephrased the sentences.

**Referees Comments:**

Referee #2: *P19, L490: Again, you have to mention the limits of a ceilometer. It was too weak to see the layers higher up. No chance to see the main part of the dust layers and dust layer top.*

**Author's response:** Comment accepted.

**Author's changes in manuscript:** We have listed the ceilometers limits in the Instruments section and referred to these limits in the results.

**Referees Comments:**

Referee #2: *P13, Fig. 9 shows the dust base height. To my opining it is misleading to denote the near-surface layer a 'mixed layer' at these conditions with no vertical exchange P13: The text on this page is poor and needs to be significantly improved.*

**Author's response:** We accept the referee's opinion on the 8 September since the global radiation (Fig. X2) was marginal for a significant thermal creation.

**Author's changes in manuscript:** The "Results and discussion" paragraph was rewritten according to the overall comments of the referees.

**Referees Comments:**

Referee #1: *line 381 ff: A short paragraph should be included here to prepare the reader: The event is divided into several phases (not necessarily split into single days, one phase can be shorter, another can last for more than one day; development, main phase, decay can be an alternative) according to certain criteria and to highlight consistencies/inconsistencies of different data sets/models (by doing this "Next, we describe the decrease of aerosols aloft on mid-day" in line 299 can be omitted). Then, each subsection such as" Entrance of dust into Israel – 7 Sep 2015" should include the full discussion and interpretation of all available data sets for the corresponding period, i.e., parts from Sect. 2 should be included here whenever applicable.*

**Author's response:** Comments accepted.

**Author's changes in manuscript:** We changed the results section and described the dust storm evolution along it's generation rather than by dates.

**Referees Comments:**

Referee #2: *P15, L383: Please clearly state where the dust layer top was found by Solomos et al. (2017).*

**Author's response:** Stavros et al (not Solomos, my mistake) analyzed by CALIPSO overpass two layers, 2 km (total attenuation was reported up to 1.5 km) and between 3-4 km.

**Author's changes in manuscript**: The summary of Stavros et al. was extended.

**Referees Comments:**

Referee #2: *"The AERONET (Fig. 6) and ceilometer plots (Fig. 8, 11-16) reveal that the first dust plume penetrated Israel at approximately 04:00 UTC". What day? Sept. 7 or Sept 8?*

**Author's response:** *7* September 2015.

**Author's changes in manuscript:** The "Results and discussion" paragraph was rewritten according to the overall comments of the referees.

**Referees Comments:**

Referee #1: *line 401: What is meant by "decrease"? Concentration or altitude?*

**Author's response:** We meant the subsidence of the dust plume.

**Author's changes in manuscript:** The "Results and discussion" paragraph was rewritten according to the overall comments of the referees.

**Referees Comments:**

Referee #1: line 405:" clearly shown in Fig. 13-16 between 08-16 UTC)": This is indeed hard to see. Can you explain this in a more quantitative way?

**Author's response:** We based our conclusions on the wind direction profiles from radiosonde launches in Beit Dagan. Apart the radiosonde, we do not have auxiliary measurements to prove our assumption. Therefore, if the referee could not relate to our conclusions, we decided to omit this hypothesis.

**Author's changes in manuscript:** The sentence was erased.

**Referees Comments:**

Referee #1: *line 408: Please clarify what" these model findings" are? The vertical distribution of dust (backscatter)?*

**Author's response:** We were referring to the hydraulic jump upstream the dead sea rift valley explained by Gasch et al., as the dust plume entered from northeast and progressed southwest. In contrary, the ceilometers plots revealed the penetration of the dust plume occurred at once from north (Ramat David site) to south (Hazerim and Nevatim sites) and to west (Hadera, Tel Aviv, Beit Dagan and Weizmann sites).

**Author's changes in manuscript:** The "Results and discussion" paragraph was rewritten according to the overall comments of the referees.

**Referees Comments:**

Referee #1: *line 422: 2000 μg/m3: is this in contradiction to 9800 μg/m3 in line 416*?

**Author's response:** We were referring to the second "jump" in PM10 measurements which occurred simultaneously in all monitoring stations on the 8 September at~17:00 (UTC+2) after the extreme values of 9800 μg/m3 were measured only in the high altitude stations on the 8 September at ~ 12:00 (UTC +2).

**Author's changes in manuscript:** We rephrased the sentence in the scope of the "Results and discussion" section.

**Referees Comments:**

Referee #1: *line 437: "...limited radiative transmitted...": what does this mean?*

**Author's response:** We meant the decrease in the global solar radiation.

**Author's changes in manuscript:** The sentence was rephrased.

**Referees Comments:**

Referee #2: *P17, L435: At very high optical depth as on 9 Sep, I would assume that convective motions in the PBL as well as a sea breeze winds cannot develop. Are you sure that sea breeze developments were possible at these days with almost no sun and differential sea/land heating? Please keep the discussion free of speculation.*

**Author's response:** On the 8 September, the maximum wind speed was below 3m/s and maximum global solar radiation up to 264 W m-2. The ability to generation of thermals under these would be rather weak. on the next day, on the 9 of September the maximum global radiation more than doubled reaching 621 (w m-2) and wind speed increased to a maximum of 5 m s-1 at 13:00 (UTC+2). At these conditions, convection and creation of thermals is possible.

**Author's changes in manuscript:** We've added an explanation regarding the possibility of thermals creation on the 9 September.

**Referees Comments:**

Referee #1: *Fig. 17: This figure is misleading as the range of the color code is different from Fig. 14. This should be pointed out clearly. As long as the inter-comparison with the AOD (AERONET) is qualitative only, it would be helpful to show the typical(?) background(?) values of the AOD and attenuated backscatter of the days before the event for comparison.*

Referee #2: P17.... Figure 17 indicates that there was dust higher up. The AOD decreased towards 0.5-1 on 12-14 Sep. A perfect mixing layer could develop now up to 750 m, as seen on 13 and 14 Sep. Nice to see, that the aerosol dried in the PBL during the morning hours and thus the color of the range corrected signals changed from red to green and blue (for dry particles producing less backscatter later on).

**Author's response:** In order to visualize the maximum information from each ceilometer plot, the plot scale varied from site to site in reference to the ceilometer type (CL51, CL31) and installation definitions (e.g. ceilometer Beit Dagan produces significantly less signal counts for the same phenomena).

**Author's changes in manuscript**: New ceilometer plots with a unified scale of 0-10,000 signal counts (Fig. X13-X18) except for the CL51 Weizmann ceilometer plots ( 0-15,000 signal counts, Fig. X19)  and Beit Dagan ceilometer (0-800 signal counts, Fig. X20).

[Figure]

Figure X13. Ramat David ceilometer signal counts plots for 7-9 September 2015. Y-axis is the height up to 2000 m ASL, X-axis is the time in UTC, signal counts scale range between 0-10,000.

[Figure]

Figure X14. Hadera ceilometer signal counts plots for 7-9 September 2015. Y-axis is the height from site deployment to 2000 m ASL, X-axis is the time in LST (UTC+2), signal counts scale range between 0-10,000.

[Figure]

Figure X15. Tel Aviv ceilometer signal counts plots for 7-9 September 2015. Y-axis is the height from site deployment to 2000 m ASL, X-axis is the time in UTC, signal counts scale range between 0-10,000.

[Figure]

Figure X16. Nevatim ceilometer signal counts plots for 7-9 September 2015. Y-axis is the height from site deployment to 2000 m ASL, X-axis is the time in UTC, signal counts scale range between 0-10,000.

[Figure]

Figure X17. Hazerim ceilometer signal counts plots for 7-9 September 2015. Y-axis is the height from site deployment to 2000 m ASL, X-axis is the time in UTC, signal counts scale range between 0-10,000.

[Figure]

Figure X18. Mount Meron ceilometer signal counts plots for 7-9 September 2015. Y-axis is the height from site deployment to 2000 m ASL, X-axis is the time in UTC, signal counts scale range between 0-10,000.

[Figure]

Figure X19. Weizmann ceilometer signal counts plots for 7-9 September 2015. Y-axis is the height from site deployment to 2000 m ASL, X-axis is in UTC, signal counts scale range between 0-15,000.

[Figure]

Figure X20. Beit Dagan ceilometer signal counts plots for 7-9 September 2015. Y-axis is the height from site deployment to 2000 m ASL, X-axis is in LST (UTC+2), signal counts scale range between 0-800.

**Referees Comments:**

Referee #2: *P17, L444: You state: The ceilometer reveals total clearance on 10 Sep! But the Weizmann Institute AERONET shows AODs of 2 and more on 10 Sep! What is wrong, what is true? Please clarify?*

*P18, L461: The AOD was >1.0 all the time on 9 and 10 Sep...until 12 Sep. What do you thus mean with dissipation of dust?*

*P19, L498-499: When were the AE values high again? They were continuously <0.5 even on 14 Sep (Weizmann AERONET).*

**Author's response:** The "clearance" we mentioned was referred to the lower part of the atmosphere (up to ~1 km), as the PM10 values decreased considerably and the amount of signal counts from all 8 ceilometers declined. On 10 September, the dust plume motion continued southwest to Egypt, with indication of a dust layer between 2-4 km measured by CALIPSO. The progression of the dust storm southwest over Israel, from the Syria-Iraqi border to Egypt, continued in two separated levels. The lower level (up to 1 km ASL) dissipated at 13 September while the level aloft (above 1 km ASL) was observed until 17 September.

**Author's changes in manuscript:** Supplementary results and a discussion on the dust dissipation.

**Referees Comments:**

Referee #1: *line 454: 250 m: is this the vertical extent or the altitude?*

Referee #2: *P18, L454: ...as a dust layer of 250 m thickness (fig 11-13, 15-16) penetrated Israel at a height of 1000-1500m.... How do you know the depth of the dust layer? The ceilometer fails to see higher up.... So, how do you know? I would leave out to mention any dust layer depth.*

**Author's response:** Comments accepted.

**Author's changes in manuscript:** Considering the ceilometers limitations to detect attenuated backscatter signals above a dense dust layer, we omitted the assumptions regarding the vertical extent of the dust layer.

**Referees Comments:**

Referee #1: *line 486: The PM10 measurements are considered as in-situ measurements, not remote sensing.*

**Author's response:** Comments accepted.

**Author's changes in manuscript:** We corrected the sentence referring to PM10 measurements as in situ.

**Conclusions and discussion: lines 480-514**

**Referees Comments:**
Referee #2: P19: The conclusions have to be rewritten completely after improving all the text before along the lines this review and the other review.
**Author's response:** Comment accepted.
**Author's changes in manuscript:** We rephrased the "Conclusions " section according to the referees' comments.

**Referees Comments:**
Referee #2: What sources of errors do they have when using the ceilometers? They should critically state the limitations, disadvantages and advantages. Including comparison between different ceilometers that authors used for the analyses. Without this comprehensive critical discussion on authors findings, the outcome of the paper is doubtful.
**Author's response:** Comments accepted.
**Author's changes in manuscript:** We listed the ceilometers limitations in the Instruments sections. In the discussion section we referred to these limitations in the as part of the process of evaluation.

**Referees Comments:**
Referee #2: *Figure 18: The major claim - the dust penetrates from the East. But- combining PM10 from the Haifa Bay area, there is a "jump" towards values of 2500-3000 micrograms/m3 at 8 of September, similarly to East, which means it has two entrances/sources. Do the authors see the "North region" dust entrance using ceilometer data? The authors must justify what new information they get using ceilometer more clearly than in Figure 18, what new insights they get about the extreme dust event? And number/summarize all "new insights" about the event that they discover*
**Author's response:** We analyzed PM10 and PM2.5 measurements from all available monitoring stations in Israel. Unfortunately, we did not recognize a second jump from northern region. To emphasize our conclusions, we prepared Fig. X21 presenting PM10 values from monitoring sites the referee mentioned: Haifa Bay, and the northest station in Israel (Karmiel).  We added three more stations: Hadera (central shoreline), (Jerusalem (mountains) and Arad (south). To our opinion, it is difficult to define a second "jump" on the 8 September exclusively in the northern sites.

[Figure]

Fig. X21. (a) Map of PM10 monitoring sites (Karmiel, Haifa Bay ,Hadera, Jerusalem, Arad) (b) 3 hourly average values of PM10 measured at each site during 7-10 September 2015.

**Author's changes in manuscript:** Figure 18 was replaced by additional solar radiation figures and a discussion on the contribution of the ceilometers to understand the evolution of the dust storm in the lower atmosphere (under 1 km ASL).

**Referees Comments:**
Referee #1: *line 488:" ...for the first time, such an event is vertically analyzed using an array of ceilometers...". On the one hand this is true, on the other hand it is slightly misleading as the vertical structure (by other means) has already been investigated. So it might be advisable to use a less strong statement in the next sentence (a note on the limited measurement range).*
Referee #2: *P19, L488: for the first time such an event is vertically analyzed...... this is misleading because Mamouri et al. already used lidar to characterize the dust storm. You probably wanted to say, for the first time .... with a ceilometer network. However, you should mention that there were already lidar studies with Cyprus lidar and CALIOP lidar, and now you come with a ceilometer network study...... Then this would be more clear, and of course this is a new aspect.*
**Author's response:** Comments accepted.
**Author's changes in manuscript:** We rephrased the sentence and emphasized the contribution of the ceilometer measurements to the analysis of the lower part of the atmosphere (from ground level up to ~1 km) as a completion to previous studies concentrating on the generation and propagation of the dust plume down to ~1.5 km.

**Referees Comments:**
Referee #1: lines 492, 497:" plume"!
**Author's response:** Comment accepted.

**Author's changes in manuscript:** Typing mistakes were corrected.

**Referees Comments:**

Referee #1: *line 494:" mainly of mineral dust": where is this information coming from?*

**Author's response:** We based our conclusions on Sede-Boker AERONET Angstöm exponent (Fig. X10)**.** Mamouri et al., (2016) studied the dust layer particle linear depolarization by an EARLINET lidar stationed in Limassol Cyprus. They concluded the linear depolarization ratio of 0.25-0.32 on 7, 10 September, indicated the dominance of mineral dust (the lidar was inoperative on 8 September).

**Author's changes in manuscript:** We added the citation to the AERONET Angstöm exponent measurements in Israel and referred to the conclusions from Mamouri et al., (2016).

**Referees Comments:**

Referee #2: *P19, L494: As a result, ...... of what?*

**Author's response:** As a result of the low boundary layer.

**Author's changes in manuscript:** We rephrased the "Conclusions and discussion" section according aforementioned overall comments of the referees.

**Referees Comments:**

Referee #2: *P19, L502-504: This is speculation, at least to my opinion. Be more save with your statements.*

**Author's response:** Comment accepted.

**Author's changes in manuscript:** The sentence was omitted and a comprehensive analysis of the meteorological measurements (global radiation, direct radiation, diffused radiation, ground temperature, wind speed) and environmental measurements (PM10, PM2.5, TSP) were added in the attempt to explain and reveal the meteorological conditions held as the dust storm prevailed in Israel.

**Referees Comments:**

Referee #2: P19, L506-511: Again, dangerous statements. I would remove. Otherwise, you need to check the CALIPSO overflight over Israel to corroborate your speculative suggestions. However, the modeling papers of Solomos et al and Gasch et al. (partly based on model plus CALIOP results) do not leave room for statements like ... who knows to what height the dust plumes reached over Israel. To my opinion, in the Middle East dust layer top was up to 4-5 km height everywhere.

**Author's response:** We accept the comment. Additional data from the CALIPSO passage over Israel on the 10 September (given here in Fig. X) indeed shows a dust plume between ~2.5-4.5 km.

**Author's changes in manuscript:** We omitted the statement and refereed to the ceilometers data in the context of the evolution of the dust plume at the lowest level of the troposphere. We stated the ceilometers' limited ability to detect attenuated backscatter signals above the dust layer detected at ~ 1 km.

**References: lines 546-637**

**Referees Comments:**
Referee #2: *P21, L554: No authors.*
**Author's response:** The reference is a report edited and distributed by the U.S Environmental protection Agency with no specific authors mentioned upon the report.
**Author's changes in manuscript:** No change.

**Referees Comments:**
Referee #2: *P23, L621: TOASJ...?*
**Author's response:** Acronym for The open atmospheric science journal.
**Author's changes in manuscript:** The acronym was converted to the full name of the journal.

**A list of relevant changes made in the manuscript:**

1. Updated abstract
2. Updated introduction (Sect. 1) including an expanded review of previous studies.
3. Updated "Instruments" section (Sect. 2) including new instrument (solar radiation and satellite imagery).
4. The structure of the "Results and discussion" section had been changed. Each day (7-10 September 2015) had been analyzed by the same set of tools (instruments).
5. Updated conclusions.
6. Previous figures have been either deleted or updated.
7. Supplement of new figures and tables.

[revised manuscript text omitted]

Formatted Table

Formatted Table
Formatted Table
Deleted Cells
Formatted Table

| Storm over Middle East.severe dust event in the Eastern Mediterranean | cold-pool outflows producing the dust storm. Model lacked development of a super critical flow to produce excessive wind speedsTwo dust storms simultaneously, from northern Syria and Sinai desert created by two low pressure systems. |

Table 2. Ceilometer CL31Ceilometers locations

| Location | Site | Long/Lat | Distance from shoreline (km) | Height (m AGL) |
|---|---|---|---|---|
| Mount Meron | Northern | 33.0/35.4 | 31 | 1,150 |
| Ramat David | Northern | 32.7/35.2 | 24 | 50 |
| Hadera | Onshore | 32.5/34.9 | 3.5 | 10 |
| Tel Aviv | Onshore | 32.1/34.8 | 0.05 | 5 |
| Beit Dagan | Inland | 32.0/34.8 | 7.5 | 33 |
| Weizmann | Inland | 31.9/34.8 | 11.5 | 60 |
| Nevatim | Southern | 31.2/34.9 | 44 | 400 |
| Hazerim | Southern | 31.2/34.7 | 70 | 200 |

*Ceilometer Weizmann is a CL51

Table 3. Some detail on Israel and Cyprus radiosonde detailCeilometers configurations

| Site Location | # Station | Lon/Lat Type | Time Launching time (UTC)  *Height  range resolution(sec) | Height (m ASL) resolution  (m) |  |
|---|---|---|---|---|---|
| Israel | 40179 | 34.8/32.0 , 12:00 (km) | 35 | 00: |  |
| Mount Meron | CL31 | 16 | 10 | 7. |  |
| Ramat David | CL31 | 16 | 10 | 7. |  |
| Hadera | CL31 | 16 | 10 | 7. |  |
| Tel Aviv | CL31 | 16 | 10 | 7. |  |
| Beit Dagan | CL31 | 15 | 10 | 7.7 |  |
|  | 17607CL51 | 33.4/35.2 , 12:00 | 161 | 06: 16 |  |
| CyprusWeizmann | 15.4 | | | |  |
| Nevatim | CL31 | 16 | 10 | 7.7 |  |

| Hazerim | CL31 | | 16 | 10 | 7. |

* Height range dependents on sky conditions and is limited as AOD increases.

* In all ceilometers but in Beit Dagan site, data acquisition was limited to 4.5 km based on the BLview firmware

Table 4.  Ceilometer technical information

|  Location |  board |  Receiver |  Transmitter | Type Firmware | Engine |
|---|---|---|---|---|---|
| Beit Dagan |  | CLE311 | CLR311 | CLT311 | 1.72 |
| |  | | | | |
| |  CL2 | | | | |
|  |  |  |  | | |
|  |  |  |  | | |
|  |  |  |  | | |
|  Weizmann |  CL51 |  |  | | CLE3 |
| | CLRE321 | CLT521 | 1.03 | | |

Table 5. Hourly maximum concentration of PM2.5, collected from 21 monitoring sites, between 7-10 September 2015. The values are ranked from low (dark green) to high (dark red) values.

| | | | | PM2.5 ($\mu g\ m^{-3}$) | | | |
|---|---|---|---|---|---|---|---|
| No. | Site | Height (m ASL) | Region | 7-Sep-15 | 8-Sep-15 | 9-Sep-15 | 10-Sep-15 |
| 1 | Kefar Masarik | 8 | North | 52 | 378 | 389 | 378 |
| 2 | Ahuza | 280 | North | 36 | 743 | 650 | 419 |
| 3 | Newe Shaanan | 240 | North | 43 | 400 | 466 | 525 |
| 4 | Nesher | 90 | North | 43 | 564 | 496 | 349 |
| 5 | Kiryat Biyalic | 25 | North | 53 | 424 | 703 | 447 |
| 6 | Kiryat Binyamin | 5 | North | 40 | 223 | 412 | 256 |
| 7 | Kiryat Tivon | 201 | North | 47 | 413 | 416 | 300 |
| 8 | Afula | 57 | North | 44 | 836 | 550 | 405 |
| 9 | Raanana | 54 | Coast | 38 | 173 | 291 | 229 |
| 10 | Antolonsky | 34 | Coast | 32 | 470 | 626 | 386 |
| 11 | Ashdod | 25 | Coast | 36 | 303 | 750 | 332 |
| 12 | Ironi D | 12 | Coast | 34 | 424 | 507 | 327 |
| 13 | Tel aviv Central Station | 29 | Coast | 41 | 716 | 803 | 451 |
| 14 | Ashkelon | 25 | Coast | 61 | 182 | 537 | 119 |
| 15 | Jerusalem Efrata | 749 | Mountain | 106 | 2285 | 434 | 403 |
| 16 | Jerusalem Bar Ilan | 770 | Mountain | 107 | 3063 | 641 | 518 |
| 17 | Gedera | 70 | South | 34 | 433 | 683 | 308 |
| 18 | Nir Israel | 30 | South | 25 | 363 | 638 | 228 |
| 19 | Kiryat Gvaram | 95 | South | 42 | 376 | 870 | 300 |
| 20 | Sede Yoav | 105 | South | 45 | 323 | 245 | 228 |
| 21 | Negev Mizrahi | 577 | South | 42 | 1748 | 526 | 317 |

Table 6. Hourly maximum concentration of PM10, collected from 31 monitoring sites, between 7-10
September 2015. The values are ranked from low (dark green) to high (dark red) values.

| No. | Site | Height (m ASL) | Region | PM10 ($\mu g\ m^{-3}$) | | | |
|---|---|---|---|---|---|---|---|
| | | | | 7-Sep-15 | 8-Sep-15 | 9-Sep-15 | 10-Sep-15 |
| 1 | Galil Maaravi | 297 | North | 114 | 3130 | 1987 | 1562 |
| 2 | Karmelia | 215 | North | 39 | 1120 | 1008 | 765 |
| 3 | Newe Shaanan | 240 | North | 104 | 3459 | 2471 | 1518 |
| 4 | Haifa Port | 0 | North | 78 | 1600 | 1965 | 1699 |
| 5 | Nesher | 90 | North | 117 | 3265 | 2746 | 1270 |
| 6 | Kiryat Haim | 0 | North | 82 | 1161 | 1625 | 1088 |
| 7 | Afula | 57 | North | 97 | 3239 | 2322 | 1961 |
| 8 | Um El Kotof | 0 | Coast | 99 | 2025 | 2028 | 1630 |
| 9 | Orot Rabin | 0 | Coast | 58 | 1152 | 1455 | 999 |
| 10 | Barta | 0 | Coast | 112 | 2540 | 2345 | 1612 |
| 11 | Qysaria | 19 | Coast | 54 | 1067 | 2116 | 1272 |
| 12 | Rehuvot | 70 | Coast | 88 | 2236 | 3045 | 1257 |
| 13 | Givataim | 0 | Coast | 112 | 1909 | 4014 | 1484 |
| 14 | Yad Avner | 77 | Coast | 61 | 1738 | 2902 | 1252 |
| 15 | Ameil | 20 | Coast | 96 | 2027 | 3472 | 1321 |
| 16 | Shikun Lamed | 17 | Coast | 51 | 1701 | 3244 | 1097 |
| 17 | Station | 29 | Coast | 87 | 1420 | 2176 | 998 |
| 18 | Ashkelon | 29 | Coast | 117 | 953 | 1692 | 551 |
| 19 | Ariel | 546 | Mountain | 128 | 2723 | 1481 | 1358 |
| 20 | Jerusalem Efrata | 770 | Mountain | 273 | 7820 | 1630 | 1437 |
| 21 | Jerusalem Bar Ilan | 749 | Mountain | 181 | 5588 | 1191 | 966 |
| 22 | Jerusalem Safra | 797 | Mountain | 491 | 10280 | 2389 | 1780 |
| 23 | Gush Ezion | 960 | Mountain | 310 | 6230 | 1679 | 1119 |
| 24 | Erez | 80 | South | 44 | 1000 | 1000 | 718 |
| 25 | Beit Shemesh | 350 | South | 115 | 2097 | 1943 | 1788 |
| 26 | Carmy Yosef | 260 | South | 85 | 1047 | 784 | 594 |
| 27 | Modiin | 267 | South | 185 | 2701 | 2245 | 1980 |
| 28 | Bat Hadar | 54 | South | 65 | 1342 | 2563 | 841 |
| 29 | Nir Galim | 0 | South | 94 | 1479 | 2292 | 1027 |
| 30 | Negev Mizrahi | 577 | South | 183 | 9031 | 2806 | 1730 |
| 31 | Eilat | 0 | South | 275 | 1867 | 1592 | 1684 |

---

## Referee Report (RR1)

**New insights into the vertical structure of the September 2015 dust storm employing 8 ceilometers over Israel (revised version)**

**by Leenes Uzan et al.**

The authors in fact spent a lot of efforts and improved the manuscript significantly. Due to the large number of changes it is almost impossible to check point by point if and how each individual comment of the reviewers was treated in the revised version (pages 39 ff of `acp-2017-634-author_response-version2.pdf`). As a consequence I focus on how convincingly the replies to the reviewers' comments are (taking into account, that the views of both reviewers have been similar but not identical), and on the revised version as a whole (`acp-2017-634-manuscript-version3.pdf`, "V3").

As a result, a few points remain where I suggest a second iteration step to prepare a final version of the paper suitable for publication. I don't comment on typos etc. – this could be left to the typesetting.

1. Abstract: From the title one would expect that ceilometers provide the main contribution to the study. This is not reflected in the abstract anymore. Please highlight the role of ceilometers more clearly.

2. Introduction: In general the authors followed the recommendations of the reviewers. However: check the "Stavros/Solomos"-problem (throughout the text; Stavros is the first name, so use Solomos!). Include a short section on the radiative forcing because this has now been covered in the results-section (as mentioned by reviewer #2 there are several papers in Tellus B special issues of 2009 and 2011).

   In my view the introduction is now sort of long. Certainly, the extension was triggered by trying to fulfill the requirements of review #2. Anyway, check if there is potential to slightly shorten it.

3. Ceilometer section: Include the statement of the limited measurement range more clearly – this was the major criticism of reviewer #2 (and also raised by me) – after (e.g.) line 286. This is indeed mandatory, and is independent on the problems discussed before (overlap, cosmetic shift, etc.): it is an effect of signal attenuation due to reduced atmospheric transmission and happens to all lidar systems (think about a dense cumulus cloud for example – the same effect). Thus, add 1-2 sentences here and refer to them later in the manuscript whenever necessary.

Line 246: Note that the statement in the Vaisala user's guide on the output does not fulfil strict scientific standards: "two-way attenuated backscatter profile with sensitivity normalized units $(100000 \text{ srad km})^{-1}$" is not a physical quantity. These numbers are not the correct definition of attenuated backscatter (see $\beta^*$ in one of the cited Wiegner-papers and the explanations of reviewer #2) as it requires a calibration. Just state in the paper, that the range corrected signal (in arbitrary units) is stored (even this is not necessarily true if the h2-parameter is not set – this is fortunately not relevant for the lowest 2.4 km) and replace all cases of "attenuated backscatter". By the way: the authors correctly mentioned in line 264 that the "real" attenuated backscatter cannot be derived. Final comment (to lines 265ff, "Nevertheless..."): what is the purpose of these sentences: Rayleigh calibration is not possible, or Rayleigh requires averaging over 4 hours? Cloud calibration (see O'Connor et al., 2004) should be used? What is really meant with "background correction"?

Please reconsider my suggestions of the use of the (already listed) citations; they were not properly included. Ansmann et al. (2011) and Papayannis et al. (2008) are not covering ceilometers (but the benefit of lidars for dust observations in general), so these citations do not fit to lines 230–232 in V3. The statement on the water vapour absorption should be more precise (lines 276–278 of V3), maybe something like: "... water vapour distribution has a small effect on the pronounced change of the signal shape at the top of the mixed layer or at boundaries of an elevated aerosol layer (Wiegner and Gasteiger, 2015)". The citation of Mona et al. (2012) is missing in the text.

4. Results section: In context of Fig. 17 the authors refer to "attenuated backscatter". Either this should be changed to something like "range corrected signal with the Vaisala's inherent normalization" or better just "range corrected signal (in arbitrary units)". If the ceilometer's sensitivity remains constant during the event (which is likely) and the effect of changing water vapor absorption is neglected, even uncalibrated signals at a given site can be compared, i.e., discussion of the temporal evolution of the dust at that site is feasible and provides a useful contribution to the paper.

In view of the title of the paper I suggest to extend a little bit the discussion of the ceilometer data: As the MLH cannot be derived from the ceilometer data when the (strong) decrease of the transmission overcompensates the large backscatter coefficients of the dust, at least statements like "dust was present up to at least a height of xxx m" can be made. CALIOP data indicate that the top of the dust layer was typically between 2 and 4 km, however quite variable (in time and space) and sometimes multi-layered. Because of lack of co-located measurements (except the example shown in the revised paper, where some, but not all, ceilometer sites are met) no independent measurements of the actual distribution of the dust are available. Thus, the interpretation of the (upper part of the) ceilometer profiles must remain ambiguous. This can be discussed in the paper. However, when describing the lower boundary of the dust layer, the authors can rely on the ceilometer data (as they do in the revised version).

Line 571: What is meant with a "two-layer shape"? An elevated layer? Similar problem at line 618 ("arc shape ascent").

Fig. 17: It seems that the height above sea level is shown, not the height above ground level as indicated by the y-axis. The figure caption seems to be okay.

Fig. 21: Give wavelength of the CALIOP-data. A blow-up is advisable to better see the situation over Israel. Comment on the "blue range" below 2 km: is this total attenuation? What is the elevation of the ground? What is the green line in the right panel showing?

5. Conclusions: A general message to remember is missing: I suggest to clearly point out the contribution, the strengths, and the limitations of the ceilometer network when observing dust storms (not necessarily as strong as this event). In the paper they was demonstrated for one very strong event, but conclusions that are beneficial for future investigations and (maybe) can be used for other regions should be made.

6. Miscellaneous: refer to Fig. 20 instead of Table 6 in line 355. Change "uncelebrated" to "uncalibrated" in line 281, captions of Figs. 6-13 should be changed to 7.-10. September.

---

## Author Response (AR2)

**Author's Response:**

We thank both referees for acknowledging the significant changes done in the manuscript following their previous comments. We most appreciate the suggestion given in the scope of "minor changes" to better improve the manuscript. A point to point response to each referee is given below.

Sincerely,
Leenes Uzan, Dr. Smadar Egert and Prof. Pinhas Alpert

**Authors' response to referee #1:**

**Referee comment:**
*Abstract: From the title one would expect that ceilometers provide the main contribution to the study. This is not reflected in the abstract anymore. Please highlight the role of ceilometers more clearly.*
**Author's response:** Comment accepted.
**Author's changes in manuscript:** We changed the title and the abstract accordingly.

**Referee comment:**
*Introduction: In general the authors followed the recommendations of the reviewers. However: check the "Stavros/Solomos"-problem (throughout the text; Stavros is the first name, so use Solomos!). Include a short section on the radiative forcing because this has now been covered in the results-section (as mentioned by reviewer #2 there are several papers in Tellus B special issues of 2009 and 2011). In my view the introduction is now sort of long. Certainly, the extension was triggered by trying to fulfill the requirements of review #2. Anyway, check if there is potential to slightly shorten it.*
**Author's response:** Comments accepted.
**Author's changes in manuscript:** Stavros was changed to Solomos. Reference to radiative heating was added to the "Results and discussion" section. The introduction was slightly shortened.

**Referee comment:**
*Ceilometer section: Include the statement of the limited measurement range more clearly – this was the major criticism of reviewer #2 (and also raised by me) – after (e.g.) line 286. This is indeed mandatory, and is independent on the problems discussed before (overlap, cosmetic shift, etc.): it is an effect of signal attenuation due to reduced atmospheric transmission and happens to all lidar systems (think*

*about a dense cumulus cloud for example – the same effect). Thus, add 1-2 sentences here and refer to them later in the manuscript whenever necessary*.

**Author's response:** Comments accepted.

**Author's changes in manuscript:** We added sentences referring to the limitations of ceilometers under dense conditions in the "ceilometers" subsection 2.1.

**Referee comment:**

*Ceilometer section: Line 246: Note that the statement in the Vaisala user's guide on the output does not fulfil strict scientific standards: "two-way attenuated backscatter profile with sensitivity normalized units (100000 srad km)−1" is not a physical quantity. These numbers are not the correct definition of attenuated backscatter (see β ∗ in one of the cited Wiegner-papers and the explanations of reviewer #2) as it requires a calibration. Just state in the paper, that the range corrected signal (in arbitrary units) is stored (even this is not necessarily true if the h2-parameter is not set – this is fortunately not relevant for the lowest 2.4 km) and replace all cases of "attenuated backscatter". By the way: the authors correctly mentioned in line 264 that the "real" attenuated backscatter cannot be derived.*

**Author's response:** Comments accepted.

**Author's changes in manuscript:** We added the following text in "Ceilometers" subsection .2.1: "The internal calibration applied to convert the signal count output to "attenuated backscatter" does not always fully represent the actual lidar constant, therefore, it is not accurate enough for meteorological research. Hence, in this study we defined the ceilometer profiles as range corrected signal profiles in arbitrary units". We changed the "attenuated backscatter profiles" produced in this research to range corrected signal profiles (in a.u.).

**Referee comment:**

*Ceilometer section: Final comment (to lines 265ff, "Nevertheless..."): what is the purpose of these sentences: Rayleigh calibration is not possible, or Rayleigh requires averaging over 4 hours? Cloud calibration (see O'Connor et al., 2004) should be used? What is really meant with "background correction"?*

**Author's response:** Comments accepted.

**Author's changes in manuscript:** We changed the text to make it clearer.

**Referee comment:**

*Ceilometer section: Please reconsider my suggestions of the use of the (already listed) citations; they were not properly included. Ansmann et al. (2011) and Papayannis et al. (2008) are not covering ceilometers (but the benefit of lidars for dust observations in general), so these citations do not fit to lines 230–232 in V3. The statement on the water vapour absorption should be more precise (lines 276– 278 of V3), maybe something like: "... water vapour distribution has a small effect on*

*the pronounced change of the signal shape at the top of the mixed layer or at boundaries of an elevated aerosol layer (Wiegner and Gasteiger, 2015)". The citation of Mona et al. (2012) is missing in the text.*

**Author's response:** Comments accepted.

**Author's changes in manuscript:** We relocated the citation and added the Mona et al. (2012) citation that was unfortunately neglected.

**Referee comment:**

*Results section: In context of Fig. 17 the authors refer to "attenuated backscatter". Either this should be changed to something like "range corrected signal with the Vaisala's inherent normalization" or better just "range corrected signal (in arbitrary units)". If the ceilometer's sensitivity remains constant during the event (which is likely) and the effect of changing water vapor absorption is neglected, even uncalibrated signals at a given site can be compared, i.e., discussion of the temporal evolution of the dust at that site is feasible and provides a useful contribution to the paper.*

**Author's response:** Comments accepted.

**Author's changes in manuscript:** The units in Fig. 17 were changed to range corrected signal profiles in a.u.

**Referee comment:**

*Results section: In view of the title of the paper I suggest to extend a little bit the discussion of the ceilometer data: As the MLH cannot be derived from the ceilometer data when the (strong) decrease of the transmission overcompensates the large backscatter coefficients of the dust, at least statements like "dust was present up to at least a height of xxx m" can be made. CALIOP data indicate that the top of the dust layer was typically between 2 and 4 km, however quite variable (in time and space) and sometimes multi-layered. Because of lack of co-located measurements (except the example shown in the revised paper, where some, but not all, ceilometer sites are met) no independent measurements of the actual distribution of the dust are available. Thus, the interpretation of the (upper part of the) ceilometer profiles must remain ambiguous. This can be discussed in the paper. However, when describing the lower boundary of the dust layer, the authors can rely on the ceilometer data (as they do in the revised version).*

**Author's response:** Comments accepted.

**Author's changes in manuscript:** The ceilometer contribution in the lower part of the atmosphere was better emphasized in the results section.

**Referee comment:**

*Results section: Line 571: What is meant with a "two-layer shape"? An elevated layer? Similar problem at line 618 ("arc shape ascent").*

**Author's response:** We referred to the ununiformed dust layer visible by the ceilometers' plots as two separate layers (descending down to 500 m ASL and rising from ground level). The "arc-shape" was referred to the thin dust layer formed aloft in the shoreline ceilometers following the entrance of the sea breeze front.

**Author's changes in manuscript:** We rephrased "two-layer shape" to " an ununiformed dust layer" and "arc-shape " to "a narrow dust layer".

**Referee comment:**

*Results section: Fig. 17: It seems that the height above sea level is shown, not the height above ground level as indicated by the y-axis. The figure caption seems to be okay.*

**Author's response:** Comment accepted.

**Author's changes in manuscript:** The y-axis was corrected height ASL.

**Referee comment:**

*Results section: Fig. 21: Give wavelength of the CALIOP-data. A blow-up is advisable to better see the situation over Israel. Comment on the "blue range" below 2 km: is this total attenuation? What is the elevation of the ground? What is the green line in the right panel showing?*

**Author's response:** We commented in line 627: "We assume the CALIOP lidar did not produce data beneath 2 km ASL due to total attenuation."
The green line indicates the overpass on 6 September 2015 referred by previous studies showing the distance from Israel. It is omitted in the updated CALIPSO plots.

**Author's changes in manuscript:** We focused the CALIPSO plots on the data produced above Israel. We addressed ground elevation height and mentioned the CALIOP wavelength ( 532 nm)  in the caption of Fig. 21.

**Referee comment:**

*Conclusions: A general message to remember is missing: I suggest to clearly point out the contribution, the strengths, and the limitations of the ceilometer network when observing dust storms (not necessarily as strong as this event). In the paper they was demonstrated for one very strong event, but conclusions that are beneficial for future investigations and (maybe) can be used for other regions should be made.*

**Author's response:** *Comments accepted.*

**Author's changes in manuscript:** We have changed the conclusions section accordingly.

**Referee comment:**

*Miscellaneous: refer to Fig. 20 instead of Table 6 in line 355. Change "uncelebrated" to "uncalibrated" in line 281, captions of Figs. 6-13 should be changed to 7.-10. September.*

**Author's response:** Comments Accepted.

**Author's changes in manuscript:** Corrections were made accordingly.

**Author's' response to referee #2:**

**Referee comment:**

*P13, L355: Global radiation measured … Table 6. …. ? In Table 6, I find PM10 for 31 stations only. Do you mean Table 7 (but I do not have Table 7?)*

**Author's response:** Mistake. The reference is Fig. 20

**Author's changes in manuscript:** The reference was changed from Table 6 to Fig. 20.

**Referee comment:**

*P13, L378: I have generally my problems with the uncertainty statements (here MSG, somewhere else MODIS). I think in the paper of Mamouri they compared MODIS with AERONET for these very large AODs and gave some uncertainty statements. At least, 15% uncertainty appears to me rather low… for these extreme dust conditions with AOD probably even larger than 3… , I think the AOD uncertainty is of the order of 1.0 or even more.*

*P14, L395: Again, an uncertainty of 0.1 is probably ok for AOD< 1.0, but what about cases with >2.5? Then the error is certainly much larger, of the order of 0.5 to 1.0.*

**Author's response:** We did not find explicit estimation of the MSG SEVIRI AOD uncertainty during high AOD values except for a general remark based on a comparison done between SEVIRI AOD and AERONET AOD in the scope of EUMETSAT Scientific Validation Report SEVIRI Aerosol Optical Depth (23 Oct 2017) :" *For very high AOD  conditions (>1.5) a negative bias is observed".* Similar conclusions were found for MODIS AOD. We wish to thank Dr. Yevgeny Derimian from the university of Lille for data and personal correspondence on this issue.

**Author's changes in manuscript:** We added comments of increased uncertainty under high AODs (>1.5).

**Referee comment:**

*Section 3, I would introduce subsections! … with head lines (titles): 3.1 7 September 2015, 3.2 8 September 2015 etc. That would make the full and very complex discussion section easier to read.*

*P15, L426: Now the ceilometer observations are introduced….*

**Author's response:** Comment accepted.

**Author's changes in manuscript:** We added subsection by dates as recommended.

**Referee comment:**

*P10, L286: you state: …. we refer to the ceilometer signal count profiles between 100-1000m. This statement is good, but I am afraid that non-lidar dust-interested readers will not remember that statement when they see Figures 6-12. I will come to this point again, below.*

*P23, L557: .. and now, the discussion of the 8 September starts. As mentioned above, we need a clear statement that the colored areas in Figs 6-12 only show the lowest few hundred meters of the dust layer (which actually reached up to 4-5 km according to all the articles published before: Mamouri, Solomos, Gasch, and all the CALIOP observations, including the one shown in Fig 21, and Fig.21 comes much too late, to my opinion). Without such a clear remark (and this remark is definitely not given), most readers will intuitively think, the ceilometers show the entire dust layer.*

*P24, L590: … which may indicate the dust plume base height….. Again this statement is confusing when having the colored areas observed with ceilometers in mind, which suggest--- TOP HEIGHT. Non-lidar people will not understand what you want to say… without these explanations I suggested above.*

**Author's response:** The "Results and discussion" section is arranged by dates starting from 8 September 2015 to 10 September. The CALIOP observations refer to10 September therefore they were mentioned last.

**Author's changes in manuscript:** We added a short paragraph in the "Results and discussion" section clarifying the colours of the ceilometers' plots in context of the ceilometers' measurement limitations.

**Referee comment:**

*If I compare Fig 9 with Fig 16 (the radiosonde profiles are very nice now!), then my opinion is confirmed that the ceilometers see only the lowest few hundred meters of the dust layer because the top height of the colored ceilometer backscatter areas coincide with the temperature inversion height (the base of the main dust layer, higher up…).*

**Author's response:** Agreed.

**Author's changes in manuscript:** No change.

**Referee comment:**

*Fig 13: This plots shows nothing? Clear skies? Or just overloaded by heavy dust? The plot is almost entirely deep blue! But there should be a lot of dust according to all the satellite observations presented. So, maybe an APD, PMT overloading effect. The ceilometer detectors were just overloaded because of the huge amount of dust around? Please clarify.*

**Author's response:** The deep blue scale evident in all Mount Meron ceilometer plots (Fig. 13) indicate total attenuation distinctively from 7 September ~ 14 UTC to 8 September ~ 16 UTC. Due to the complexity of the dust plume progress and the weak signal counts shown up to 3.5 km ASL (before 7 September ~ 14 UTC and after 8 September ~ 16 UTC), the assumption of a total attenuation throughout the period analysed is uncertain. Unfortunately, we did not have auxiliary measurements from the Mount Meron region to justify our assumptions.

**Author's changes in manuscript:** We added the aforementioned assumption in the manuscript.

**Referee comment:**

*Fig 17: Now the quantitative analysis is much improved. But I want to say, just that you think about it….: An attenuated backscatter coefficient of 10-4 m-1 sr-1 is equivalent to 4 km-1 extinction coefficient (for a lidar ratio of 40sr), and thus 2x10-4 means 8 km-1 extinction. Do you have the feeling, this is correct? I just ask you! Do you believe in your numbers?*

**Author's response:** Unfortunately Israel does not poses a calibrated lidar capable of true extinction measurements. However, previous study of the dust event in Cyprus by Raman lidar (Mamouri et al., 2016) provided extinction coefficient with similar order of magnitude. Based on the CL31 manual the ceilometer is capable to measure visibility values in clouds in the range of 15-150 m (dust visibility was estimation by IMS observers to be ~ 200 m) and hence, measurement capabilities within this range.

**Author's changes in manuscript:** No change.

**Referee comment:**

*The other point: Fig 17 nicely shows that backscatter signals (from five of the seven ceilometers) which are backscattered from heights above 500-700m were completely attenuated so that the attenuated backscatter coefficient is simply zero. That does not mean that the backscatter coefficient was zero…… The backscatter coefficient was probably even larger than at ground. So, attenuated backscatter is a very 'dangerous' parameter.*

**Author's response:** The attenuated backscatter profiles are the ones available from the ceilometers. Nevertheless, following the referees' recommendations and changed the "attenuated backscatter" units to " range corrected signal profiles (in a.u.)".

**Author's changes in manuscript:** "Attenuated backscatter" units were changed to " range corrected signal profiles in a.u. ".

**Referee comment:**

*P23, L571: … reveal a two-layer shape …. How do you know, there may have been even 4 or 5 distinct layers up to 4-5 km height.*

**Author's response:** We referred to the layers visible by the ceilometers (below ~ 1 km).

**Author's changes in manuscript:** We rephrased "two-layer shape" to " an ununiformed dust layer"

**Referee comment:**

*Fig.20: Black numbers on dark blue or dark brownish background… any idea to improve that?*

**Author's response:** Comment accepted.

**Author's changes in manuscript:** We changed the color of the numbers in figures 19-20 from black to white where the background was dark.

**Referee comment:**

*P26, L625: The dust top height of 2-4 km is mentioned here for the first time (if I read the paper carefully enough). This is simply much too late. And the overall context given by the papers of Mamouri et al., Solomos et al. (presenting CALIOP obs.), Gasch et al. (also presenting CALIOP obs.) clearly shows that the top height was always and everywhere at 4-5km height, in full agreement with your Fig 21 for 10 September. So, please improve…. Not 2-4 km, it was 4-5 km …*

**Author's response:** Comments accepted.

**Author's changes in manuscript:** We rephrased the sentence to " ...a dust layer up to 5 km ASL".

**A list of relevant changes made in the manuscript:**

1. Updated abstract
2. The introduction was shortened (Sect. 1)
3. The structure of "Results and discussion" section was changed.
4. The "Conclusions" section was broadened.
5. Several figures' presentation, unites and axis were changed.

[revised manuscript text omitted]